

# Impact of physical parameterizations on wind simulation with WRF V3.9.1.1 over the coastal regions of North China at PBL gray-zone resolution

Entao Yu[1,2], Rui Bai[1,3], Xia Chen[4,5], Lifang Shao[4]

[1] Nansen-Zhu International Research Centre, Institute of Atmospheric Physics, Chinese Academy of Sciences, Beijing, China
[2] Collaborative Innovation Center on Forecast and Evaluation of Meteorological Disasters, Nanjing University of Information Science and Technology, Nanjing, China
[3] University of Chinese Academy of Science (UCAS), Beijing, China
[4] Hebei Climate Center, Shijiazhuang, China
[5] Hebei Key Laboratory for Meteorology and Eco-environment, Shijiazhuang, China

*Correspondence to*: Entao Yu (yuet@mail.iap.ac.cn)

**Abstract.** Reliable simulation of wind patterns that form under stable weather conditions is vital to prevent air pollution. Here, we investigated how different physical parameterization schemes impact surface wind simulations over the coastal regions of

North China using the Weather Research and Forecasting (WRF) model with a horizontal resolution of 0.5 km. We performed 640 ensemble simulations using multiple combinations of 10 planetary boundary layer (PBL), 16 microphysics (MP), and four longwave/shortwave (LW/SW) radiation schemes. Model performance was evaluated using measurements from weather station observations. These data show that WRF model can reproduce the temporal variation of wind speed and direction to a high degree of accuracy. The simulated wind speed is most sensitive to the PBL schemes, followed by LW/SW radiation

schemes and MP schemes, while wind direction is less sensitive to variation of the physical parameterizations. Among all PBL schemes, the MYJ scheme shows the strongest correlation with observation data, while the YSU scheme had the smallest model bias. A combined Dudhia and RRTM scheme and MYDM7 scheme show the best model performances out of all LW/SW radiation and MP schemes, respectively. Our results show that the interactions among physical components also play an important role in wind simulations. Further investigations indicate that land type has a profound influence on model

sensitivity. For example, for the weather stations located in coastal regions. The MYNN scheme showed the highest correlation among all PBL schemes, while LES and YSU has the smallest model errors, the RRTMG and Goddard schemes showed the highest correlations and smallest biases out of all LW/SW radiation schemes. Our results indicate the roles that various parameterization schemes play in wind simulations under stable weather conditions, and provide a valuable reference for further research in this region and in other locations around the world.



## 1 Introduction

Megacities that experience rapid urbanization and economic development also commonly suffer from a simultaneous decline in air quality (Ulpiani, 2021). For example, numerous haze events have been reported in the Beijing, Tianjin, and Hebei regions of North China over the past few decades. Haze-related weather and associated high concentrations of fine particulate matter have negative impacts on public health and the environment (Wang and Mauzerall, 2006). These events can significantly

disrupt economic growth, as demonstrated by the severe haze events that occurred over North China in January 2013 (Zhang et al., 2014; Cai et al., 2017; Zhang et al., 2015). These events are most frequent in boreal winter and are closely related to local weather conditions, with haze forming in regions with low wind speeds (Wang et al., 2021; Li et al., 2015). Projections of future climate change suggest that global temperatures will increase, which in turn may increase the frequency of haze events over North China (Cai et al., 2017). Therefore, it is crucial to improve predictions of wind field behavior under stable

weather conditions in order to minimize associated economic losses and environmental impacts.

In recent years, numerical models have been used extensively to study the weather and climate over China, as they have high spatial and temporal resolutions, and employ sophisticated physical parameterization schemes that can reproduce detailed atmospheric and land surface processes (Wang et al., 2011; Kong et al., 2021; Zhou et al., 2019). However, these models mostly focus on temperature or precipitation, and only a few studies have attempted to simulate wind patterns over China (Li

et al., 2019; Pan et al., 2021; Xia et al., 2019). Furthermore, numerical models inherently involve many sources of uncertainty, as they cannot resolve all processes that occur in the real world; instead, parameterizations are needed to represent the effect of key physical processes, such as radiative heating, planetary boundary layers, and cloud microphysics. Different physical parameterization packages reproduce natural phenomena to different degrees of accuracy and choosing appropriate combinations is extremely important, as this decision strongly influences the model simulation results (Yu et al., 2011; Gao et

al., 2016; Yang et al., 2017; Taraphdar et al., 2021; Gómez-Navarro et al., 2015; Stegehuis et al., 2015).

The impact of planetary boundary layer (PBL) schemes on wind simulations has been studied in recent years, as PBL schemes play a critical role in modulating mass, energy, and moisture fluxes between the land and atmosphere, which in turn influence the simulation of low-level temperatures, cloud formation, and wind fields (Falasca et al., 2021; Gholami et al., 2021; Gómez-Navarro et al., 2015; Jiménez and Dudhia, 2012; Gonçalves-Ageitos et al., 2015). Gómez-Navarro et al. (2015) investigated

the sensitivity of WRF model to PBL schemes by simulating wind storms over complex terrain at a horizontal resolution of 2 km. In that study, the WRF model was combined with the Mellor-Yamada-Janjic (MYJ) scheme and overestimated wind speed by up to 100%, however, this bias was significantly reduced when the non-local scheme developed at Yonsei University (YSU) was used instead. Other studies conducted over the northeastern Iberian Peninsula, Persian Gulf, Tyrrhenian coast, and western Argentina produced similar results (Falasca et al., 2021; Jiménez and Dudhia, 2012; Puliafito et al., 2015; Gholami et al.,

2021), although some studies have suggested that the MYNN and ACM2 schemes are more appropriate for wind simulations (Prieto-Herráez et al., 2021; Rybchuk et al., 2021; Chang et al., 2015; Carvalho et al., 2014b).





The performance of numerical models simulating wind patterns is also affected by the choice of cloud microphysics (MP) parameterization scheme. Cloud microphysical processes, such as moisture evaporation and condensation, can affect thermodynamic and dynamic interactions in the atmosphere (Santos-Alamillos et al., 2013; Rajeevan et al., 2010; Li et al.,
2020), and so affect the vertical distribution of heat and the behavior of wind fields close to the Earth's surface. Recently, Cheng et al. (2013) reported that predictions of summer wind speeds in northern Colorado were strongly affected by the choice of cloud MP parameterization scheme, and that the WRF double-moment 6-class (WDM6) scheme performed best. Another factor that influences the veracity of wind simulations is the choice of radiation parameterization scheme, which includes shortwave (SW) radiation and longwave (LW) radiation. Differences in surface radiation intensities can generate thermal
contrasts in regions with complex topography, which in turn affect local and low-level wind distribution patterns (Santos-Alamillos et al., 2013).

Most of the aforementioned studies only considered a small number of combinations of physical parameterization schemes, meaning that the sensitivity of physical processes on simulated wind patterns has not yet been explored in a systematic way. Performing such an investigation is important, as error compensation among processes that involve low-level atmosphere–
land interactions, radiation, and clouds may predict incorrect wind distribution patterns. In this study, we systematically evaluated the performance of 640 combinations of physical parameterization schemes, including PBL, MP, and SW/LW radiation processes. This was conducted using WRF model at a very high resolution of 0.5 km, which belongs to the gray zone for the PBL scale. Our main objective was to identify a set of multi-physics configurations of the WRF model that can best reproduce wind fields under stable weather conditions. This study addressed the following research themes: (1) quantify the
sensitivity of simulated wind field under stable conditions to different physical parameterization schemes, and (2) refine an optimized model configuration of WRF model to reproduce the wind field. These results provide valuable insights into model performance, and can be used to produce more reliable and accurate wind and air quality forecasting. The founding reported here may also be applied to other regions around the world.

This paper is structured as follows: section 2 describes the model setup and the evaluation data, model results are reported in
section 3, and discussion and concluding remarks are given in section 4.

## 2. Description of the study area, model, and experimental design

### 2.1. Study area and stable weather events in 2019

The study area is located in the central section of the "Bohai Economic Rim", which is bordered to the southeast by the Bohai Sea and to the northwest by the Yan Mountains (Figure 1a). This region traditionally hosts heavy industry and manufacturing
businesses, and is a significant region of economic growth and development in North China.

Air quality in this area has declined over the past few decades, and the frequency of winter haze events has increased due to increased pollutant emissions and favorable weather conditions. For instance, during the severe haze events over eastern China in January 2013, an anomalous southerly wind in the lower troposphere caused by the weak East Asian winter monsoon





weakened the synoptic disturbances and extent of vertical mixing the atmosphere, thus increasing the stability of surface air

and favoring the local concentration of hazes (Zhang et al., 2014).

A severe haze event occurred in the study area during January 11th to January 15th, 2019. At this time, the peak PM2.5 and PM10 concentrations measured in Tangshan city, located in the center of the study region, exceeded 279 μg/m3 and 357 μg/m3, respectively. The haze event also occurred under stable weather conditions with an average wind speed lower than 5 m/s (Figure 2a). The wind rose for this event showed a primarily southerly wind at the surface (Figure 2b), indicating a weakened

East Asia winter monsoon and stable lower troposphere. This event was used to evaluate the performance of the physical parameters of the WRF model, as shown below.

## 2.2 WRF model configurations

WRF model (version 3.9.1.1) with an advanced research WRF (ARW) core was used in this study, which is a non-hydrostatic atmospheric model with terrain-following vertical coordinates (Skamarock et al., 2008). The simulation contained three one-

way nested domains with horizontal resolutions of 8 km, 2 km, and 0.5 km for D01, D02, and D03, respectively (Figure 1b). The computational domains were based on a Lambert conformal conic projection centered at 38.5°N and 120°E, with 360 × 480, 381 × 381, and 341 × 421 grid points for D01, D02, and D03, respectively. The following analysis evaluates model performance based on the innermost domain, which covers the coastal and surrounding regions of North China. This domain had 61 vertical levels, which ensured that several levels existed below the PBL height at any time.

The ERA5 reanalysis dataset has a horizontal resolution of 0.25° and 38 vertical levels, and was used as to define the initial and boundary conditions for all simulations. The WRF model was initialized at 00:00 UTC January 9th, 2019 and integrated until 00:00 UTC January 16th, 2019, with the first 40 hours treated as a spin-up period. Default physical parameterizations were applied in the single set of WRF model simulations for the outer two domains, and the output from these domains was used to drive inner domain simulations with different combinations of PBL, MP, and SW/LW radiation parameterization

schemes (see section 2.3). This approach helped to isolate the impacts of parameterization within the inner domain from changes in boundary forcing (Yang et al., 2017). No other parameterizations were changed in any of the other simulations, including the Noah-MP land surface scheme (Yang et al., 2011; Niu et al., 2011). The lateral boundary and sea surface temperature were updated every three hours, and the modeled frequency of wind was calculated hourly, which matches the frequency of observations in the study area.

## 2.3 Experimental design

The WRF model contains different components that represent different physical processes (e.g. PBL, MP, and LW/SW radiation). Further, every physical process in the model has many parameters, such that a model can range from being simple and efficient to sophisticated and computationally costly. A comprehensive test of all parameterization schemes was achieved by considering 10 PBL schemes, 16 MP schemes, and four LW/SW radiation schemes, which produced a total of 640 (i.e. 10

× 16 × 4) combinations that were investigated in this study.



The PBL schemes investigated in this study are listed in Table 1. The horizontal resolution of 0.5 km is within the PBL "gray zone" grid spacing that is too fine to utilize mesoscale turbulence parameterizations and too coarse for a large-eddy-simulation (LES) scheme to resolve turbulent eddies; therefore, both PBL and LES assumptions are imperfect at this resolution. In this study, both PBL and LES schemes were tested. For the LES configuration, a 1.5-order turbulence kinetic energy closure model

was used to parameterize motion at the sub-grid scale. For the YSU scheme, a topographic correction for surface winds was included to represent extra drag from sub-grid topography and enhanced flow at hill tops (Jiménez and Dudhia, 2012). The option for top-down mixing driven by radiative cooling was also turned on during the integration. For the rest of the PBL schemes, the default configurations were chosen.

The atmospheric surface layer (SL) is the lowest part of the atmospheric boundary layer. Calculated friction velocities and

exchange coefficients within this SL were used to quantify surface heat and moisture fluxes in LSM schemes, alongside surface stress in the PBL scheme. In the current generation of models, the SL options are tied to the PBL options. Consequently, the ETA, QNSE, MYNN, Pleim-Xiu, and TEMF SL schemes were chosen separately for the MYJ, QNSE, MYNN, ACM2, and TEMF PBL schemes. The revised MM5 scheme was used for all other PBL schemes.

Sixteen MP schemes were used in this study, as documented in Table 2. Lin scheme is a sophisticated scheme suitable for

high-resolution simulations of real data. The WSM3, WSM5, WSM6, and WDM6 schemes share a similar underlying theory, and WSM6 and WDM6 are suitable for high-resolution simulations. The ETA scheme represents the operational microphysics behavior in NCEP models. Goddard, Thompson, Morrison, NSSL2, and NSSL1 are new schemes suitable for high-resolution simulations. The MYDM7 scheme includes separate categories for hail and graupel, and models double-moment clouds, rain, ice, snow, graupel, and hail. Finally, CAM is a double moment 5-class scheme, and ThompsonAA considers water- and ice-

friendly aerosols.

Four combinations were selected for LW/SW radiation schemes. Dudhia is a simple and efficient scheme for clouds and clear-sky absorption and scattering; the RRTM scheme provides accurate and efficient look-up tables; and the CAM scheme is derived from the CAM3 climate model used in CCSM, and allows modeling of aerosols and trace gases. Finally, RRTMG is a new LW/SW scheme that utilizes a Monte Carlo independent column approximation (MCICA) method of random cloud

overlap.

### 2.4 Observational data and evaluation metrics

Observations from weather stations across the study region were used to evaluate the performance of the model. These stations are operated by the China Meteorology Administration (CMA), and report wind speed and direction at an altitude of 10 m. In this study, we used two-minute-averaged wind speed at hourly frequency. All data that were collected were screened before

analysis in order to remove stations with data showing spurious jumps. After this filtering, 105 weather stations (Figure 1a) remained. The results of WRF simulations were directly compared with observations made at each weather station, which was achieved by using the model result that was geographically closest to the weather station under consideration. Although some





errors are introduced when performing these comparisons, they are systematic and shared by all simulations, and therefore have minor effects on the evaluation of model performances.

Several metrics were employed for evaluating the performance of each model configuration, including the Pearson's correlation coefficient (CORR), BIAS, root mean square error (RMSE), and the Taylor skill score (T). These are defined as follows:

$$CORR = \frac{\sum(m_i - \overline{m}) \cdot \sum(o_i - \overline{o})}{\sqrt{\sum(m_i - \overline{m})^2 \cdot \sum(o_i - \overline{o})^2}}$$

$$BIAS = \sum(m_i - o_i)$$

$$RMSE = \sqrt{\frac{\sum(m_i - o_i)^2}{N_i}}$$

$$T = \frac{2(1 + CORR)}{(SD + 1/SD)^2}$$

Here, $m_i$ is the value of the model output, $o_i$ is the value of the observation, $N_i$ is the number of observations, and SD is the ratio of simulated to observed standard deviations. The Taylor skill score ranges from zero to one, and a higher score indicates a more accurate simulation (Gan et al., 2019).

The difference in wind direction was calculated as follows:

$$\Delta = \begin{cases} m - o, & \text{when } |m - o| < 180° \\ (m - o)(1 - \dfrac{360}{|m - o|}), & \text{when } |m - o| > 180° \end{cases}$$

The correlation between simulated and measured angles was determined by a circular correlation coefficient (CORR), which was calculated as follows:

$$CORR = \frac{\sum \sin(\alpha - \bar{\alpha}) \sin(\beta - \bar{\beta})}{\sqrt{\sin^2(\alpha - \bar{\alpha}) \sin^2(\beta - \bar{\beta})}}$$

Here, α and β are simulated and observed wind direction angles, respectively.

## 3. Results

Simulated wind speeds and directions were compared with observations recorded by CMA weather stations. The comparison presented here is based on the average of all 106 weather stations, which provides general evaluation of the model performance when using different physical parameterization schemes. The section below first analyzes the impact of physical

parameterization schemes, and then investigates the effects of topography and land type.





### 3.1 Sensitivity to physical parameterization schemes

### 3.1.1 Impact of PBL parameterization

Figure 3 shows the time series of observed wind speeds and simulated data produced by different PBL schemes. The WRF model closely reproduces the temporal variation of observed wind speed in the study area; in particular, the shift from low to
high values recorded on January 14[th], 2019, is reproduced by all schemes except for QNSE, which showed no obvious daily wind speed change during the simulation period. Despite this correlation, simulations generally overestimated the wind speed, with absolute differences exceeding 10 m/s. The overestimation of wind speed in such models has been widely reported by previous studies performed in other locations and using various WRF configurations (Jiménez and Dudhia, 2012; Carvalho et al., 2014a, b; Pan et al., 2021; Gholami et al., 2021; Dzebre and Adaramola, 2020). The QNSE scheme is a clear outlier by
having model results that deviate significantly from observation data. This indicates that the scheme fails to reproduce reasonable wind speeds, although the other nine PBL schemes performed much better to similar degrees. Further comparison indicates that model BIAS value was relatively smaller for LES configurations during the first three days of simulation time, while YSU showed a smaller BIAS value during the final two days of the simulation. The differences among different PBL schemes were relatively large in January 15[th], 2019, partly due to faster observed wind speeds (>4 m/s).

Table 4 shows statistics for wind speed data calculated for different PBL schemes. In each case, bold and underlined data represent the highest CORR values, the smallest BIAS values, and the smallest RMSE values. WRF configurations that used the MYJ scheme showed the strongest correlations with observed data (0.956), followed by those that used MYNN, ACM2, UW, YSU, and the Shin-Hong schemes (>0.93). However, evaluation of model errors showed that the YSU scheme had a BIAS of 0.448 m/s and RMSE of 1.585 m/s, showing that it performed best out of all combinations. Simulations performed
with the MYNN scheme show similar statistics to those with the YSU scheme. For the QNSE scheme, the extremely low CORR value (0.643) and high degree of error (BIAS = 3.218 and RMSE = 4.839) indicates that the scheme is not useful to model stable weather conditions in the study area. Its failure to reproduce wind speed data may be related to the eddy diffusivity mass-flux method used by QNSE during the daytime, which produces unrealistically fast wind speeds when compared with observed values (Figure 3). As such, all simulations using the QNSE scheme (64 simulations in total) were omitted from
further investigation in order that these anomalous data did not affect our overall analysis.

Figure 4 presents a wind rose showing observed and simulated data produced via different PBL schemes. The simulated wind directions were very similar in all schemes, indicating that it is insensitive to changes in physical parameterizations; however, the main wind direction observed during this period was south, while the simulations primarily showed a southwesterly wind, suggesting that almost all PBL schemes have a clockwise bias in the modeled wind direction. As the simulated wind direction
was calculated using the wind speed, the bias in modeled wind direction can be attributed to bias in the wind speed simulation. Table 5 shows statistics for wind direction determined for different PBL schemes. Correlation coefficients for wind direction are generally lower than those for wind speed. The WRF configuration using the BouLac scheme shows the highest value of CORR (0.488), and LES simulations have the smallest RMSE values. As mentioned previously, the simulated wind directions





are not sensitive to the physical schemes chosen. Consequently, the model RMSE and BIAS values for different PBL schemes

are similar, and so the following analysis mainly focuses on the wind speed simulations.

### 3.1.2 Impact of microphysics parameterization

Figure 5 shows the time series of observed wind speeds and simulated data produced using different MP schemes. Data from these simulations are similar, especially during the latter parts of the analysis period. The spread of simulation data is smaller than that shown by the PBL schemes, indicating that MP schemes have a smaller influence on surface wind speed. Nonetheless,

some differences still exist among the MP schemes.

Statistics indicate that WRF configurations using the MYDM7 scheme show the highest CORR values (0.944), smallest BIAS values (0.660 m/s), and smallest RMSE values (1.788 m/s) when compared with observed wind speeds (Table 6). As such, this scheme showed the best model performance for wind speed simulation. Simulations performed with the ETA, Goddard, NSSL1, Thompson, P2, and Lin schemes show CORR values that are similar to MYDM7, and simulations performed with the

P3, NSSL1, and ETA schemes have BIAS values that are slightly higher than that for MYDM7.

Simulated wind directions were similar in all cases, and models using the Goddard scheme showed the highest CORR values. By contrast, WRF configurations using the MYDM7 scheme had the smallest model error (i.e., BIAS and RMSE values).

### 3.1.3 Impact of radiation parameterization

Figure 6 shows the time series of observed wind speeds and simulated data, where it can be seen that simulated wind speeds

are more sensitive to LW/SW radiation parameters than MP schemes. The LW/SW schemes have a larger model spread, but are still less sensitive than PBL schemes. Simulations with the RRTMG and CAM schemes overestimated the wind speed, especially for the peak values during the daytime. Conversely, the WRF configurations using the Dudhia and RRTM schemes substantially reduced this overestimation, and thus produced values that were much closer to weather station observations. Table 7 lists wind speed and wind direction statistics for these models. For simulated wind speeds, the Dudhia and RRTM

schemes produced the best model results, as defined by having the highest CORR values (0.943), the smallest BIAS values (0.646 m/s), and smallest RMSE values (1.754 m/s). Simulations that employed the Goddard scheme produced similar wind speeds. For simulated wind directions, WRF configurations that used the CAM scheme had the highest CORR and lowest RMSE values, while simulations that employed the RRTMG scheme showed the lowest BIAS values.

### 3.1.4 Impact of physical components interactions

Interactions among physical components also play an important role in wind simulations. Since it is not possible to show or discuss all possible combinations of PBL, MP, and LW/SW radiation schemes in this study, the results of interactions between PBL and LW/SW radiation schemes are used as an example (Figure 7). All simulations shown in Figure 7 employed MYDM7 as the MP scheme, given that it recorded the best performance in earlier examples (see section 3.1.2). A total of 40 simulations were evaluated in this way, which produced outcomes that were fully consistent with other results. For wind speed simulations,



as illustrated in previous analyses, WRF configurations that used the MYJ scheme showed the highest CORR values, and the YSU scheme produced the smallest model errors (Table 4). However, deeper investigation showed that the YSU scheme produced lower BIAS and RMSE values only when combined with the Dudhia and RRTM schemes together, or the Goddard scheme by itself. When it was combined with the RRTMG scheme, the simulation errors were larger than those for the MYNN and combined Dudhia and RRTM schemes. Thus, for each physical scheme, the choice of a suitable combination of related

physical components is equally important to optimize model performance. These data show that simulations conducted using schemes with high correlation coefficients for PBL (MYJ), MP (MYDM7), and LW/SW radiation (Dudhia and RRTM) were not necessarily the same ones that had the highest individual values of CORR in all other simulations. The same result can be seen for wind direction simulations (Figure 7).

### 3.1.5 Evaluation of model configurations with the best performance

Taylor skill scores for wind speed were calculated for all simulations. The statistics for the 10 best WRF configurations that showed the highest skill scores are listed in Table 8. The PBL and LW/SW radiation schemes used in this set were YSU and a combination of Dudhia and RRTM, respectively. This indicates the significant influence of PBL and radiation schemes on wind speed simulations when compared with the MP schemes, and highlights the better model performance of them both. Because the Taylor skill score considers both correlation and stand deviation, the scheme with the highest Taylor skill score

value (i.e., WDM6) is not the scheme that had the highest CORR value, lowest BIAS value, or lowest RMSE value. Indeed, there was no scheme that had both the highest CORR value and lowest model error (i.e., BIAS and RMSE values). Thus, ensemble simulations are needed to improve model performance, although the number of members used in the ensemble also plays an important role in the determining its performance. Table 8 also list the statistics of the ensemble means of the top two, three, four, five, seven, and 10 simulations, as well as a super ensemble of all the simulations (excluding the QNSE scheme).

These results show that the ensemble mean of the top four schemes (WDM6, Goddard, MYDM7, and NSSL1) performs best and has the smallest model error. Additionally, the ensemble of the top configurations reduces model bias by approximately half, while its CORR value was highest of all the simulations. Figure 8 shows a time series for wind speed determined by the ensemble mean of the top four schemes, which was compared against data for all simulations. The general simulations show a large spread, whereas the ensemble mean of the top four schemes reduced the model bias.

**3.2 Performance dependency on land type and topography**

Land surface conditions (e.g., land–sea type or topography) can affect the partitioning of sensible and latent heat fluxes, which therefore impacts local low-level circulation patterns and the wind distribution in a region. The weather stations in the study region were classified into different groups according to their underlying surface type and topography. The effects of these parameters on the model results are presented below.





### 3.2.1 Comparison between coastal and inland stations

Figure 9 compares the results of simulations performed at coastal and inland stations. The locations of these stations are shown in Figure 1a. Simulations from all stations indicate that modeled wind speed is most sensitive to the PBL scheme chosen, followed by LW/SW radiation and MP schemes, which is consistent with the results of previous analyses. The WRF model commonly overestimates wind speed at both coastal and inland stations, and generally by the same magnitude. All simulations could reproduce the temporal variations of wind speed during the simulation period; however, the model spread was relatively larger for coastal stations, especially for the first three days of the simulation period that exhibited low wind speeds. As such, land–sea interactions represent another source of model uncertainty, which generates greater model differences in wind simulations performed under stable weather conditions.

Table 9 presents statistics for simulations that considered differences between coastal and inland stations. One notable result is that CORR values for coastal stations are consistently lower than those for inland stations. Additionally, as there are no clear differences in wind speed values and variations, the difference in these CORR values indicates that models are generally less reliable over coastal regions. Thus, special attention should be paid to land–sea interactions in future model developments.

Our comparison of parameterization schemes shows that the results produced for inland stations are generally consistent with those of previous investigations, and that the same schemes show the best statistical scores. Nonetheless, the ranking of scheme performance for coastal stations was different. For instance, when comparing PBL schemes, the MYNN scheme showed the highest CORR value, while LES had the smallest BIAS value and YSU had the smallest RMSE value. However, for LW/SW radiation, the RRTMG and Goddard schemes showed the highest CORR values, but had the smallest BIAS and RMSE values. Finally, the combined Dudhia and RRTM scheme, which performed best in previous comparisons, showed poor temporal relationships with observational data in this study.

### 3.2.2 Comparison among stations with different topography

A comparison was also made between stations with different elevations (below 50 m, between 50 m and 250 m, and above 250 m) (Figure 10). Our results show that wind speed reduced in tandem with the station's elevation; for example, the peak modeled wind speed at high-elevation stations (>250 m) was 1 m/s slower than that for low-elevation stations (<50 m). However, the simulated peak values were generally the same, which created a large model bias above the high-elevation stations. As shown in Table 10, calculated BIAS and RMSE values for high-elevation stations were almost twice as large as those for low-elevation stations.

Interestingly, model performances were generally similar for stations with different elevations, and the combination of schemes that provided an optimized performance was mostly consistent with previous analyses. For example, MYJ was the PBL scheme with the highest CORR value, and the YSU scheme recorded the smallest BIAS and RMSE values out of all stations. Thus, as the WRF model uses terrain-following coordinates, surface topography does not induce new uncertainties into the simulation, although land–sea interaction does, as shown in section 3.2.1.





## 4. Conclusions and discussion

In this study, we investigated the impacts of physical parameterization schemes on simulated wind fields that generate under stable weather conditions. This was achieved by performing sensitivity simulations over the coastal regions of North China,
which were characterized by a horizontal resolution of 0.5 km and considered PBL, MP, and LW/SW radiation physical components. A total of 640 simulations were conducted, which considered combinations of 10 different PBL schemes, 16 different MP schemes, and four different LW/SW schemes. The influence of these parameterization schemes on simulated wind speeds and directions were analyzed using weather station observations as validation data. Further investigations considering the underlying land type and topography were conducted to provide more complete insight into model sensitivities.
In general, the WRF model reproduced the temporal variation of wind speed and direction over the study area to a high degree of accuracy. The simulated wind speed is most sensitive to the PBL scheme chosen, and to a lesser degree to the LW/SW radiation and MP schemes. Simulated wind direction is notably less sensitive to the choice of parameterization. This result is consistent with the findings of previous simulations performed in other locations (Dzebre and Adaramola, 2020; Gómez-Navarro et al., 2015; Santos-Alamillos et al., 2013).
The MYJ scheme showed the closest correlation with observation data among all of the considered PBL schemes, while the YSU scheme exhibited the smallest values for BIAS and RMSE. However, the QNSE scheme, which produced unrealistic model errors during daytime periods, is not suitable for the study area. The combined Dudhia and RRTM schemes, and the individual MYDM7 scheme both show the best model performances when compared with other LW/SW radiation and MP schemes, as defined by their high CORR values, low BIAS values, and low RMSE values. Interactions among physical
components also play an important role in wind simulations, and can either increase or decrease the accuracy of the simulation. Choosing the appropriate number of members in an ensemble is vital for precise wind simulation; here, an ensemble mean comprised of the best four individual configurations substantially reduced overestimation of wind speed and provided the best combined performance of all simulations.

The conclusions stated above are directly applicable for inland weather stations; however, for coastal stations, the MYNN
scheme showed the highest CORR value of all PBL schemes, while the LES and YSU schemes showed the smallest BIAS and RMSE values, respectively. The RRTMG and Goddard schemes showed the highest CORR and smallest BIAS and RMSE values of all LW/SW radiation schemes. Our modeling also showed that performances are generally the same for stations located at different elevations.

We conducted an innovative and thorough evaluation of model performance for wind simulation during a typical severe
pollution event over North China. Further, to test the sensitivity of the model, we considered all possible combinations of physical parameterization schemes that are available. This model also used a horizontal resolution belonging to the PBL "gray zone", which has rarely been used in previous simulation studies. The findings of this study therefore provide a foundation for future research in this region, as well as a benchmark for investigation of other localities worldwide.



Finally, we note that the ensemble mean results reproduced the characteristics of the wind field more accurately than those of
individual simulations, which indicates that constructing an ensemble was a more effective technique for optimizing model
performance in this case. However, this does not imply that ensemble studies always provide a better and deeper understanding
of related physical processes and interactions. Additional tuning of the parameters within key PBL schemes, such as YSU, are
needed to achieve this goal, which is the focus of our future work.

**Code and data availability**

The Weather Research and Forecasting (WRF) model is freely available online and can be downloaded from the page:
https://github.com/wrf-model/WRF.          The          ERA5          data          are          available          at          ECMWF
(https://www.ecmwf.int/en/forecasts/datasets/reanalysis-datasets/era5). The observations and model's hourly output upon
which this work is based are available at Zonodo (https://doi.org/10.5281/zenodo.6505423), all the results can also be obtained
from yuet@mail.iap.ac.cn.

**Author contributions**

EY conceptualized the study and conducted the simulations. The analysis was carried out by all the authors. The original draft
of the paper was written by EY, all the authors took part in the edition and revision of it.

**Competing interests**

The authors declare that they have no conflict of interest.

**Acknowledgements**

This work was supported by the National Natural Science Foundation of China (No. 42088101 and 42075168), the Technology
Innovation Guidance Program of Hebei Province (No. 21475401D) and the National Key Research and Development Program
of China (No. 2020YFF0304401). The simulation was supported by the National Key Scientific and Technological
Infrastructure project "Earth System Numerical Simulation Facility" (EarthLab).

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





**Table 1: PBL scheme options investigated in this study.**

| No | Scheme | Reference |
|----|--------|-----------|
| 1 | LES | (Mirocha et al., 2010) |
| 2 | Yonsei University (YSU) | (Hong et al., 2006) |
| 3 | Mellor–Yamada–Janjic (MYJ) | (Janjić, 1994; Mesinger, 1993) |
| 4 | Quasi-normal scale elimination (QNSE) | (Sukoriansky et al., 2005) |
| 5 | Mellor–Yamada Nakanishi Niino (MYNN) | (Nakanishi and Niino, 2009) |
| 6 | Asymmetric convection model 2 (ACM2) | (Pleim, 2007) |
| 7 | Bougeault–Lacarrere (BouLac) | (Bougeault and Lacarrere, 1989) |
| 8 | University of Washington (UW) | (Bretherton and Park, 2009) |
| 9 | TEMF | (Angevine et al., 2010) |
| 10 | Shin-Hong scale-aware | (Shin and Hong, 2015) |






**Table 2: MP scheme options investigated in this study.**

|    | Scheme | Reference |
|----|--------|-----------|
| 1  | Purdue Lin (Lin) | (Chen and Sun, 2002) |
| 2  | WRF single-moment 3-class (WSM3) | (Hong et al., 2004) |
| 3  | WRF single-moment 5-class (WSM5) | (Hong et al., 2004) |
| 4  | ETA Ferrier | (Rogers et al., 2001) |
| 5  | WRF single-moment 6-class (WSM6) | (Hong and Lim, 2006) |
| 6  | Goddard | (Tao et al., 1989) |
| 7  | Thompson | (Thompson et al., 2008) |
| 8  | Milbrandt-Yau double-moment 7-class (MYDM7) | (Milbrandt and Yau, 2005) |
| 9  | Morrison double moment (Morrison) | (Morrison et al., 2009) |
| 10 | CAM double-moment 5-class (CAM) | (Eaton, 2011) |
| 11 | Stony-Brook University (SBU) | (Lin and Colle, 2011) |
| 12 | WRF double-moment 6-class (WDM6) | (Lim and Hong, 2010) |
| 13 | NSSL double moment (NSSL2) | (Mansell et al., 2010) |
| 14 | NSSL single-moment 7-class (NSSL1) | |
| 15 | Aerosol-Aware Thompson (ThompsonAA) | (Thompson and Eidhammer, 2014) |
| 16 | P3 | (Morrison and Milbrandt, 2015) |



**Table 3: Radiation scheme options investigated in this study.**

|   | Short-wave radiation | | Long-wave radiation | |
|---|---|---|---|---|
|   | Scheme | Reference | Scheme | Reference |
| 1 | Dudhia | (Dudhia, 1989) | RRTM | (Mlawer et al., 1997) |
| 2 | CAM | (Collins et al., 2004) | CAM | (Collins et al., 2004) |
| 3 | RRTMG | (Iacono et al., 2008) | RRTMG | (Iacono et al., 2008) |
| 4 | Goddard | (Matsui et al., 2020) | Goddard | (Matsui et al., 2020) |






**Table 4: Wind speed statistics averaged over the weather stations used for different PBL schemes.**

|  | CORR | BIAS | RMSE |
| --- | --- | --- | --- |
| LES | 0.916 | 0.573 | 1.789 |
| YSU | 0.930 | **0.448** | **1.585** |
| MYJ | **0.956** | 0.928 | 2.010 |
| *QNSE* | *0.643* | *3.218* | *4.839* |
| MYNN | 0.941 | 0.553 | 1.732 |
| ACM2 | 0.939 | 0.691 | 1.872 |
| BouLac | 0.917 | 0.831 | 2.036 |
| UW | 0.938 | 0.649 | 1.746 |
| TEMF | 0.894 | 1.015 | 1.884 |
| Shin-Hong | 0.930 | 0.705 | 1.738 |





**Table 5: Wind direction statistics for different PBL schemes.**

|  | CORR | BIAS | RMSE |
|---|---|---|---|
| LES | 0.420 | 5.171 | **36.646** |
| YSU | 0.339 | 10.436 | 42.438 |
| MYJ | 0.280 | 8.218 | 38.406 |
| QNSE | 0.480 | **1.670** | 39.792 |
| MYNN | 0.234 | 8.110 | 40.013 |
| ACM2 | 0.381 | 9.703 | 41.830 |
| BouLac | **0.488** | 13.153 | 43.495 |
| UW | 0.378 | 10.830 | 41.110 |
| TEMF | 0.317 | 7.267 | 39.900 |
| Shin-Hong | 0.284 | 11.661 | 42.365 |



**Table 6: Statistics for wind speed and direction for different MP schemes.**

|  | Wind speed | | | Wind direction | | |
|---|---|---|---|---|---|---|
|  | CORR | BIAS | RMSE | CORR | BIAS | RMSE |
| Lin | 0.941 | 0.692 | 1.800 | 0.374 | 8.926 | 32.329 |
| WSM3 | 0.939 | 0.771 | 1.824 | 0.340 | 10.829 | 34.007 |
| WSM5 | 0.940 | 0.702 | 1.804 | 0.373 | 9.316 | 32.489 |
| ETA | 0.943 | 0.677 | 1.791 | 0.377 | 8.615 | 32.129 |
| WSM6 | 0.940 | 0.703 | 1.805 | 0.366 | 9.295 | 32.616 |
| Goddard | 0.943 | 0.685 | 1.800 | **0.395** | 8.905 | 31.762 |
| Thompson | 0.942 | 0.695 | 1.792 | 0.372 | 8.653 | 32.239 |
| MYDM7 | **0.944** | **0.660** | **1.788** | 0.392 | **8.050** | **31.505** |
| Morrison | 0.940 | 0.707 | 1.800 | 0.357 | 9.234 | 32.904 |
| CAM5 | 0.940 | 0.742 | 1.805 | 0.344 | 10.366 | 33.756 |
| SBU | 0.938 | 0.762 | 1.821 | 0.343 | 10.820 | 34.066 |
| WDM6 | 0.940 | 0.736 | 1.801 | 0.363 | 9.716 | 33.104 |
| NSSL2 | 0.940 | 0.736 | 1.801 | 0.363 | 9.716 | 33.104 |
| NSSL1 | 0.943 | 0.682 | 1.794 | 0.377 | 8.491 | 32.119 |
| ThompsonAA | 0.940 | 0.736 | 1.806 | 0.351 | 10.290 | 33.612 |
| P3 | 0.942 | 0.670 | 1.789 | 0.380 | 8.151 | 31.737 |






**Table 7: Statistics for wind speed and direction for different LW/SW radiation schemes.**

|  | Wind speed | | | Wind direction | | |
|---|---|---|---|---|---|---|
|  | CORR | BIAS | RMSE | CORR | BIAS | RMSE |
| Dudhia and RRTM | **0.943** | **0.646** | **1.754** | 0.355 | 9.155 | 32.765 |
| CAM | 0.941 | 0.736 | 1.834 | **0.377** | 9.276 | **32.519** |
| RRTMG | 0.939 | 0.763 | 1.842 | 0.370 | 9.983 | 33.054 |
| Goddard | 0.940 | 0.697 | 1.783 | 0.362 | **9.040** | 32.598 |





**Table 8: Wind speed statistics for the top 10 schemes. The ensembles show the mean values for different numbers of**
**the best simulations, shown in parentheses, alongside the mean values of all simulations (without QNSE).**

|  | CORR | BIAS | RMSE |
|---|---|---|---|
| WDM6 | 0.934 | 0.361 | 0.530 |
| Goddard | 0.937 | 0.364 | 0.527 |
| MYDM7 | 0.928 | 0.331 | 0.525 |
| NSSL1 | 0.934 | 0.355 | 0.524 |
| WSM5 | 0.930 | 0.370 | 0.545 |
| Lin | 0.931 | 0.370 | 0.544 |
| P3 | 0.927 | 0.351 | 0.539 |
| SBU | 0.930 | 0.453 | 0.605 |
| ETA | 0.931 | 0.338 | 0.526 |
| WSM6 | 0.931 | 0.370 | 0.544 |
| *Ensemble (2)* | *0.936* | *0.362* | *0.528* |
| *Ensemble (3)* | *0.934* | *0.352* | *0.525* |
| ***Ensemble (4)*** | ***0.934*** | ***0.352*** | ***0.524*** |
| *Ensemble (5)* | *0.934* | *0.355* | *0.528* |
| *Ensemble (7)* | *0.933* | *0.357* | *0.531* |
| *Ensemble (10)* | *0.933* | *0.366* | *0.537* |
| ***Ensemble (all)*** | ***0.937*** | ***0.686*** | ***0.808*** |

*Notes: The PBL and LW/SW radiation schemes used in the 10 best schemes were YSU and Dudhia and RRTM.*



**Table 9: Wind speed statistics for coastal and inland stations.**

| | | Costal stations | | | Inland stations | | |
|---|---|---|---|---|---|---|---|
| | | CORR | BIAS | RMSE | CORR | BIAS | RMSE |
| PBL | LES | 0.790 | **0.352** | 0.844 | 0.912 | 0.607 | 0.818 |
| | YSU | 0.841 | 0.426 | **0.732** | 0.918 | **0.456** | **0.638** |
| | MYJ | 0.838 | 0.846 | 1.059 | **0.951** | 0.950 | 1.092 |
| | MYNN | **0.856** | 0.553 | 0.799 | 0.932 | 0.554 | 0.725 |
| | ACM2 | 0.848 | 0.629 | 0.913 | 0.931 | 0.701 | 0.882 |
| | BouLac | 0.834 | 0.533 | 0.880 | 0.912 | 0.882 | 1.130 |
| | UW | 0.835 | 0.629 | 0.893 | 0.933 | 0.651 | 0.788 |
| | TEMF | 0.846 | 0.925 | 1.098 | 0.876 | 1.031 | 1.181 |
| | Shin-Hong | 0.850 | 0.687 | 0.902 | 0.919 | 0.709 | 0.849 |
| MP | Lin | 0.841 | 0.573 | 0.849 | 0.934 | 0.718 | 0.857 |
| | WSM3 | **0.871** | 0.784 | 0.955 | 0.931 | 0.774 | 0.910 |
| | WSM5 | 0.839 | 0.602 | 0.870 | 0.933 | 0.725 | 0.865 |
| | ETA | 0.850 | 0.537 | 0.815 | 0.935 | 0.707 | 0.845 |
| | WSM6 | 0.839 | 0.602 | 0.870 | 0.933 | 0.726 | 0.865 |
| | Goddrd | 0.843 | 0.548 | 0.835 | **0.936** | 0.715 | 0.853 |
| | Thompson | 0.852 | 0.580 | 0.835 | 0.934 | 0.719 | 0.855 |
| | MYDM7 | 0.846 | **0.514** | **0.805** | **0.936** | **0.691** | **0.830** |
| | Morrison | 0.848 | 0.615 | 0.864 | 0.932 | 0.728 | 0.866 |
| | CAM5 | 0.864 | 0.745 | 0.932 | 0.931 | 0.746 | 0.882 |
| | SBU | 0.866 | 0.750 | 0.937 | 0.929 | 0.768 | 0.906 |
| | WDM6 | 0.861 | 0.701 | 0.903 | 0.932 | 0.746 | 0.880 |
| | NSSL2 | 0.861 | 0.701 | 0.903 | 0.932 | 0.746 | 0.880 |
| | NSSL1 | 0.848 | 0.548 | 0.825 | 0.935 | 0.711 | 0.848 |
| | ThompsonAA | 0.867 | 0.724 | 0.912 | 0.931 | 0.742 | 0.880 |
| | P3 | 0.843 | 0.532 | 0.821 | 0.935 | 0.700 | 0.839 |
| RAD | Dudhia and RRTM | 0.844 | 0.609 | 0.855 | **0.937** | **0.652** | **0.787** |
| | CAM | 0.852 | 0.610 | 0.865 | 0.933 | 0.758 | 0.900 |
| | RRTMG | **0.857** | 0.712 | 0.924 | 0.931 | 0.772 | 0.915 |
| | Goddard | 0.853 | **0.550** | **0.811** | 0.931 | 0.724 | 0.859 |






**Table 10: Wind speed statistics for stations located at different altitudes**

| | | <50 m | | | >50 m and <250 m | | | >250 m | | |
| --- | --- | --- | --- | --- | --- | --- | --- | --- | --- | --- |
| | | CORR | BIAS | RMSE | CORR | BIAS | RMSE | CORR | BIAS | RMSE |
| PBL | LES | 0.878 | 0.307 | 0.693 | 0.902 | 0.708 | 0.906 | 0.902 | 1.034 | 1.335 |
| | YSU | 0.910 | **0.248** | **0.559** | 0.868 | **0.516** | **0.758** | 0.868 | **0.862** | **1.078** |
| | MYJ | **0.923** | 0.599 | 0.766 | **0.939** | 1.052 | 1.188 | **0.939** | 1.584 | 1.982 |
| | MYNN | 0.913 | 0.374 | 0.618 | 0.904 | 0.590 | 0.801 | 0.904 | 0.971 | 1.319 |
| | ACM2 | 0.916 | 0.470 | 0.694 | 0.912 | 0.755 | 0.940 | 0.912 | 1.171 | 1.576 |
| | BouLac | 0.890 | 0.566 | 0.865 | 0.905 | 0.976 | 1.194 | 0.905 | 1.269 | 1.728 |
| | UW | 0.919 | 0.453 | 0.657 | 0.901 | 0.683 | 0.847 | 0.901 | 1.115 | 1.436 |
| | TEMF | 0.895 | 0.823 | 0.982 | 0.845 | 1.074 | 1.247 | 0.845 | 1.428 | 1.733 |
| | Shin-Hong | 0.910 | 0.501 | 0.707 | 0.885 | 0.758 | 0.926 | 0.885 | 1.159 | 1.453 |
| MP | Lin | 0.916 | 0.454 | 0.668 | 0.916 | 0.788 | 0.931 | 0.916 | 1.179 | 1.493 |
| | WSM3 | 0.922 | 0.587 | 0.750 | 0.915 | 0.823 | 0.966 | 0.915 | 1.195 | 1.504 |
| | WSM5 | 0.913 | 0.471 | 0.685 | 0.916 | 0.792 | 0.935 | 0.916 | 1.179 | 1.491 |
| | ETA | 0.920 | 0.432 | 0.643 | 0.915 | 0.779 | 0.922 | 0.915 | 1.174 | 1.484 |
| | WSM6 | 0.914 | 0.472 | 0.684 | 0.915 | 0.794 | 0.937 | 0.915 | 1.177 | 1.489 |
| | Goddrd | 0.917 | 0.442 | 0.659 | **0.918** | 0.784 | 0.925 | **0.918** | 1.181 | 1.495 |
| | Thompson | 0.921 | 0.461 | 0.660 | 0.914 | 0.787 | 0.930 | 0.914 | 1.174 | 1.485 |
| | MYD7 | 0.919 | 0.407 | **0.630** | 0.917 | **0.766** | **0.909** | 0.917 | 1.168 | 1.480 |
| | Morrison | 0.918 | 0.483 | 0.681 | 0.912 | 0.792 | 0.938 | 0.912 | 1.173 | 1.484 |
| | CAM5 | 0.923 | 0.550 | 0.719 | 0.913 | 0.797 | 0.941 | 0.913 | 1.180 | 1.489 |
| | SBU | 0.922 | 0.574 | 0.740 | 0.911 | 0.815 | 0.962 | 0.911 | 1.190 | 1.502 |
| | WDM6 | 0.923 | 0.530 | 0.705 | 0.913 | 0.803 | 0.945 | 0.913 | 1.187 | 1.488 |
| | NSSL2 | 0.923 | 0.530 | 0.705 | 0.913 | 0.803 | 0.945 | 0.913 | 1.187 | 1.488 |
| | NSSL1 | 0.920 | 0.442 | 0.651 | 0.915 | 0.782 | 0.927 | 0.915 | 1.168 | 1.479 |
| | ThompsonAA | **0.925** | 0.535 | 0.705 | 0.911 | 0.799 | 0.946 | 0.911 | 1.179 | 1.488 |
| | P3 | 0.916 | **0.428** | 0.650 | 0.917 | 0.771 | 0.913 | 0.917 | **1.163** | **1.476** |
| RAD | Dudhia and RRTM | 0.917 | **0.436** | **0.648** | **0.921** | **0.707** | **0.847** | **0.921** | **1.099** | **1.416** |
| | CAM | 0.919 | 0.497 | 0.692 | 0.915 | 0.827 | 0.973 | 0.915 | 1.210 | 1.526 |
| | RRTMG | **0.921** | 0.551 | 0.729 | 0.911 | 0.831 | 0.983 | 0.911 | 1.206 | 1.521 |
| | Goddard | 0.919 | 0.445 | 0.654 | 0.911 | 0.796 | 0.937 | 0.911 | 1.193 | 1.492 |


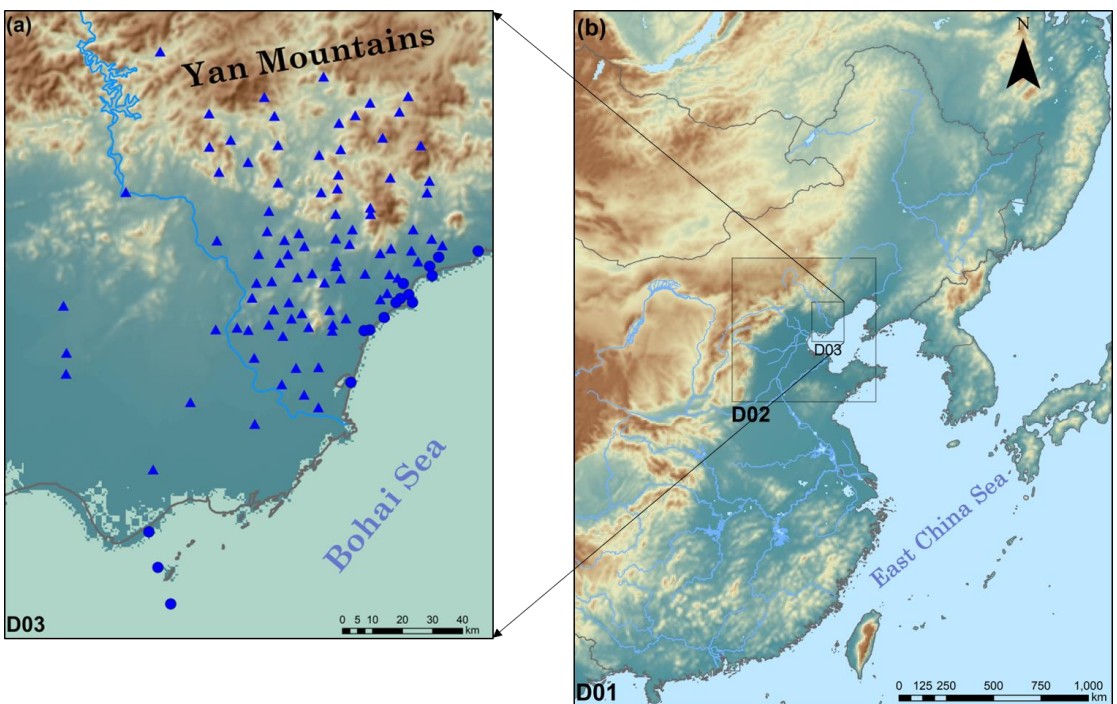

**Figure 1: Map showing the study area (a) and WRF nested domains (D01–D03) (b). Solid blue circles and triangles in (a) represent coastal weather stations (16 in total) and inland weather stations (89 in total), respectively.**






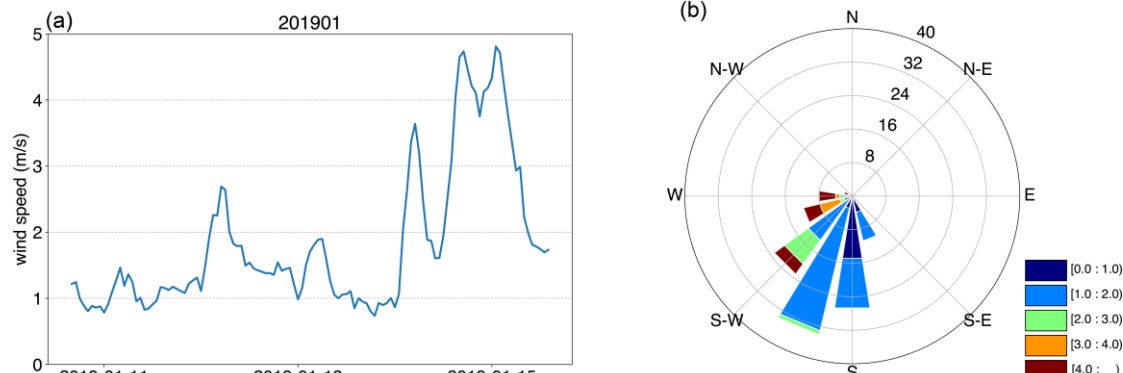

**Figure 2: Time series of hourly wind speed (a) and the wind rose (b) for January 11th to January 15th, 2019, averaged over the 106 weather stations shown in Figure 1a.**





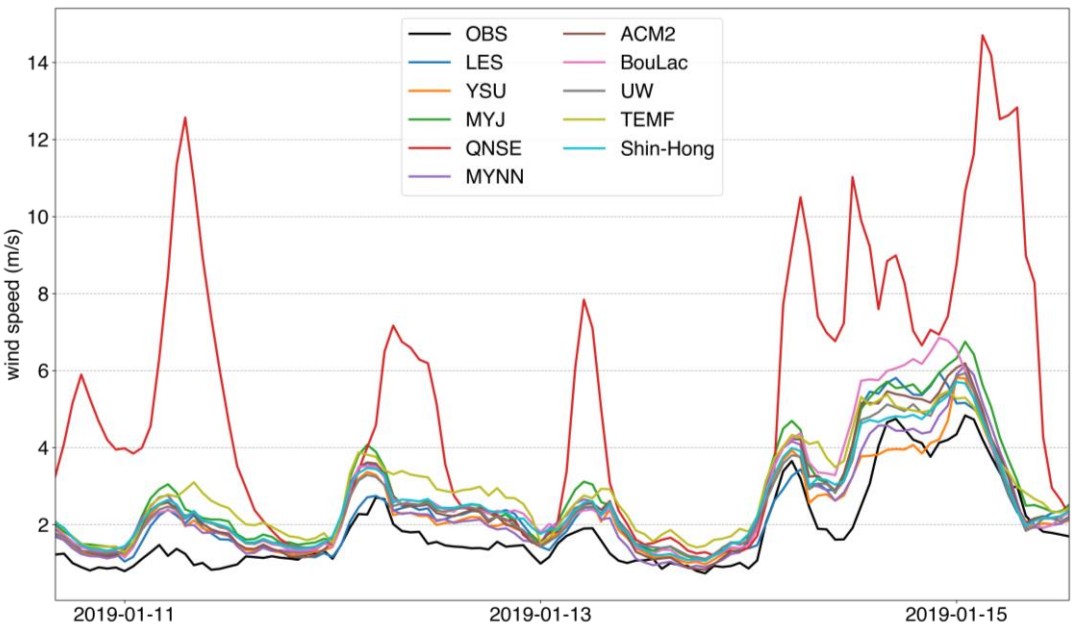

**Figure 3: Time series of observed and simulated wind speeds (m/s) averaged over 106 weather stations. The result for each PBL scheme is the average of all the simulations using that scheme.**





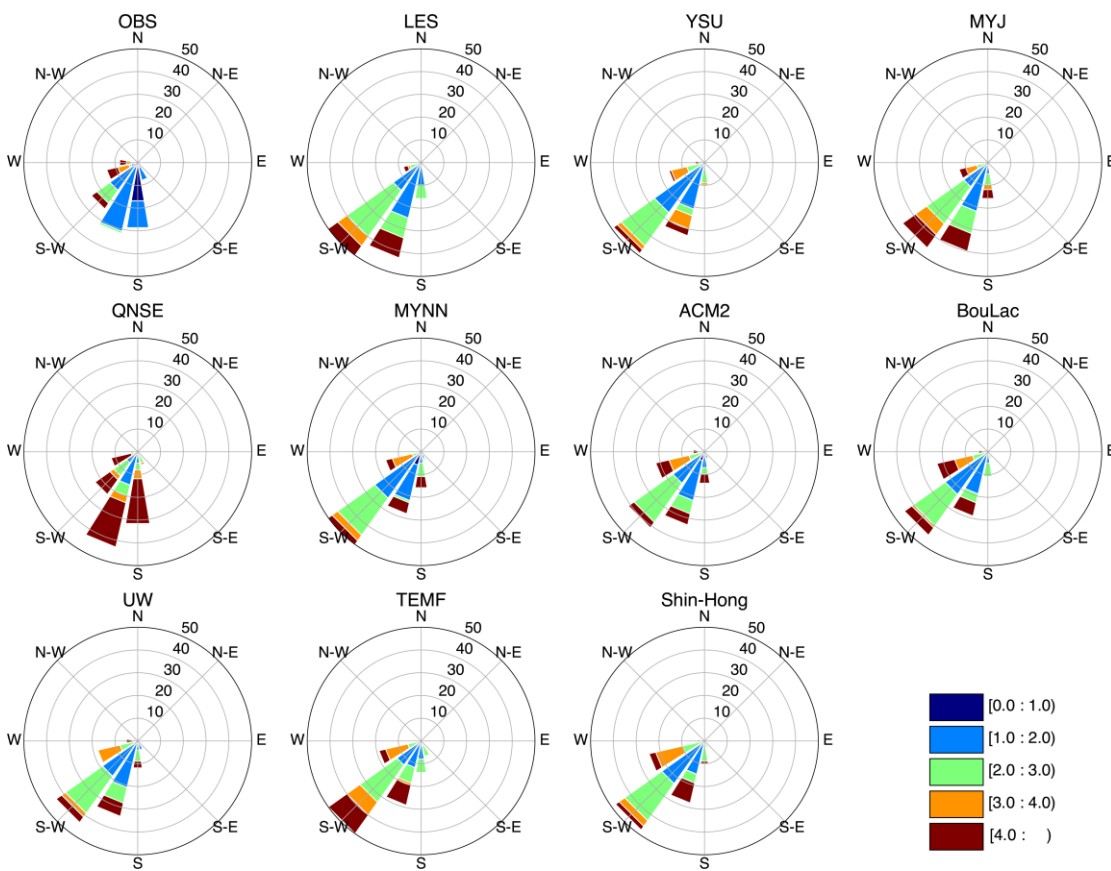

**Figure 4: Wind roses showing observed and simulated differences between PBL schemes.**






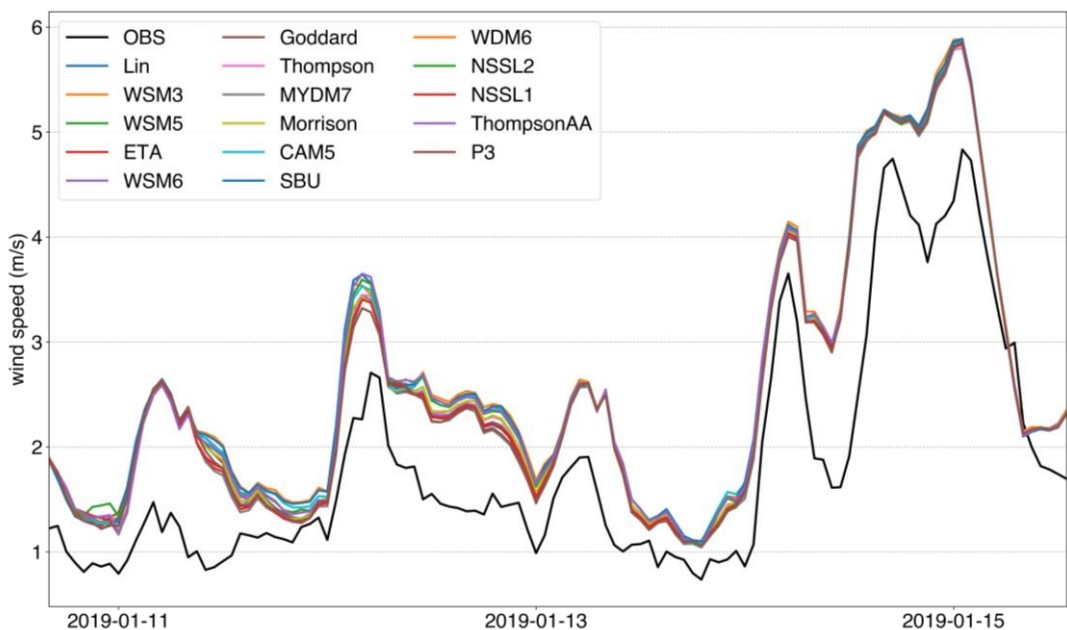

Figure 5: Time series for observed and simulated wind speeds (m/s) for different MP schemes.





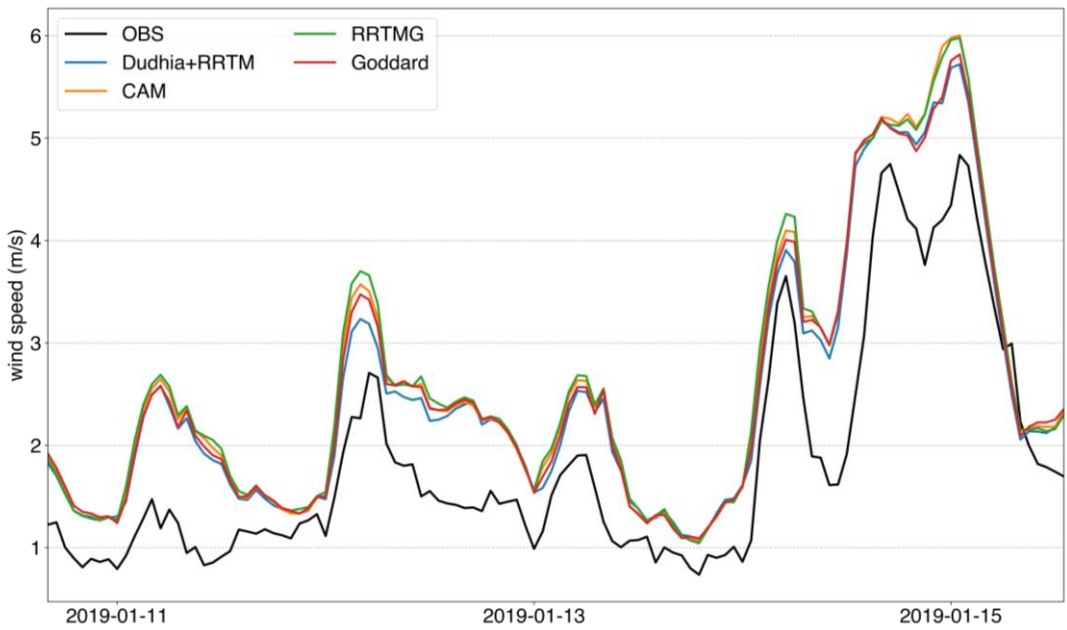

**Figure 6: Time series for observed and simulated wind speeds (m/s) for different LW/SW radiation schemes.**





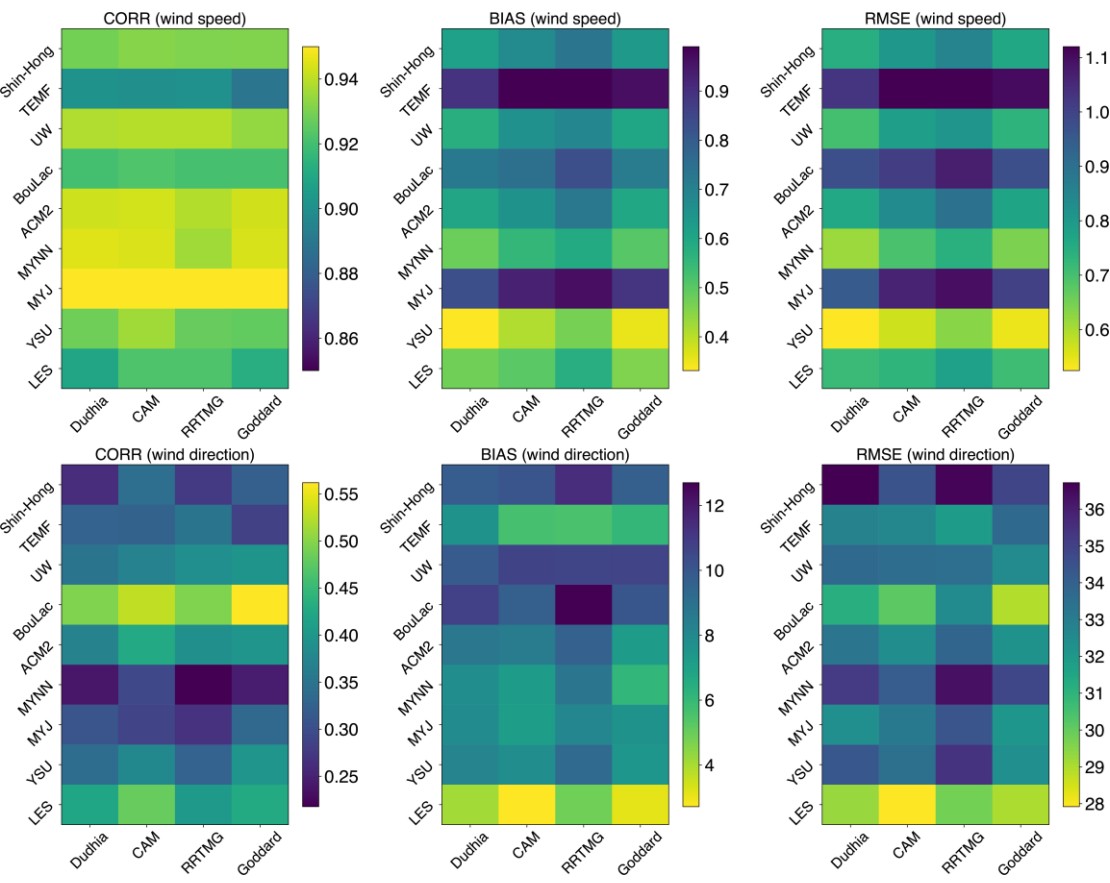

**Figure 7: Statistics for wind speed (upper panels) and direction (lower panels) for different combinations of PBL and LW/SW radiation schemes. The MP scheme used was MYDM7.**


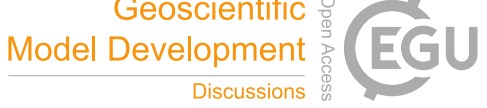



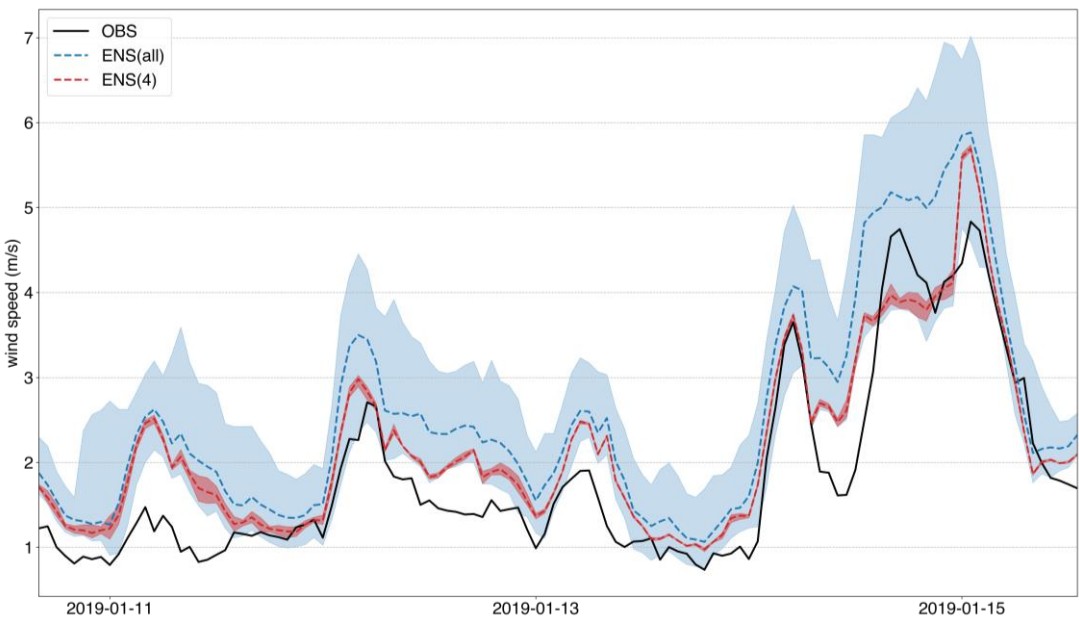

**Figure 8: Time series of wind speed (m/s) observations, including an ensemble of the top four schemes and an ensemble of all the simulations. Shading shows the spread of the corresponding simulations.**



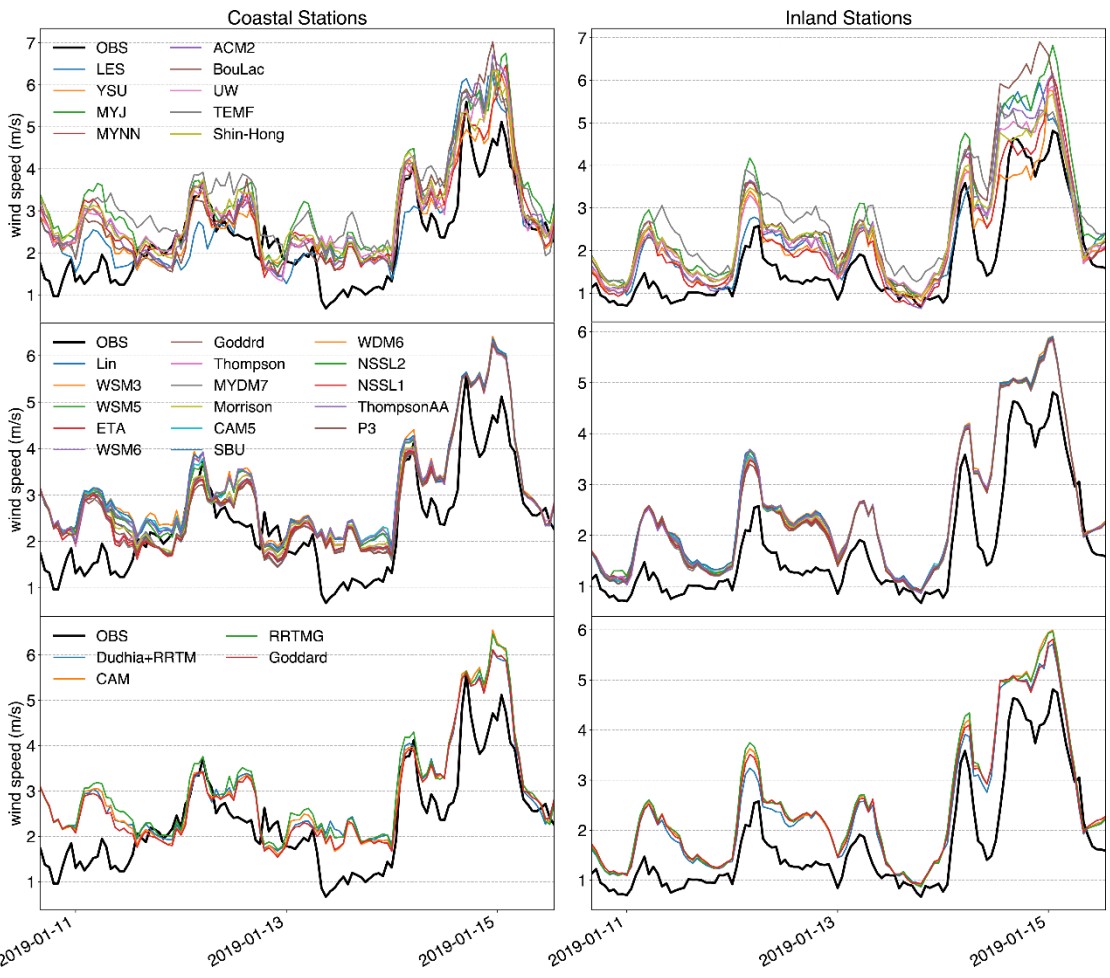

**Figure 9: Comparison of simulated wind speeds between the coastal (left) and inland (right) stations shown in Figure 1a.**





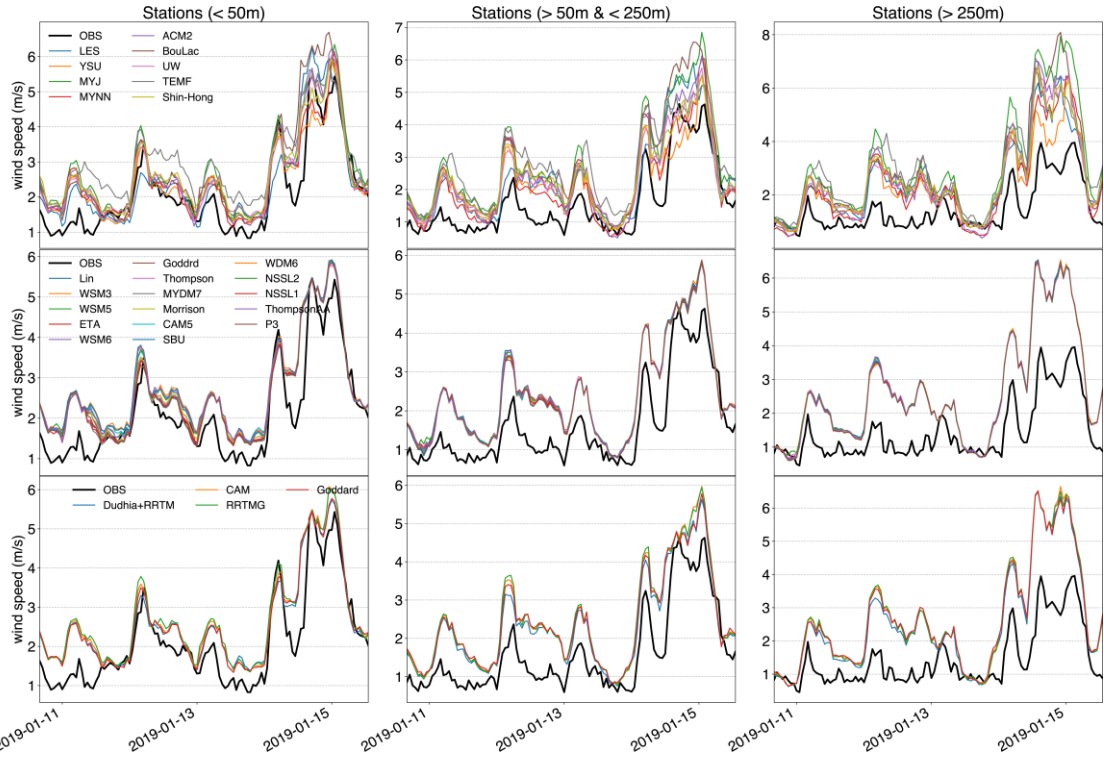

**Figure 10: Comparison of simulated wind speeds (m/s) for stations located at different altitudes.**