# Peer review of "Impact of physical parameterizations on wind simulation with WRF V3.9.1.1 under stable conditions at PBL gray-zone resolution: a case study over the coastal regions of North China"

_Geoscientific Model Development, 2022_

## Referee Comment (RC1)

Title: Impact of physical parameterizations on wind simulation with WRF V3.9.1.1 over the coastal regions of North China at PBL gray-zone resolution
Author(s): Entao Yu et al.
MS No.: gmd-2022-53
MS type: Model evaluation paper

**General Comments**

This paper presents a sensitivity case study of WRF wind forecasts under calm and stable conditions with a systematic variation of planetary boundary layer (PBL), microphysics (MP), and radiation parameterizations. For a case study this work is result of an extensive computational effort. However, there are some aspects in the methodology that need clarification and better motivation. Considering the impressive volume of the generated model data, the presented evaluation is limited, and I suggest expanding on the analysis to improve the scientific quality of the paper. Several findings require more insightful interpretations and discussions; some of the presented conclusions require clarification or correction. The presentation of the data and results can be improved upon.

**Specific Comments**

1) This study aims to assess the ability of various physics parameterization configurations to predict **calm and stable weather conditions that favor air pollution**. However, the authors verify merely wind speed (and wind direction to a lesser degree). In order to gain insight what drives these differences in wind speed and to evaluate atmospheric properties that are crucial for air quality (such as static stability), it would be valuable to further assess vertical profiles.

2) This paper presents a **case study** for a specific location and event, and the manuscript should be framed accordingly (in the title, results, and discussion). However, the authors draw general conclusions from their findings and presume transferability of their results across the world and across variables (e.g., lines 81-83 and lines 337/338) and generalize shortcoming in the WRF model (lines 305/306). The current manuscript lacks an objective error discussion on the various sources of uncertainty and limitations of the study: The presented findings could be unique to the meteorological setup of the event, the location, the input dataset, the domain setup, other unchanged parameterization types or model settings, etc.

3) It is not clear how the authors arrive at the conclusion that "**wind direction** is insensitive to changes in physics parameterizations" (line 207 & 213-15). The manuscript only shows variations between PBL-scheme groups and differences are visible in figure 4 and table 5. The authors have not performed any hypothesis testing to show whether these differences are actually insignificant compared to the variation in wind speeds. The authors write that "the

model RMSE and BIAS values for different PBL schemes are similar", yet table 5 reveals that the worst PBL scheme (namely, BouLac) has a wind-direction Bias that is 154% larger than the Bias of the best PBL scheme (namely, LES – excluding QNSE), see table 5. In comparison the wind-speed Bias of the worst PBL scheme (namely, TEMF) is 126% larger than the Bias of the best PBL scheme (namely YSU), see table 4.

4) A considerable amount of computational resources was spent to test **16 MP schemes** – this is the largest number of schemes within any of the parameterisation categories tested in this paper. Although the authors cite Cheng et al. (2013) to justify that MP can have an impact on wind fields, there is little justification to evaluate MP sensitivity to this extent under the dry and stable conditions of this case study. As opposed to the conditions described in this manuscript, Cheng et al. (2013) pointed out that MP can affect wind fields for convective weather phenomena during the summer (with weak large-scale forcing) that are associated with gust fronts / outflow boundaries from cold pools that result from strong downdrafts of thunderstorms. One would not expect to see significant impact from the choice of MP parameterization on wind forecasts under stable conditions, hence, these results are not surprising. Instead of assessing a vast amount of MP schemes without valid motivation, it could have been informative to expand on the number of radiative schemes and/or include an investigation on the impact of the land surface schemes.

5) **Tables and Figures**:
   - Tables 4-10 show verification data values that are not always easily to grasp from plain numbers in this quantity. The authors should consider visualizing this data for better presentation to the reader. For example, this could be in the form of bar plots, boxplots or heatmaps. I also encourage the authors to include the distribution across the 105 stations (e.g., the range across stations could be shown in box plots or with error bars on bar or line plots). The verification plots could be combined with the respective timeseries plot (e.g., figure 3 would have a subplot visualising the error metrics from table 4).
   - Figure 2: This figure shows identical data to other figures - i.e., 2a is repeated in figures 3, 5, 6, and 8, whereas 2b is identical to the top left panel in figure 4. This makes figure 2 redundant. An alternative could be to show the observational data of Tangshan city as a specific example in connection to the paragraph in line 96-101. When updating, please also include a more descriptive title to figure 2a.

6) Lines 247-249: For wind speed Bias and RMSE, within each radiation group the same PBL schemes rank best; and within each PBL group, the same radiation schemes rank best. This indicates that a systematic variation of parameterizations as presented in this paper, is *not* necessary. (E.g., for wind-speed RMSE no matter which PBL scheme, Dudhia or Goddard radiation are always best.) However, for CORR and wind direction, this pattern is not always consistent, which indicates that a systematic variation of parameterizations *is* useful when focusing on these variables. (E.g., for wind-direction RMSE, the best radiation scheme depends on the choice of PBL scheme - for TEMF PBL, RRTMG radiation is best; for BouLac PBL, Goddard radiation is best; for LES, CAM radiation is best.)

7) **Evaluation of model configurations with the best individual performance** (lines 254-269):
- Lines 266/267: This statement is misleading. Perhaps correct to "the ensemble of the top *four* configurations reduces model bias by approximately half *compared to the ensemble that uses all configurations,* while the CORR value *of the super-ensemble mean* was highest *among* all the *ensembles*." Note that the highest CORR seen in table 8 (0.937) is result of either the single-model configuration using Goddard MP, or the ensemble using all configurations. The lowest BIAS (0.331) is result of the single-model configuration using MYDM7. And the lowest RMSE (0.524) is result of either the single-model configuration using NSSL1, or the ensemble using the 4 best configurations. This disagrees with the concluding statement in line 339-341.
- For simulations that struggle with systematic overprediction, it is implied that an ensemble of a subgroup of members with smaller biases improves the ensemble-mean bias compared to using all members. However, (a) systematic errors can be significantly reduced with bias-correction, and (b) ensembles generate probabilistic forecasts and the authors present no discussion on the probabilistic features of their ensembles (e.g., ensembles with narrow spread are often under-dispersive / overconfident). More on this in point 8).
- Please clarify if the SD in the Taylor skill scores was calculated over the various stations and averaged over time, or if SD was calculated over the time series and averaged over stations. This is important to understand why the MP scheme with best CORR is not also the scheme with best Taylor skill score considering that all MP schemes have very similar temporal patterns (as seen figure 5). If the variation across stations is significant among MP schemes, it would be informative to analyse this spatial variation.
- Table 8 – If the 10 best WRF configurations are based on the ranks of Taylor skill scores, please add the ranks with corresponding Taylor skill score values to the table.

8) The authors present correlation coefficients, biases, and RMSE without any insightful discussion what the different verification metrics represent and why it is conceivable that some schemes perform best according to one metric and worst according to another. It is also not discussed that the performance of raw model forecasts can be significantly enhanced by **post-processing** - in particular the systematic-errors component (biases), which appear to be the main issue in this case study. As already mentioned in point 7), the author's suggestion of the ensemble generation by picking a small number of members with lowest bias without considering the effects of calibration is problematic.

9) Section 2.1: Please add a more comprehensive **synoptic analysis** of the event. Consider adding a weather analysis map and/or radio soundings to show the observed stable stratification (perhaps in figure 2). Considering the substantial testing of MP schemes, it would be useful to mention if there was any cloudiness. With regards to section 3.2.2 it would be interesting to analyse why observed wind speeds decrease with elevation.

10) Ensemble spread usually grows with **forecast lead time** as predictive skill declines. Please discuss how the authors explain the narrowing ensemble spread in figure 5.

**Technical Corrections**

The title needs to state that (a) this is a case study and (b) calm wind speeds under stable conditions are investigated.

There are inconsistencies with the tense, please make consistent everywhere.
E.g., line 25/26 "The MYNN scheme *showed* the highest correlation among all PBL schemes, while LES and YSU *has* the smallest model errors, the RRTMG and Goddard schemes *showed* the highest correlations …"

Line 14: Technically wind speeds are always zero at the surface. The wind simulations in this paper likely correspond to the 10-m above surface level.

Line 18: "The data show that *the* WRF model …"

Line 25: "For example, for the weather stations located in coastal regions [*no new sentence*] the MYNN …"

Line 35/36: "(Zhang  et al., 2014; Cai et al., 2017; Zhang et al., 2015)" – please sort references either alphabetically or according to publication time (check journal guidelines) throughout the manuscript

Line 36-39: Please provide a brief explanation on how increasing global temperatures relate to haze and low wind speeds.

Line 39/40: Before saying that it is crucial to improve wind predictions because haze events are hazardous and may become more frequent in future, inform the reader about the problems with WRF wind forecasts - i.e., include a short literature review on known biases and challenges in simulating wind fields in your region of interest.

Line 49: "choosing appropriate combinations is  important"

Line 67: Start a new paragraph to separate the sections on MP vs radiation.

Line 72/73: Add "to this extent", "in China" and "to our knowledge". WRF wind performance has been evaluated in a systematic manner in other studies, e.g., Fernández-González et al. 2018, Santos-Alamillos et al. 2013, Siuta et al. 2017.

Line 74/75: "error compensation among processes […] may predict incorrect wind patterns" If errors compensate each other, that would imply that the result would be more accurate; however, in the chaotic system of the atmosphere errors are usually amplified and grow from various imperfections in the model.

Lines 77 & 81: "*the* WRF model" (also check everywhere else in the manuscript)

Lines 82: "founding"?

Line 86: The caption repeats all sub-captions. I suggest re-naming it to "Data and Methods" or similar.

Line 87: "stable weather events in 2019" implies a series of events – this paper only discusses a single event!

Line 92: "favorable weather conditions" – be more specific please

Line 94: "synoptic *forcing*"?; "vertical mixing *in* the atmosphere"; "increasing the stability of surface air" – air in the boundary layer?

Line 97: Could the location of Tangshan city (or the station measuring these values) please be included to figure 1? Thank you!

Line 98: Are any statistics available on the socioeconomic or health impacts of this event that could be included to exemplify the severity of the event (e.g., increased hospitalization rates)?

Line 104: Plural "simulations"

Line 109: Better "within the PBL" – specifically for this event, how many levels were in the PBL on average and at the minimum?

Line 112: Why was such a long spin-up time selected? It is important to note that the model dataset was generated from a single initialization with a 7-day forecast horizon and that forecast skill is expected to degrade with lead time.

Line 112: What are the default parameterizations? Please name and reference them.

Line 113/114: Please clarify: Were the simulations first run for D01 and D02, then D03 was initialized with the output from D02, or was each of the 640 simulations run with one-way-nested feedback across all three domains at each time step?

Line 117/118: "The lateral boundary *conditions* and sea surface temperature were updated …" – also, which dataset did the SST come from?

Line 118: wind was *calculated,* or output retrieved from the model?

Line 121 & 124: Add comma "e.g., …" & "i.e.," – please check throughout the manuscript

Line 126: The gray zone is first mentioned at Line 77 and should be defined there. After definition, the quotation marks for this term can be removed.

Line 134: Both "atmospheric boundary layer" or "planetary boundary layer" are fine, but please be consistent throughout the manuscript.

Lines 139-145:
This paragraph should be revised for a more insightful summary.
1) The concept of single- vs double-moment and hydrometeor classes should be briefly explained.
2) Explain why some MP schemes are "suitable for high-resolution simulations".
3) The Goddard has a reference from 1989 and is described as one for the "new schemes".
4) NSSL1 has no reference in Table 2.

Line 146: Table 3 is not referenced anywhere in the manuscript.
Tables 1, 2, and 3 could be combined.

Line 155: How did you define / identify "spurious jumps"?
105 stations remained – out of how many?

Line 167: Also define "i"

Line 170/171: Where in the paper is this metric considered?

Line 174: This equation misses sums.

Lines 177-180: Are there 105 or 106 stations?
This paragraph can be skipped.

Line 183: Wind speed data is not directly *produced by* PBL schemes. Wind speed is a dynamic variable that is adjusted by the PBL scheme. So perhaps the data "is produced *using* PBL schemes".

Figure 3 and all other figures with time series: Clarify in your manuscript whether these are UTC or local times. If times are in UTC, please mention to which local times these translate. Please also include tick marks for each date and possibly a vertical line separating each day as a reference for diurnal periods.

Line 184: I am seeing that "The WRF model *exaggerates* the temporal variation of observed wind speed in the study area"

Line 185: I disagree with the statement that "QNSE showed no obvious daily wind speed change during the simulation period" – The wind speed change is considerably larger than with all other schemes.

Lines 186-189: This section needs revision. It is unclear which correlation the authors refer to; it needs to be clarified that the 10m/s bias applies to QNSE only; it should be elaborated what other studies found ("such models" – referring to the QNSE models or the general overprediction by all WRF models?) Note that a more thorough literature review is needed in the introduction which could be referred to here.

Line 194: "*on* January 15th"
"partly due to faster observed wind speeds" – right, the bias looks multiplicative, but probably also due to the general error growth in NWP with lead time

Line 195/196: The description of what bold and italicized numbers mean belongs in the figure caption.

Line 200, 224, 236: The authors often describe the next-best schemes as having "similar statistics". Please be specific to avoid confusion. For example, "X1 is the best scheme, followed by X2 and X3" or "X1 shows the best verification score. X2 and X3 are slightly worse according to this verifications score."

Line 200/201: A correlation coefficient of 0.643 would usually not be considered "extremely low". Please revise your wording.

Line 202/203: Please justify this assumption. What does other literature suggest?

Lines 204/205: Note that QNSE is still included in figure 4 and table 5. Either exclude QNSE from there, or move the statement that QNSE will be omitted after referencing figure 4 and table 5.

Line 206: Plural "wind roses"
Wind roses in figures 2 and 4 need a legend description.

Line 210: "As the simulated wind direction was calculated using the wind speed *components*, the bias in modeled wind direction can be attributed to bias in the wind-speed *component* simulation." – this statement is somewhat redundant.

Line 224: "P*3*" (not "P2")

Line 230: "*ensemble* spread"

Line 233: " reduced this overestimation, and thus produced values that were  closer to weather station observations." – these differences are relatively small but consistent

Line 238: "while simulations that employed the *Goddard* [not RRTMG] scheme showed the lowest BIAS values."

Line 240: The authors use the term "physical components" interchangeably with "physics parameterizations". "Physical components" can be ambiguous, I recommend consistently referring to "parameterizations" throughout the manuscript.

Line 243: "A total of 40 simulations" – it is actually only 36 because QNSE is missing.

Line 244: "which produced outcomes that were  consistent with other results" –which results? Other MP groups; previously shown results in this manuscript; or different studies? Please be specific!

Line 247: "*further* investigation"

Lines 247/248: "Dudhia and RRTM schemes , or the Goddard *schemes* " – Goddard also has two schemes, LW and SW

Lines 256/257 and Table 8: "The PBL and LW/SW radiation schemes used in the 10 best schemes were YSU and Dudhia and RRTM" – This wording sounds like the authors decided upon using these PBL and radiation schemes, rather than this being a results of their analysis. If I understand correctly, I suggest clarifying "The best 10 WRF model configurations have in common that they use the same PBL and radiation schemes, namely YSU and Dudhia-RRTM. Due to the slight differences between models using different MP schemes, the 10 best performing WRF configuration only vary in MP option." or similar

Line 263: "plays an important role in  determining its performance"

Table 8: Add a separating horizontal line between individual model configurations and ensembles.

Line 254: "Evaluation of model configurations with the best *individual* performance"

Line 276: Please explain how the classification between coastal and inland stations was conducted! Did you use an objective distance to the shoreline?

Line 277: add that this is based on the ensemble spread

Line 278: "consistent with the results of previous analyses *in this study*"

Line 279: "generally by the same magnitude" – no, figure 9 and table 9 both show that the bias is larger for inland stations

Line 280/281: "the *ensemble* spread was relatively larger for coastal stations, especially *among MP schemes and* for the first three days of the simulation period that exhibited low wind speeds"

Line 282: "source of model uncertainty" – although the authors observe different sensitivities for coastal vs inland stations between parameterizations, is that a source of model uncertainty?

"generates greater model differences" – compared to what?

Line 285: Is this the temporal CORR averaged over stations? Otherwise the sample-size difference between the two groups (16 coastal stations vs 89 inland stations) needs to be considered.

Line 285/286: What is meant by "as there are no clear differences in wind speed values and variations"?

Line 289: "previous investigations" – in this study or in different studies?

Line 294: "Dudhia and RRTM ... showed *worst temporal agreement* with observational data *for coastal stations*"

Line 298: Which subfigure was this info taken from? Perhaps the authors mean "the peak *observed* wind speed at high-elevation stations (>250 m) was *1.5* m/s slower than that for low-elevation stations (<50 m)." ?

Line 299: "the simulated peak values were generally *similar*"

Lines 302/303: "Interestingly, model performances *of different parameterization types* were generally similar for stations with different elevations, ..."

Line 304: "... smallest BIAS and RMSE values *at all elevation categories*" ?

Lines 305/306: It is plausible that physics-configuration performance depends on surface topography in other cases and locations with different topography. It is not appropriate to generalize these findings to overarching WRF performance like this. The authors provide no foundation to suggest that the limited configuration dependency seen in figure 10 is result of the terrain-following coordinates.

Line 314: "underlying land type and topography" – better "coast proximity and elevation" – land types include factors that were not investigated (e.g., soil texture, vegetation, roughness, canopy, etc.); topography includes aspects that were not investigated (e.g., slope steepness and directional angles of slopes)

Lines 315/316: "the WRF model reproduced the temporal variation of wind speed and direction over the study area *well*" – I don't agree that "to a high degree of accuracy" is an accurate description considering the biases presented.

Line 322/323: "The combined Dudhia and RRTM *radiation* schemes, and the  MYDM7 *MP* scheme both show the best *wind-speed* performances..."

Line 325: "." (redundant)

Line 327/328: "... substantially reduced overestimation of wind speed *compared to the ensemble of all 640 configurations* "

Line 329: Most (85%) of the 105 stations are inland stations, so it is implied that the overall pattern matches the inland stations most. These conclusions are not "applicable to the inland station" but they were mostly derived from them.

Lines 332/333: The best-configuration ranking might be similar, but the model results are different.

Line 335: These are not "all possible combinations that are available". (1) There are other parameterization types that were not investigated (e.g., cumulus convection, land surface, etc.); (2) within the parameterization types that were assessed there are more available (e.g., Kessler and WDM5 for MP, GBM and MRF for PBL, Fu–Liou–Gu and GFDL for radiation)

Line 337: "which has rarely been used in previous simulation studies *in China*"

Lines 339-343: This paragraph needs to be revised considering the major comments above.

---

## Referee Comment (RC2)

Review of "Impact of physical parameterizations on wind simulation with WRF V3.9.1.1 over the coastal regions of North China at PBL gray-zone resolution" by Yu et al.

This paper examines wind forecasts during a relatively long period of stable conditions when a haze event affected China. Surface meteorological observations are used to evaluate the WRF model's ability to predict the evolution of winds during the event. The authors conduct a number of WRF simulations (640 total), altering the PBL, radiation, and microphysics schemes to determine the sensitivity of wind speed and direction forecasts to choice of model physics. Pearson's correlation coefficient, bias, RMSE, and Taylor skill score are utilized to perform the model evaluation. Overall, the study shows the largest spread in wind speed within the PBL schemes tested, followed by radiation, and then microphysics schemes. Delineation between coastal/inland stations as well as stations at different elevation are examined to understand any model biases specific to land type and characteristics. An important finding is that WRF predicts wind speed less accurately for coastal stations compared to inland stations, and error metrics tended to degrade with increasing elevation.

Overall, this study has interesting components and would be a nice contribution to the literature especially due to the very large ensemble that was run. However, there are several aspects that should be addressed before the paper is suitable for publication. My main concerns are related to the authors' model setup, lack of some background information about the case, and insufficient physical explanations for some of their results. My general and specific comments are listed below.

Major/general comments:
1. WRF model configurations: I wonder why the authors chose to run all of the physics parameterizations as default except for YSU, which was run using a topographic correction for surface winds and the top-down mixing option. Were the impacts of these YSU options tested? I believe that YSU is not run this way by default, so it would be good to know the impact, especially since your results show that YSU is one of the best performers. For instance, MYNN has a number of namelist tuning options, so why not modify these? Also, what is the motivation for running with the top-down mixing option in YSU if this is a statically stable case? Are stratus/fog conditions expected in some of the coastal regions? Please explain.

As a separate but related issue, the authors use different surface layer schemes between the PBL schemes. This means that it is impossible to attribute all of the differences in results specifically to the PBL scheme. There is no discussion about this at all in the paper, although it is definitely important considering the station observations likely fall within the first model grid cell. Furthermore, why not use the revised MM5 scheme for all of the PBL schemes? I believe that it is compatible with all of them (I may be wrong here). Regardless, it would be good to run an additional simulation to determine the impact of the surface layer scheme (which I suspect is more important than the microphysics scheme under stable conditions).

2. Case study: The authors select a 4-day study period when stability conditions were stable to

evaluate the WRF model; however, there is only a few sentence discussion about the case in Section 2.1. Although the authors do conduct many simulations, this is still a case study, and unfortunately, the authors do not present any large-scale meteorological information. It would be good to know the synoptic pattern and what type of evolution occurred; I imagine there is a pattern change over the course of the event since the regional wind speeds went from ~1 m/s to ~5 m/s according to Fig. 2. Moreover, the authors consider the impact of microphysical schemes in WRF even though this is a stable case. Are the authors anticipating cloud effects? Despite the relatively small impact of the microphysics options (e.g., Fig. 5), there are noticeable differences on 2019-01-12. I think the authors should include some metric of observed clouds (e.g., satellite images) since clearly the model is producing clouds.

3. Physical explanations: By and large, this paper reports on the model performance with respect to near-surface wind speed and direction. However, I think the authors do not provide any physical explanations for any of their results. Ultimately, this ends up limiting the applicability of the study to other stable events in different seasonal periods and geographical locations. It would be good to address questions such as: why is QNSE so different from the other PBL schemes? How does the YSU topographic correction affect the forecast? Additionally, linking the low-level wind results to the PBL vertical structure (e.g., wind, temperature/stability, and moisture), as well as x-y spatial variability in the wind field, would be useful. For the planview, the authors could zoom in on a region where they see the largest differences in PBL schemes and overlay their observations. I think these comparisons are especially important for the different PBL schemes because the authors highlight the gray zone, which is where the choice of turbulent mixing tends to dominate the solution.

Minor/specific comments:
1. Abstract, L14: Should be, "near-surface" rather than "surface".

2. Abstract, L24: Please be more specific by what you mean by "land type".

3. Abstract, L25: "For example, for the weather stations located in coastal regions." This is an awkward sentence; it would be good to combine with another sentence or rewrite.

4. L47: Should add, "that occur at the sub-grid scale" after you say "key physical processes". Also, some of the processes listed (i.e., planetary boundary layers and cloud microphysics) aren't really processes. Please be more specific or re-phrase the beginning of the sentence.

5. L51: The impact of PBL schemes on wind simulations has been studied for many years (probably >60 years at this point).

6. L55: Should be, "horizontal grid spacing" rather than "horizontal resolution".

7. L77: I would say, "high resolution mesoscale simulation" rather than "very high resolution" considering dx=500 m is not high resolution with respect to typical LES scales (10s of meters).

8. L77: Please explain briefly what is mean by "gray zone" and provide references.

9. L82: Should be, "findings" rather than "foundings".

10. L88-92: Some references would be nice here, especially for an audience who is not familiar with this region.

11. L99-100: Please add references for why this wind direction indicated a weakened East Asia winter monsoon.

12. L103: Why not use a newer version of WRF? Version 3.9.1.1 was released years ago (August 2017, I believe).

13. L108-109: Were eta levels specified? It would be important to report the vertical grid spacing near the surface.

14. L126: Discussing the "gray zone" here is fine, but please reference this section in the introduction so that the reader is not confused by what you mean by the terminology.

15. L126-130: Please provide references for the "gray zone" discussion as well as the Deardorff SGS TKE scheme.

16. L139: I disagree that the "Lin scheme is a sophisticated scheme", considering it is single moment and there are several double (or higher order) moment schemes.

17. L142: These microphysics schemes are not new, as they are over 10 years old now.

18. L144: Please define "ThompsonAA".

19. L154-155: How exactly are the data screened? Please explain.

20. L187: Only in the extreme case of QNSE did the absolute difference exceed 10 m/s; however, as written currently it sounds like many simulations have large differences.

21. L202-204: Do you have proof for the claim about QNSE? Are there other studies that support this claim? I wonder if MYNN used EDMF or just ED in these simulations. I know that in newer versions of WRF, EDMF is default for EDMF.

22. L209-210: I don't understand this sentence. Wind direction depends upon the u and v-components, so one can have the correct wind speed but not the correct wind direction because the components are incorrect.

23. Fig. 5: There are repeat colors in this figures, please fix this.

24. L252-253: It would be good here to briefly confirm/reiterate the best combination schemes for wind direction.

25. L265-266: The difference in statistical measures between the ensemble and WDM6 is very small, suggesting the ensemble is not improving things too much.

26. L266-267: "reduces model bias by approximately half", compared to what?

27. L271-272: Please provide references.

28. L287: This is a broad statement. What exactly do you mean by it with respect to future model developments? Perhaps some discussion should be added to Section 4.

29. L289: "previous investigations", please add references.

30. L305: "surface topography does not induce new uncertainties into the simulation", this is simply not true. What about errors associated with horizontal diffusion and topographic data sets? These actually worse as terrain becomes steeper/more complex.

31. L321-322: In general, the QNSE finding is surprising, given that it was originally developed for stable conditions. Do your results agree with other studies that have used QNSE under stable conditions? Discussion here is needed.

32. L336-337: This broad statement about these "gray zone" resolutions being used "rarely" in previous simulation studies is not true. There are dozens (if not hundreds) of papers looking at gray zone modeling, especially in the last ~5-10 years. Please be more specific with your statement.

33. L342: To which PBL scheme parameters are you referring? It would be good to give some examples.

34. Figs. 9 and 10: Please note in the caption that the y-axis range is not the same (alternatively, make them the same).

---

## Author Comment (AC2)

**Response to referee comments**

**We are grateful for the thoughtful and constructive feedback from the reviewer. We have revised the text in the manuscript to answer the referee's points and we believe this revision has improved the clarity and quality of our manuscript. This response provides a complete description of the changes that have been made in response to each comment. Referee comments are shown in plain text, author responses are shown in bold blue text. All line numbers in the responses refer to locations in the revised manuscript.**

**Referee Comments**

Title: Impact of physical parameterizations on wind simulation with WRF V3.9.1.1 over the coastal regions of North China at PBL gray-zone resolution

Author(s): Entao Yu et al.

MS No.: gmd-2022-53

MS type: Model evaluation paper

**General Comments**

This paper presents a sensitivity case study of WRF wind forecasts under calm and stable conditions with a systematic variation of planetary boundary layer (PBL), microphysics (MP), and radiation parameterizations. For a case study this work is result of an extensive computational effort. However, there are some aspects in the methodology that need clarification and better motivation. Considering the impressive volume of the generated model data, the presented evaluation is limited, and I suggest expanding on the analysis to improve the scientific quality of the paper. Several findings require more insightful interpretations and discussions; some of the presented conclusions require clarification or correction. The presentation of the data and results can be improved upon.

*Response: Thank you for taking time out of your busy schedule to review this paper, I really appreciate all your comments and suggestions. Please find my responses in below and my revisions/corrections in the re-submitted files. Thanks again.*

**Specific Comments**

(1) This study aims to assess the ability of various physics parameterization configurations to predict **calm and stable weather conditions that favor air pollution**. However, the authors verify merely wind speed (and wind direction to a lesser degree). In order to gain insight what drives these differences in wind speed and to evaluate atmospheric properties that are crucial for air quality (such as static stability), it would be valuable to further assess vertical profiles.

*Response: Thanks for the suggestion, according to this comment, we extended the analysis of wind direction by adding two figures (figure6 and figure8), and added the evaluation of vertical profile using a sounding station (location shown in figure1) in section 4.2, the revisions are as follows:*

*(1) Lines 227-230 and figure 6: "The sensitivity of wind direction to the MP schemes is also low, as the wind roses from simulations with different MP schemes are very similar (Figure 6). WDM6, NSSL2 and*

*ThompsonAA show the best CORR score of 0.52, followed by Thompson and CAM5. Meanwhile, WSM3 is the best scheme according to the BIAS score, and ThompsonAA is the best scheme according to the RMSE score".*

*(2) Lines 239-242 and figure 8: "Figure 8 shows the wind roses during 11-15 January 2019 from simulations with different SW-LW schemes and the corresponding statistic scores. The simulations indicate the wind is mostly from the southwest direction during the study period, which is different from the observation. According to the CORR score, Dudhia-RRTM is the best scheme with the highest value (0.55), meanwhile, RRTMG shows the best BIAS score of -16°, and Dudhia-RRTM shows the best RMSE score of 61°".*

*(3) Lines 331-340 and figure 14: "4.2 Vertical profile of wind speed*

*Figure 14 shows the observed and simulated vertical profile of wind speed at 08:00 and 20:00 during the study period, the location of the sounding station is illustrated in Figure 1. YSU reproduces the vertical structure of wind speed reasonably, with slightly larger model bias above the height of 15 km. Within the low levels below 2.5 km, simulated wind speed from the YSU scheme is close to the observation, with the bias lower than 2.5 m/s in most cases. Meanwhile, QNSE shows worse performance in reproducing the vertical structure of wind speed, with significant larger model bias compared to YSU. for example, QNSE overestimates the low-level (< 2.5 km) wind speed by about 10 m/s at 20:00 on 11 January 2019, and overestimate wind speed by 20 m/s at 20:00 on 11 January 2019. It is interesting to note that the simulation with QNSE is pretty similar to that with YSU at 08:00 during the study period, indicating that large difference between YSU and QNSE only occurs at specific time during the study period, which is also revealed in Figure 3a."*

(2) This paper presents a **case study** for a specific location and event, and the manuscript should be framed accordingly (in the title, results, and discussion). However, the authors draw general conclusions from their findings and presume transferability of their results across the world and across variables (e.g., lines 81-83 and lines 337/338) and generalize shortcoming in the WRF model (lines 305/306). The current manuscript lacks an objective error discussion on the various sources of uncertainty and limitations of the study: The presented findings could be unique to the meteorological setup of the event, the location, the input dataset, the domain setup, other unchanged parameterization types or model settings, etc.

*Response: Thanks for the suggestion, the points are well taken, we removed the general statements and focused our findings to the specific case and location, as follows:*

*(1) Lines 28-30: "Our results indicate the roles parameterizations play in wind simulation under stable weather conditions and provide a valuable reference for further research in the study area and nearby regions."*

*(2) Lines 91-92: "be helpful in the wind and air quality forecast in the study area and other coastal regions of China under stable weather conditions."*

*(3) Lines 377-379: "Finally, it is worth pointing out that the presented findings in this study could be unique to the meteorological setup of the event, the location, the input dataset, the domain setup, and other unchanged parameterization types or model settings."*

(3) It is not clear how the authors arrive at the conclusion that "**wind direction** is insensitive to changes in physics parameterizations" (line 207 & 213-15). The manuscript only shows variations between PBL-scheme groups and differences are visible in figure 4 and table 5. The authors have not performed any hypothesis testing to show whether these differences are actually insignificant compared to the variation in wind speeds. The authors write that "the model RMSE and BIAS values for different PBL schemes are similar", yet table 5 reveals that the worst PBL scheme (namely, BouLac) has a wind-direction Bias that is 154% larger than the Bias of the best PBL scheme (namely, LES – excluding QNSE), see table 5. In comparison the wind- speed Bias of the worst PBL scheme (namely, TEMF) is 126% larger than the Bias of the best PBL scheme (namely YSU), see table 4.

*Response: Thanks for the comments, the points are well taken, and we added the results of wind direction for MP and radiation schemes (Figure 6 and Figure 8), and we extended the analysis of wind direction in Figure 9, as follows:*

*(1) Lines 227-30 and figure 6: "The sensitivity of wind direction to the MP schemes is also low, as the wind roses from simulations with different MP schemes are very similar (Figure 6). WDM6, NSSL2 and ThompsonAA show the best CORR score of 0.52, followed by Thompson and CAM5. Meanwhile, WSM3 is the best scheme according to the BIAS score, and ThompsonAA is the best scheme according to the RMSE score".*

*(2) Lines 239-242 and figure 8: "Figure 8 shows the wind roses during 11-15 January 2019 from simulations with different SW-LW schemes and the corresponding statistic scores. The simulations indicate the wind is mostly from the southwest direction during the study period, which is different from the observation. According to the CORR score, Dudhia-RRTM is the best scheme with the highest value (0.55), meanwhile, RRTMG shows the best BIAS score of -16°, and Dudhia-RRTM shows the best RMSE score of 61°".*

*(3) Lines 252-253: "For the wind direction simulation, LES combined with Dudhia-RRTM shows the best CORR score, while TEMF is the best scheme according to BIAS and RMSE scores."*

(4) A considerable amount of computational resources was spent to test **16 MP schemes** - this is the largest number of schemes within any of the parameterisation categories tested in this paper. Although the authors cite Cheng et al. (2013) to justify that MP can have an impact on wind fields, there is little justification to evaluate MP sensitivity to this extent under the dry and stable conditions of this case study. As opposed to the conditions described in this manuscript, Cheng et al. (2013) pointed out that MP can affect wind fields for convective weather phenomena during the summer (with weak large-scale forcing) that are associated with gust fronts / outflow boundaries from cold pools that result from strong downdrafts of thunderstorms. One would not expect to see significant impact from the choice of MP parameterization on wind forecasts under stable conditions, hence, these results are not surprising. Instead of assessing a vast amount of MP schemes without valid motivation, it could have been informative to expand on the number of radiative schemes and/or include an investigation on the impact of the land surface schemes.

*Response: Thanks for the comments, we totally agree with the comments on the MP schemes, we added an investigation on the impact of land surface schemes in section 4.4, lines 350-360:*

*"4.4 Impact of land surface model*

*Figure 16 shows the evaluation of different land surface parameterizations with the same model configuration as the simulation with best the Taylor skill score, the land surface models (LSM) considered are the five-layer*

*thermal diffusion scheme (SLAB, Dudhia, 1996), the Noah scheme (NOAH, Chen and Dudhia, 2001), the Rapid Update Cycle scheme (RUC, Smirnova et al., 2000), the Noah-MP scheme (NOAHMP), and the Community Land Model Version 4 scheme (CLM4, Lawrence et al., 2011). The simulations with different LSMs reproduce the timeseries of wind speed well, with larger spread among the LSMs during 14-15 January 2019 (Figure 16a). NOAHMP shows the best CORR score of 0.94, CLM4 and NOAH are slightly worse according to this score. Meanwhile, according to the RMSE score, NOAHMP is the best scheme, followed by RUC and NOAH. In addition, RUC and NOAHMP show better BIAS scores than the other LSMs. Thus, NOAHMP shows the best performance among different LSMs in wind-speed simulation under stable conditions in this study, however, the large difference among LSMs indicates that we should take land surface parameterizations into consideration in future studies."*

(5) **Tables and Figures:**

Tables 4-10 show verification data values that are not always easily to grasp from plain numbers in this quantity. The authors should consider visualizing this data for better presentation to the reader. For example, this could be in the form of bar plots, boxplots or heatmaps. I also encourage the authors to include the distribution across the 105 stations (e.g., the range across stations could be shown in box plots or with error bars on bar or line plots). The verification plots could be combined with the respective timeseries plot (e.g., figure 3 would have a subplot visualising the error metrics from table 4).

*Response: Thanks for the comments, according to the suggestion, we plotted the data of Tables 4-10 as subplots of the timeseries figures, and the range across stations was included in the subplots. Thus, in the revision, Tables 4-10 were removed and shown as subplots in Figure 2-8, 10-12, the analysis of these figures are revised accordingly.*

Figure 2: This figure shows identical data to other figures - i.e., 2a is repeated in figures 3, 5, 6, and 8, whereas 2b is identical to the top left panel in figure 4. This makes figure 2 redundant. An alternative could be to show the observational data of Tangshan city as a specific example in connection to the paragraph in line 96-101. When updating, please also include a more descriptive title to figure 2a.

*Response: Thanks for the comments, figure 2 is redundant, thus we removed it, and added synoptic plots of the stable weather event as figure 2 in the revision, we also added the information of Tangshan city in figures 1 and 2.*
*The description of wind rose charts are revised to "for each wind rose chart, the circles represent the relative frequency (%), and the colors represent wind speed (m/s)" in lines 658-660.*

(6) Lines 247-249: For wind speed Bias and RMSE, within each radiation group the same PBL schemes rank best; and within each PBL group, the same radiation schemes rank best. This indicates that a systematic variation of parameterizations as presented in this paper, is *not* necessary. (E.g., for wind-speed RMSE no matter which PBL scheme, Dudhia or Goddard radiation are always best.) However, for CORR and wind direction, this pattern is not always consistent, which indicates that a systematic variation of parameterizations *is* useful when focusing on these variables. (E.g., for wind-direction RMSE, the best radiation scheme depends on the choice of PBL scheme

- for TEMF PBL, RRTMG radiation is best; for BouLac PBL, Goddard radiation is best; for LES, CAM radiation is best.)

*Response: Thanks for the comments, points are well taken, we revised in lines 255-261: "Overall, for BIAS and RMSE scores of wind speed, within each SW-LW group, the same PBL scheme ranks best (e.g., for wind-speed RMSE, no matter which PBL scheme, Dudhia-RRTM and Goddard are always best), and within each PBL scheme, the same SW-LW group ranks best, this indicates that a systematic variation of parameterizations is not necessary. However, for wind-speed CORR and wind direction, this pattern is not always consistent. For example, for wind-direction BIAS, the best SW-LW group depends on the choice of PBL scheme (e.g., for MYDM7 with TEMF, Dudhia-RRTM is best; for P3 MP scheme with TEMF, RRTMG is best), this indicates that a systematic variation of parameterizations is important when focusing on these variables".*

(7) **Evaluation of model configurations with the best individual performance** (lines 254-269):

Lines 266/267: This statement is misleading. Perhaps correct to "the ensemble of the top *four* configurations reduces model bias by approximately half *compared to the ensemble that uses all configurations,* while the CORR value *of the super-ensemble mean* was highest *among* all the *ensembles.*" Note that the highest CORR seen in table 8 (0.937) is result of either the single-model configuration using Goddard MP, or the ensemble using all configurations. The lowest BIAS (0.331) is result of the single-model configuration using MYDM7. And the lowest RMSE (0.524) is result of either the single-model configuration using NSSL1, or the ensemble using the 4 best configurations. This disagrees with the concluding statement in line 339-341.

*Response: Thanks for the comments, it was revised in lines 273-277: "The result indicates that ensemble mean of four simulations with WDM6, Goddard, NSSL1 and MYDM7 shows the best BIAS and RMSE scores. Figure 10a shows the time series of wind speed from ensemble of 4 (ENS4) and all simulations (ENSall), the spread of ENS4 is significantly lower than ENSall, and ENS4 shows smaller difference with the observation compared to ENSall. According to the statistic scores, ENS4 reduces model bias by approximately half compared to ENSall".*

*Lines 369-373: "The ensemble mean of 576 simulations in our study shows the best CORR score (0.94) in wind-speed simulation, however, this best CORR score is also result of single-model simulation with Goddard MP scheme. At the same time, the best wind-speed BIAS score (0.33 m/s) is result of the single-model simulation with MYDM7 or ETA, and the best RMSE score (0.52 m/s) is result of either the single-model simulation with Goddard, NSSL1, MYDM7, or the ensemble using the 3, 4 or 5 best simulations. Thus, model ensemble does not always provide the best performance"*

For simulations that struggle with systematic overprediction, it is implied that an ensemble of a subgroup of members with smaller biases improves the ensemble-mean bias compared to using all members. However, (a) systematic errors can be significantly reduced with bias- correction, and (b) ensembles generate probabilistic forecasts and the authors present no discussion on the probabilistic features of their ensembles (e.g., ensembles with narrow spread are often under-dispersive / overconfident). More on this in point 8).

*Response: Thanks for the comments, according to them, we revised the discussion in lines 367-375: "As none*

*of the schemes is the best according to all the scores, model ensemble is used to provide optimizing model performance, at the same time, model ensemble also provides probabilistic evaluation of the simulations and ensembles with narrow spread are often overconfident. The ensemble mean of 576 simulations in our study shows the best CORR score (0.94) in wind-speed simulation, however, this best CORR score is also result of single-model simulation with Goddard MP scheme. At the same time, the best wind-speed BIAS score (0.33 m/s) is result of the single-model simulation with MYDM7 or ETA, and the best RMSE score (0.52 m/s) is result of either the single-model simulation with Goddard, NSSL1, MYDM7, or the ensemble using the 3, 4 or 5 best simulations. Thus, model ensemble does not always provide the best performance, and model post-processing, especially the bias correction techniques are needed to be taken into consideration, which can significantly reduce the systematic errors in model simulation."*

Please clarify if the SD in the Taylor skill scores was calculated over the various stations and averaged over time, or if SD was calculated over the time series and averaged over stations. This is important to understand why the MP scheme with best CORR is not also the scheme with best Taylor skill score considering that all MP schemes have very similar temporal patterns (as seen figure 5). If the variation across stations is significant among MP schemes, it would be informative to analyse this spatial variation.

*Response: Thanks for the comments, SD is calculated over the time series and averaged over stations, and the variation across stations is significant among MP schemes. According to the comments, we added the range across the 105 stations in the revised figures, and we added an investigation on the spatial variation in section 4.1, lines 319-330: "Figure 13 shows the spatial distribution of observed and simulated wind fields during the study period, we choose 14:00 in local time as an example. The simulation using YSU, Dudhia-RRTM and WDM6 schemes is referred to as YSU, and the simulation using QNSE, Dudhia-RRTM and WDM6 is referred to as QNSE. YSU generally reproduces the wind field in the study area, especially in terms of wind speed. For example, the observed wind speed is lower on 13 January 2019, with values lower than 2 m/s in many stations, while on 15 January 2019, the observed wind speed is higher than 4 m/s in most of the stations. In the simulation with YSU, wind speed is about 2 m/s on 13 January 2019 and higher than 4 m/s on 15 January 2019 over the study area, which is close to the observation. On the contrary, simulation with QNSE fails to reproduce the distribution of wind speed, and shows strong overestimation, especially over the mountain areas of the study area (Figure 1a), for example, the peak wind speed in simulation with QNSE exceeds 20 m/s on 15 January 2019, which is more than five times greater than the observation, this overestimation is consistent with the large positive bias in previous investigation of Figure 3. For the wind-direction simulation, YSU shows degraded performance compared to wind speed, and generally fails to reproduce the wind-direction distribution for most of the stations, QNSE also fails to do so."*

Table 8 – If the 10 best WRF configurations are based on the ranks of Taylor skill scores, please add the ranks with corresponding Taylor skill score values to the table.

*Response: Thanks for pointing out this, it was revised in line 264: "the scores range from 0.2 to 1.0, with the best 10 WRF configurations having similar scores of about 1.0".*

(8) The authors present correlation coefficients, biases, and RMSE without any insightful discussion what the different verification metrics represent and why it is conceivable that some schemes perform best according to one metric and worst according to another. It is also not discussed that the performance of raw model forecasts can be significantly enhanced by **post-processing** - in particular the systematic-errors component (biases), which appear to be the main issue in this case study. As already mentioned in point 7), the author's suggestion of the ensemble generation by picking a small number of members with lowest bias without considering the effects of calibration is problematic.

*Response: Thanks for the comments, we revised them in lines 362-375: "In this study, CORR, BIAS and RMSE are used as verification scores, CORR is a measure of the strength and direction of the linear relationship between simulation and observation, BIAS is a measure of the mean difference between simulation and observation, and RMSE is the square root of the average of the set of squared differences between simulation and observation, thus each of this score gives a partial view of the model performance, and some schemes perform best according to one metric and worst according to another in our previous investigation.*

*As none of the schemes is the best according to all the scores, model ensemble is used to provide optimizing model performance, at the same time, model ensemble also provide probabilistic evaluation of the simulations as ensembles with narrow spread are often overconfident. The ensemble mean of 576 simulations in our study shows the best CORR score (0.94) in wind-speed simulation, however, this best CORR score is also result of single-model simulation with Goddard MP scheme. At the same time, the best wind-speed BIAS score (0.33 m/s) is result of the single-model simulation with MYDM7 or ETA, and the best RMSE score (0.52 m/s) is result of either the single-model simulation with Goddard, NSSL1, MYDM7, or the ensemble using the 3, 4 or 5 best simulations. Thus, model ensemble does not always provide the best performance, and model post-processing, especially the bias correction techniques are needed to be taken into consideration, which can significantly reduce the systematic errors in model simulation."*

(9) Section 2.1: Please add a more comprehensive **synoptic analysis** of the event. Consider adding a weather analysis map and/or radio soundings to show the observed stable stratification (perhaps in figure 2). Considering the substantial testing of MP schemes, it would be useful to mention if there was any cloudiness. With regards to section 3.2.2 it would be interesting to analyse why observed wind speeds decrease with elevation.

*Response: Thanks for the comments, we removed figure 2 and added the distribution of cloud, geopotential height and winds as a new figure. The descriptions are presented in lines 107-115: "Figure 2 depicts the distribution of geopotential height, winds, and cloud fraction during the study period, the geopotential height and winds data are from the ERA5 dataset (Hersbach et al., 2020), and the cloud fractions are from satellite observations of CLARA (CM SAF cLoud, Albedo and surface Radiation) product family (Karlsson et al., 2021). Figure 2 indicates that a weak high-pressure system persisted from 11 to 13 January, along with weak southwest wind in the study area, which would transport warm and wet air to the study area (Gao et al., 2016a), creating a favorable moisture condition for stable conditions and inhibiting pollutants dispersal (Zhang et al., 2014; Hua and Wu, 2022). Then the high-pressure system was replaced by strong northwest wind from 14 to 15 January 2019. The CLARA observations indicate cloud fraction exceeding 60% on 12*

*January at the study area, while for the rest of the time, cloud fraction is low. This stable event is used to investigate the impact of physical parameterizations of the WRF model".*

(10) Ensemble spread usually grows with **forecast lead time** as predictive skill declines. Please discuss how the authors explain the narrowing ensemble spread in figure 5.

*Response: Thanks for the comments, it is true that in weather forecasting operations, the ensemble spread of WRF model usually grows with forecast lead time as predictive skill declines, which is closely related to the growth of uncertainties in the global climate models that provide initial and lateral boundary conditions for the WRF model. In our study, the WRF model is driven by the ERA5 reanalysis dataset, so the simulations can be considered as dynamical downscaling simulations with imposed "perfect boundary conditions", thus the uncertainties of driving data will not grow significantly with time. At the same time, all the WRF simulations are initialized at the same starting points (00:00 UTC January 9th, 2019) with identical lateral boundary conditions, and they are configured as "climate simulations" with SST updated every 3 hours. All the aforementioned setups help to minimize the external uncertainties, making the variation in the results are only caused by the physical parameterization schemes. As the wind speed under stable conditions are insensitive to the MP parameterizations, the ensemble spread in that figure is narrow.*

**Technical Corrections**

The title needs to state that (a) this is a case study and (b) calm wind speeds under stable conditions are investigated.

*Response: Thanks for the comments, the title was revised to "Impact of physical parameterizations on wind simulation with WRF V3.9.1.1 under stable conditions at PBL gray-zone resolution: a case study over the coastal regions of North China".*

There are inconsistencies with the tense, please make consistent everywhere.

E.g., line 25/26 "The MYNN scheme *showed* the highest correlation among all PBL schemes, while LES and YSU *has* the smallest model errors, the RRTMG and Goddard schemes *showed* the highest correlations …"

*Response: Thanks for pointing out this, we revised it in lines 23-25: "for coastal stations, MYNN shows the best temporal correlation with observations among all PBL schemes, while Goddard shows the smallest bias out of SW-LW schemes, these results are different from that of inland stations."*
*We also checked the whole paper and corrected the errors.*

Line 14: Technically wind speeds are always zero at the surface. The wind simulations in this paper likely correspond to the 10-m above surface level.

*Response: Thank you for pointing out this, "surface wind" was replaced by "near-surface wind at 10-meter height" in Line 14 as: "different physical parameterization schemes impact simulated near-surface wind at 10-meter height over the coastal regions of North China", we also checked and revised them throughout the*

*paper.*

Line 18: "The data show that *the* WRF model …"
*Response: Thank you for pointing out this, the errors were corrected.*

Line 25: "For example, for the weather stations located in coastal regions [*no new sentence*] the MYNN …"
*Response: Thank you for pointing out this, the errors were corrected.*

Line 35/36: "(Zhang et al., 2014; Cai et al., 2017; Zhang et al., 2015)" – please sort references either alphabetically or according to publication time (check journal guidelines) throughout the manuscript
*Response: Thank you for pointing out this, the references were checked and the errors were corrected.*

Line 36-39: Please provide a brief explanation on how increasing global temperatures relate to haze and low wind speeds.
*Response: Thanks for the comment, the explanation was added as follows in lines 38-41: "Projections of future climate change suggest that global temperatures will increase, and the frequency of conducive weather conditions to severe haze is projected to increase substantially in response to the climate change, which in turn may increase the frequency of haze events over North China (Cai et al., 2017),".*

Line 39/40: Before saying that it is crucial to improve wind predictions because haze events are hazardous and may become more frequent in future, inform the reader about the problems with WRF wind forecasts - i.e., include a short literature review on known biases and challenges in simulating wind fields in your region of interest.
*Response: Thanks for pointing out this, we revised it in lines 40-43: "frequency of haze events over North China (Cai et al., 2017), however, numerical models always show large bias in wind prediction over China (Gao et al., 2016b; Zhao et al., 2016; Pan et al., 2021), thus it is crucial to improve wind prediction under stable weather conditions in order to minimize associated economic losses and environmental impacts".*

Line 49: "choosing appropriate combinations is  important"
*Response: Thanks for pointing out this, the errors were corrected.*

Line 67: Start a new paragraph to separate the sections on MP vs radiation.
*Response: Thank you for pointing out this, we revised the paper according to this comment.*

Line 72/73: Add "to this extent", "in China" and "to our knowledge". WRF wind performance has been evaluated in a systematic manner in other studies, e.g., Fernández-González et al. 2018, Santos-Alamillos et al. 2013, Siuta et al. 2017.
*Response: Thank you for pointing out this, according to the comment, it was revised as follows in lines 80-82: "Most of the aforementioned studies considered a small number of parameterization schemes, to the best of*

*our knowledge, the sensitivity of parameterizations on wind simulation has not yet been explored in a systematic way in China."*

*We also mentioned other systematic evaluations in lines 77-79: "The impact of parameterization combination on WRF performance has been investigated in previous studies (Santos-Alamillos et al., 2013; Fernández-González et al., 2018)"*

Line 74/75: "error compensation among processes […] may predict incorrect wind patterns" If errors compensate each other, that would imply that the result would be more accurate; however, in the chaotic system of the atmosphere errors are usually amplified and grow from various imperfections in the model.

*Response: Thanks for pointing out this, according to this comment, we revised in lines 76-77: "The interactions among physical parameterizations are also vital to wind simulation, as they may alter the processes of atmosphere-land interactions, radiation transport, and moist convection, and amplify the uncertainties in wind prediction."*

Lines 77 & 81: "*the* WRF model" (also check everywhere else in the manuscript)

*Response: Thank you for pointing out this, the errors were corrected. We also checked and corrected the errors in other places.*

Lines 82: "founding"?

*Response: Thank you for pointing out this, it was corrected by "finding" in the revision.*

Line 86: The caption repeats all sub-captions. I suggest re-naming it to "Data and Methods" or similar.

*Response: Thank you for pointing out this, the title was revised to "Data and Methods".*

Line 87: "stable weather events in 2019" implies a series of events – this paper only discusses a single event!

*Response: Thanks for the comment, according to it, we revised the title to "2.1. Study area and the stable weather event in 2019".*

Line 92: "favorable weather conditions" – be more specific please

*Response: Thanks for pointing out this, it was revised to "favorable weather conditions with lower wind speed" in line 101.*

Line 94: "synoptic *forcing*"?; "vertical mixing *in* the atmosphere"; "increasing the stability of surface air"
– air in the boundary layer?

*Response: Thank you for pointing out this, according to the comment, it was revised to "anomalous southerly wind in the lower troposphere caused by the weak East Asian winter monsoon weakened the synoptic forcing and extent of vertical mixing in the atmosphere, thus increased the stability of air in the boundary layer and the local concentration of hazes (Zhang et al., 2014)" in lines 102-105.*

Line 97: Could the location of Tangshan city (or the station measuring these values) please be included to figure 1? Thank you!

*Thank you for pointing out this, the location of Tangshan city was included in Figures 1 and 2.*

Line 98: Are any statistics available on the socioeconomic or health impacts of this event that could be included to exemplify the severity of the event (e.g., increased hospitalization rates)?

*Response: Thanks, this is a very good suggestion, however, we did not get such economic information for this haze event from the public sources. Sorry for that.*

Line 104: Plural "simulations"

*Response: Thank you for pointing out this, the errors were corrected.*

Line 109: Better "within the PBL" specifically for this event, how many levels were in the PBL on average and at the minimum?

*Response: Thanks for pointing out this, according to the comment, we revised it to "levels existed within the PBL at any time". The eta values for the first 10 levels are 0.996, 0.988, 0.978, 0.966, 0.956, 0.946, 0.933, 0.923, 0.912, and 0.901, for the event in our study, about 6 levels on average were in the PBL during the first 4 days with low PBL height, for the last day with high PBL height, there are more levels on average.*

Line 112: Why was such a long spin-up time selected? It is important to note that the model dataset was generated from a single initialization with a 7-day forecast horizon and that forecast skill is expected to degrade with lead time.

*Response: Thank you for pointing out this, in our study, WRF is driven by ERA5 reanalysis data, not the forecast products, and we update SST data during the simulation period, thus the simulations can be considered as dynamical downscaling simulations driven by "perfect boundary conditions", and the model skill will not degrade obviously along with the simulation time. The study of Kleczek et al. (2014) indicated that longer spin-up time decreased the modelled wind speed bias, thus in our study, we selected a spin-up time of 40 hours, and the results are satisfactory.*
*References:*
*Kleczek, M.A., Steeneveld, GJ. & Holtslag, A.A.M. Evaluation of the Weather Research and Forecasting Mesoscale Model for GABLS3: Impact of Boundary-Layer Schemes, Boundary Conditions and Spin-Up. Boundary-Layer Meteorol 152, 213–243 (2014). https://doi.org/10.1007/s10546-014-9925-3*

Line 112: What are the default parameterizations? Please name and reference them.

*Response: Thank you for pointing out this, the parameterization and the corresponding references were listed in Table 2 in line 639, which is cited in line 127: "Firstly, the default physical parameterization schemes (Table 2) are applied in the single set of WRF simulations"*

Line 113/114: Please clarify: Were the simulations first run for D01 and D02, then D03 was initialized with the

output from D02, or was each of the 640 simulations run with one-way-nested feedback across all three domains at each time step?

*Response: Thanks for the comment, the simulations are first run for D01 and D02, then D03 is forced with the output of D02 using ndown. it was revised in lines 127-129: "Firstly, the default physical parameterization schemes (Table 2) are applied in the single set of WRF simulations for the outer two domains (D01 and D02), and then the output of D02 is used to drive inner domain simulations with different combinations of PBL, MP, and SW-LW schemes (see section 2.3)".*

Line 117/118: "The lateral boundary *conditions* and sea surface temperature were updated …" also, which dataset did the SST come from?

*Response: Thank you for pointing out this, it was revised to "The lateral boundary conditions and sea surface temperature are updated every three hours using the ERA5 reanalysis data" in lines 132-133.*

Line 118: wind was *calculated,* or output retrieved from the model?

*Response: Thank you for pointing out this, wind was from the model, it was revised to "the frequency of wind retrieved from WRF output was hourly, which matches the frequency of observations in the study area" in line 133.*

Line 121 & 124: Add comma "e.g., …" & "i.e.,"  please check throughout the manuscript

*Response: Thank you for pointing out this, the errors were corrected, and we also checked the whole manuscript.*

Line 126: The gray zone is first mentioned at Line 77 and should be defined there. After definition, the quotation marks for this term can be removed.

*Response: Thank you for pointing out this, in the revision, we introduced "gray zone" in lines 84-86: "which belongs to the PBL "gray zone" resolution that is too fine to utilize mesoscale turbulence parameterizations and too coarse for a large-eddy-simulation (LES) scheme to resolve turbulent eddies (Shin and Hong, 2015; Honnert et al., 2016)"
Then we mentioned it in lines 140-141 "As the horizontal grid spacing of 0.5 km is within the PBL gray zone resolution, both PBL and LES assumptions are imperfect".*

Line 134: Both "atmospheric boundary layer" or "planetary boundary layer" are fine, but please be consistent throughout the manuscript.

*Response: Thank you for pointing out this, we have checked the whole paper and revised them accordingly.*

Lines 139-145:This paragraph should be revised for a more insightful summary.

The concept of single- vs double-moment and hydrometeor classes should be briefly explained.

Explain why some MP schemes are "suitable for high-resolution simulations".

The Goddard has a reference from 1989 and is described as one for the "new schemes".

NSSL1 has no reference in Table 2.

*Response: Thank you for pointing out this, we have revised them in lines 151-161: "Sixteen MP schemes are applied in this study (Table 1), Lin, WSM3, WSM5, ETA, WSM6, Goddard, SBU, and NSSL1 schemes are the single-moment bulk microphysical scheme, which predicts only the mixing ratios of hydrometeors (i.e., cloud ice, snow, graupel, rain, and cloud water) by assuming particle size distributions. The other eight schemes (Thompson, MYDM7, Morrison, CMA, WDM6, NSSL2, ThompsonAA and P3) use a double-moment approach, predicting not only mixing ratios of hydrometeors but also number concentrations. Among them, two types of hydrometeors are included in WSM3 (cloud water and rain), three types of hydrometeors are included in ETA (cloud water, rain, and snow) and P3 (cloud water, rain, and ice), four types of hydrometeors are included in WSM5 and SBU (cloud water, rain, ice, and snow), five types of hydrometeors are included in Lin, WSM6, Goddard, Thompson, Morrison, CAM, WDM6 and ThompsonAA (cloud water, rain, ice, snow, and graupel), six types of hydrometeors are included in MYDM7, NSSL1, and NSSL2 (cloud water, rain, ice, snow, graupel, and hail). According to the user's guide of ARW (Skamarock et al., 2008), WSM6, Thompson, Morrison, WDM6, NSSL1, and NSSL2 are suitable for high-resolution simulations.", and the reference for NSSL1 was added in Table 1.*

Line 146: Table 3 is not referenced anywhere in the manuscript. Tables 1, 2, and 3 could be combined.

*Response: Thanks for the comments, we have combined Tables 1-3 to Table 1 in the revision.*

Line 155: How did you define / identify "spurious jumps"?105 stations remained – out of how many?

*Response: Thanks for the comments, during the stable event, some sensors were frozen and the data are all zeroes, so we skipped the stations. We revised them in 169-171: "All data are screened before analysis in order to remove stations with data showing spurious jumps (e.g., wind speed jumps to 0 m/s due to frozen sensor). After this filtering, 105 out of 132 weather stations (Figure 1a) remained, including 89 inland stations and 16 coastal stations."*

Line 167: Also define "i"

*Response: Thank you for pointing out this, the errors were corrected.*

Line 170/171: Where in the paper is this metric considered?

*Response: Thanks for pointing out this, the BIASs and RMSEs of wind direction are calculated based on this metric.*

Line 174: This equation misses sums.

*Response: Thanks for the comment, the errors were corrected.*

Lines 177-180: Are there 105 or 106 stations? This paragraph can be skipped.

*Response: Thank you for pointing out this, we have removed the paragraph.*

Line 183: Wind speed data is not directly *produced by* PBL schemes. Wind speed is a dynamic variable that is adjusted by the PBL scheme. So perhaps the data "is produced *using* PBL schemes".

*Response: Thanks for pointing out this, it was revised to "Figure 3 shows the time series of observed wind speeds and the corresponding simulations using different PBL schemes".*

Figure 3 and all other figures with time series: Clarify in your manuscript whether these are UTC or local times. If times are in UTC, please mention to which local times these translate. Please also include tick marks for each date and possibly a vertical line separating each day as a reference for diurnal periods.

*Response: Thank you for pointing out this, according to the comment, we change the timeseries plots to include tick marks and vertical lines for each day in the revision (Figure 3, 5, 7,10,11,12,15,16). The time in all the figures is local time, which is clarified in line 194: "Figure 3 shows the time series of observed wind speeds in local time and the corresponding simulations", Line 126: "The WRF model is initialized at 00:00 UTC (08:00 in local time) January 9, 2019"*

Line 184: I am seeing that "The WRF model *exaggerates* the temporal variation of observed wind speed in the study area"

*Response: Thanks for the comment, it was revised in lines 196-197: "The WRF model generally reproduces the temporal variation of observed wind speed in the study area with exaggeration".*

Line 185: I disagree with the statement that "QNSE showed no obvious daily wind speed change during the simulation period" – The wind speed change is considerably larger than with all other schemes.

*Response: Thanks for pointing out this, it was revised to "…except for QNSE, with which the wind speed change is considerably larger than with all other schemes during the simulation period".*

Lines 186-189: This section needs revision. It is unclear which correlation the authors refer to; it needs to be clarified that the 10m/s bias applies to QNSE only; it should be elaborated what other studies found ("such models" – referring to the QNSE models or the general overprediction by all WRF models?) Note that a more thorough literature review is needed in the introduction which could be referred to here.

*Response: Thanks for the comments, it was revised as "Almost all the PBL schemes overestimate wind speed by 1 m/s, however, for the QNSE scheme, the largest overestimation exceeds 10 m/s during the daytime on 11 and 15 January 2019."*

*In the Introduction, lines 57-62: "A lot of studies indicate an overestimation of wind speed in WRF simulation with different PBL schemes (Jiménez and Dudhia, 2012; Carvalho et al., 2014a, b; Pan et al., 2021; Gholami et al., 2021; Dzebre and Adaramola, 2020), for example, Gómez-Navarro et al. (2015) investigated the sensitivity of the WRF model to PBL schemes by simulating wind storms over complex terrain at a horizontal grid spacing of 2 km. In that study, the WRF model was configured with the Mellor-Yamada-Janjic (MYJ) scheme and overestimated wind speed by up to 100%,"*

Line 194: "*on* January 15th" "partly due to faster observed wind speeds" – right, the bias looks multiplicative,

but probably also due to the general error growth in NWP with lead time.

*Response: Thank you for pointing out this, it was revised to "In addition, the spread within the PBL schemes is larger on 15 January 2019, partly due to high wind speed (> 4 m/s) or the general error growth in the model" in lines 201-202.*

Line 195/196: The description of what bold and italicized numbers mean belongs in the figure caption.

*Response: Thank you for pointing out this, we have changed table4 to figure and the paragraph was removed.*

Line 200, 224, 236: The authors often describe the next-best schemes as having "similar statistics".
Please be specific to avoid confusion. For example, "X1 is the best scheme, followed by X2 and X3" or "X1 shows the best verification score. X2 and X3 are slightly worse according to this verifications score."

*Response: Thank you for pointing out this, they were revised as follows:*
*"MYJ shows the best CORR score of 0.96, MYNN, ACM2 and UW are slightly worse according to this verification score. YSU is the best scheme in term of BIAS and RMSE with the lowest scores of 0.45 m/s and 0.61 m/s, flowed by MYNN (0.55 m/s and 0.70 m/s)."*
*"while MYDM7 is the best scheme according to BIAS and RMSE scores, followed by P3 and ETA."*
*"Dudhia-RRTM is the best scheme according to BIAS and RMSE scores, followed by Goddard"*

Line 200/201: A correlation coefficient of 0.643 would usually not be considered "extremely low". Please revise your wording.

*Response: Thanks for the comment, it was revised to "For the QNSE scheme, the maximum BIAS and RMSE scores for individual stations exceed 10 m/s and 16 m/s, indicating that it has problems in reproducing wind speed under stable conditions over the study area" in lines 208-210.*

Line 202/203: Please justify this assumption. What does other literature suggest?

*Response: Thank you for pointing out this, it is only a speculation, and may only applied to the specific area in our study, to avoid misleading, the sentence was removed.*

Lines 204/205: Note that QNSE is still included in figure 4 and table 5. Either exclude QNSE from there, or move the statement that QNSE will be omitted after referencing figure 4 and table 5.

*Response: Thank you for pointing out this, the statements were moved to lines 218-219: "Considering the large model bias in wind speed, all simulations using the QNSE scheme (64 simulations in total) are omitted from further investigation in order that these anomalous data do not affect our overall analysis".*

Line 206: Plural "wind roses" Wind roses in figures 2 and 4 need a legend description.

*Response: Thanks for pointing out this, the errors were corrected, the legend description were added in line 659: "for each wind rose chart, the circles represent the relative frequency (%), and the colors represent wind speed (m/s)".*

Line 210: "As the simulated wind direction was calculated using the wind speed *components*, the bias in modeled wind direction can be attributed to bias in the wind-speed *component* simulation." – this statement is somewhat redundant.

*Response: Thank you for pointing out this, we have removed the paragraph.*

Line 224: "P*3*" (not "P2")

*Response: Thanks for pointing out this, the errors were corrected.*

Line 230: "*ensemble* spread"

*Response: Thanks for the comment, it was revised to "The SW-LW schemes have a larger model ensemble spread".*

Line 233: " reduced this overestimation, and thus produced values that were  closer to weather station observations." – these differences are relatively small but consistent

*Response: Thanks for pointing out this, the errors were corrected.*

Line 238: "while simulations that employed the *Goddard* [not RRTMG] scheme showed the lowest BIAS values."

*Response: Thank you for pointing out this, the errors were corrected.*

Line 240: The authors use the term "physical components" interchangeably with "physics parameterizations". "Physical components" can be ambiguous, I recommend consistently referring to "parameterizations" throughout the manuscript.

*Response: Thanks for pointing out this, we revised them according to this comment.*

Line 243: "A total of 40 simulations" – it is actually only 36 because QNSE is missing.

*Response: Thank you for the comment, it was revised to "thus for each MP scheme, a total of 36 simulations (excluding QNSE) are evaluated in this way"*

Line 244: "which produced outcomes that were  consistent with other results" –which results? Other MP groups; previously shown results in this manuscript; or different studies? Please be specific!

*Response: Thank you for pointing out this, it was revised to "and the results are expected to be consistent with evaluations using other MP schemes".*

Line 247: "further investigation"

*Response: Thanks for the comment, the errors were corrected.*

Lines 247/248: "Dudhia and RRTM schemes , or the Goddard *schemes* " – Goddard also has two schemes, LW and SW

*Response: Thank you for pointing out this, the errors were corrected.*

Lines 256/257 and Table 8: "The PBL and LW/SW radiation schemes used in the 10 best schemes were YSU and Dudhia and RRTM" – This wording sounds like the authors decided upon using these PBL and radiation schemes, rather than this being a results of their analysis. If I understand correctly, I suggest clarifying "The best 10 WRF model configurations have in common that they use the same PBL and radiation schemes, namely YSU and Dudhia-RRTM. Due to the slight differences between models using different MP schemes, the 10 best performing WRF configuration only vary in MP option." or similar

*Response: Thanks for the comment, we have revised the paragraph accordingly in lines 265-267: "The timeseries and statistics are illustrated in Figure 10. The best 10 configurations have in common that they use the same PBL and SW-LW schemes, namely YSU and Dudhia-RRTM. Due to the slight differences between models using different MP schemes, the 10 best performing WRF configuration only vary in MP schemes."*

Line 263: "plays an important role in  determining its performance"

*Response: Thank you for pointing out this, the errors were corrected.*

Table 8: Add a separating horizontal line between individual model configurations and ensembles.

*Response: Thanks for the comment, Table 8 was revised to a subplot in Figure 10.*

Line 254: "Evaluation of model configurations with the best *individual* performance"

*Response: Thank you for pointing out this, the errors were corrected.*

Line 276: Please explain how the classification between coastal and inland stations was conducted! Did you use an objective distance to the shoreline?

*Response: Thanks for pointing out this, coastal and inland stations are classified by the distance to the shoreline, and it was revised in lines 284-285: "Figure 11 compares the results of wind speed for coastal stations (closer than 5 km from the shoreline, 89 stations in total) and inland stations (over 5km from the shoreline, 16 stations in total), the locations of these stations are shown in Figure 1a."*

Line 277: add that this is based on the ensemble spread

*Response: Thanks for the comment, it was revised accordingly in line 286: "For both coastal and inland stations, the ensemble spread is largest among the PBL schemes, followed by SW-LW and MP schemes".*

Line 278: "consistent with the results of previous analyses *in this study*"

*Response: Thank you for pointing out this, it was revised to "which is consistent with the results of previous analyses in this study" in line 287.*

Line 279: "generally by the same magnitude" – no, figure 9 and table 9 both show that the bias is larger for inland stations.

*Response: Thanks for pointing out this, it was revised to "WRF reproduce the timeseries of wind speed reasonably, with larger overestimation for inland stations" in line 288.*

Line 280/281: "the *ensemble* spread was relatively larger for coastal stations, especially *among MP schemes and* for the first three days of the simulation period that exhibited low wind speeds"

*Response: Thanks for the comment, the errors were corrected.*

Line 282: "source of model uncertainty" – although the authors observe different sensitivities for coastal vs inland stations between parameterizations, is that a source of model uncertainty? "generates greater model differences" – compared to what?

*Response: Thank you for pointing out this, the sentence is misleading, we revised it to "As such, in addition to physical parameterizations, model performance is also influenced by ocean proximity, and WRF simulates wind speed less accurately for coastal stations compared to inland stations" in lines 290-291.*

Line 285: Is this the temporal CORR averaged over stations? Otherwise the sample-size difference between the two groups (16 coastal stations vs 89 inland stations) needs to be considered.

*Response: Thanks for the comment, the temporal CORR is averaged over the stations, so we revised in line 297: "consistent with those of previous investigations in this study, considering most of the stations are inland stations (89 out of 105 stations), this result…"*

Line 285/286: What is meant by "as there are no clear differences in wind speed values and variations"?

*Response: Thank you for pointing out this, the sentence is misleading, we revised in lines 292-293: "the CORR scores are consistently lower and the BIAS and RMSE scores are generally worse for coastal stations compared to inland stations, which indicates degraded model performance for coastal stations."*

Line 289: "previous investigations" – in this study or in different studies?

*Response: Thanks for the comment, it was revised to "are generally consistent with those of previous investigations in this study".*

Line 294: "Dudhia and RRTM … showed *worst temporal agreement* with observational data *for coastal stations*"

*Response: Thank you for pointing out this, the errors were corrected.*

Line 298: Which subfigure was this info taken from? Perhaps the authors mean "the peak *observed* wind speed at high-elevation stations (>250 m) was *1.5* m/s slower than that for low-elevation stations (<50 m)." ?

*Response: Thanks for pointing out this, it was revised in lines 304-305: "the peak observed wind speed of high-elevation stations (>250 m) is 1.5 m/s slower than that of low-elevation stations (<50 m)."*

Line 299: "the simulated peak values were generally *similar*"

*Response: Thank you for pointing out this, it was revised according to the comment.*

Lines 302/303: "Interestingly, model performances *of different parameterization types* were generally similar for

stations with different elevations, ..."

*Response: Thanks for the comment, it was revised accordingly.*

Line 304: "… smallest BIAS and RMSE values *at all elevation categories*" ?

*Response: Thank you for pointing out this, it was revised to "MYJ is always best for all elevation categories according to the CORR score, and YSU is always best according to the BIAS and RMSE scores."*

Lines 305/306: It is plausible that physics-configuration performance depends on surface topography in other cases and locations with different topography. It is not appropriate to generalize these findings to overarching WRF performance like this. The authors provide no foundation to suggest that the limited configuration dependency seen in figure 10 is result of the terrain-following coordinates.

*Response: Thank you for pointing out this, points were well taken, and we removed the statements.*

Line 314: "underlying land type and topography" – better "coast proximity and elevation" – land types include factors that were not investigated (e.g., soil texture, vegetation, roughness, canopy, etc.); topography includes aspects that were not investigated (e.g., slope steepness and directional angles of slopes)

*Response: Thanks for pointing out this, we revised it according to the comments.*

Lines 315/316: "the WRF model reproduced the temporal variation of wind speed and direction over the study area *well*" – I don't agree that "to a high degree of accuracy" is an accurate description considering the biases presented.

*Response: Thanks for the comment, we revised it to "the WRF model reproduces the temporal variation of wind speed over the study area well".*

Line 322/323: "The combined Dudhia and RRTM *radiation* schemes, and the  MYDM7 *MP* scheme both show the best *wind-speed* performances…"

*Response: Thank you for pointing out this, the errors were corrected.*

Line 325: "." (redundant)

*Response: Thanks for the comment, the errors were corrected.*

Line 327/328: "… substantially reduced overestimation of wind speed *compared to the ensemble of all 640 configurations* "

*Response: Thank you for pointing out this, the errors were corrected.*

Line 329: Most (85%) of the 105 stations are inland stations, so it is implied that the overall pattern matches the inland stations most. These conclusions are not "applicable to the inland station" but they were mostly derived from them.

*Response: Thanks for the comment, we revised it in lines 296-298: "Our comparison indicates that the*

*parameterization schemes with the best performance for inland stations are generally consistent with those of previous investigations in this study, considering most of the stations are inland stations (89 out of 105 stations), this result is not surprising".*

Lines 332/333: The best-configuration ranking might be similar, but the model results are different.

*Response: Thank you for pointing out this, we revised it according to the comment.*

Line 335: These are not "all possible combinations that are available". (1) There are other parameterization types that were not investigated (e.g., cumulus convection, land surface, etc.); (2) within the parameterization types that were assessed there are more available (e.g., Kessler and WDM5 for MP, GBM and MRF for PBL, Fu–Liou–Gu and GFDL for radiation)

*Response: Thanks for the comment, our statements are not correct and we removed them.*

Line 337: "which has rarely been used in previous simulation studies *in China*"

*Response: Thank you for pointing out this, the errors were corrected.*

Lines 339-343: This paragraph needs to be revised considering the major comments above.

*Response: Thank you for the comment, we revised it in lines 369-375: "The ensemble mean of 576 simulations in our study shows the best CORR score (0.94) in wind-speed simulation, however, this best CORR score is also result of single-model simulation with Goddard MP scheme. At the same time, the best wind-speed BIAS score (0.33 m/s) is result of the single-model simulation with MYDM7 or ETA, and the best RMSE score (0.52 m/s) is result of either the single-model simulation with Goddard, NSSL1, MYDM7, or the ensemble using the 3, 4 or 5 best simulations. Thus, model ensemble does not always provide the best performance, and model post-processing, especially the bias correction techniques are needed to be taken into consideration, which can significantly reduce the systematic errors in model simulation"*

---

## Author Comment (AC3)

**Response to referee comments**
**We are grateful for the thoughtful and constructive feedback from the referee. We have revised the text in the manuscript to answer the referee's points and we believe this revision has improved the clarity and quality of our manuscript. This response provides a complete description of the changes that have been made in response to each comment. Referee comments are shown in plain text, author responses are shown in bold blue text. All line numbers in the responses refer to locations in the revised manuscript.**

Review of "Impact of physical parameterizations on wind simulation with WRF V3.9.1.1 over the coastal regions of North China at PBL gray-zone resolution" by Yu et al.

This paper examines wind forecasts during a relatively long period of stable conditions when a haze event affected China. Surface meteorological observations are used to evaluate the WRF model's ability to predict the evolution of winds during the event. The authors conduct a number of WRF simulations (640 total), altering the PBL, radiation, and microphysics schemes to determine the sensitivity of wind speed and direction forecasts to choice of model physics. Pearson's correlation coefficient, bias, RMSE, and Taylor skill score are utilized to perform the model evaluation. Overall, the study shows the largest spread in wind speed within the PBL schemes tested, followed by radiation, and then microphysics schemes. Delineation between coastal/inland stations as well as stations at different elevation are examined to understand any model biases specific to land type and characteristics. An important finding is that WRF predicts wind speed less accurately for coastal stations compared to inland stations, and error metrics tended to degrade with increasing elevation.

Overall, this study has interesting components and would be a nice contribution to the literature especially due to the very large ensemble that was run. However, there are several aspects that should be addressed before the paper is suitable for publication. My main concerns are related to the authors' model setup, lack of some background information about the case, and insufficient physical explanations for some of their results. My general and specific comments are listed below.

*Response: Thank you for taking time out of your busy schedule to review this paper, I really appreciate all your comments and suggestions. Please find my responses in below and my revisions/corrections in the re-submitted files. Thanks again.*

Major/general comments:

1. WRF model configurations: I wonder why the authors chose to run all of the physics

parameterizations as default except for YSU, which was run using a topographic correction for surface winds and the top-down mixing option. Were the impacts of these YSU options tested? I believe that YSU is not run this way by default, so it would be good to know the impact, especially since your results show that YSU is one of the best performers. For instance, MYNN has a number of namelist tuning options, so why not modify these? Also, what is the motivation for running with the top-down mixing option in YSU if this is a statically stable case? Are stratus/fog conditions expected in some of the coastal regions? Please explain.

*Response: Thank you for pointing out this, according to this comment, we added experiments considering the effects of YSU options, and the results are presented in section 4.3, lines 341-349:*

*"4.3 Impact of options in the YSU scheme*

*The impact of different options in YSU on wind-speed simulation is illustrated in Figure 15, the simulation with the best Taylor skill score in previous investigation is selected and referred to as YSU, three extra simulations with top-down mixing option turning off (No_mix), topographic correction option turning off (No_topo), and both options turning off (No_topo_mix) are conducted for comparison. The simulated wind speed increases when we turn off the individual or both options, which enlarges the overestimation of wind speed under stable conditions in our study (Figure 15a). Further investigation indicates that turning off the two options in YSU mainly degrades model performance with worse evaluation metrics, for example, the BIAS score increases from 0.36 m/s to 0.67 m/s in No_topo, to 0.43 m/s in No_mix, and to 0.69 m/s in No_topo_mix, the RMSE scores show similar degradation to BIAS by turning off the options in YSU.".*

*In the past few years, over North China, haze events are always accompanied by fog over the coastal areas, thus the top-down mixing option was turned on in the simulation.*

As a separate but related issue, the authors use different surface layer schemes between the PBL schemes. This means that it is impossible to attribute all of the differences in results specifically to the PBL scheme. There is no discussion about this at all in the paper, although it is definitely important considering the station observations likely fall within the first model grid cell.

Furthermore, why not use the revised MM5 scheme for all of the PBL schemes? I believe that it is compatible with all of them (I may be wrong here). Regardless, it would be good to run an additional simulation to determine the impact of the surface layer scheme (which I suspect is more important than the microphysics scheme under stable conditions).

*Response: Thanks for the comment, we totally agree with the reviewer that it is important to determine the impact of surface layer scheme, however, in the WRF model, the surface layer scheme is somehow tied to the PBL scheme, it is impossible to conduct systematic experiments using different surface layer schemes with the same PBL schemes, for example, YSU cannot run with surface layer scheme other than revised MM5 scheme, MYJ PBL scheme can only be used with MYJ surface layer scheme. In this study we designed the PBL experiments including both surface layer schemes and PBL schemes, thus the results of PBL schemes are actually the results of the combination of PBL and surface layer schemes.*

*For the revised MM5 scheme, it is compatible with a lot of PBL schemes, but not all, for example, it is not compatible with the TEMP PBL schemes.*

2. Case study: The authors select a 4-day study period when stability conditions were stable to evaluate the WRF model; however, there is only a few sentence discussion about the case in Section 2.1. Although the authors do conduct many simulations, this is still a case study, and unfortunately, the authors do not present any large-scale meteorological information. It would be good to know the synoptic pattern and what type of evolution occurred; I imagine there is a pattern change over the course of the event since the regional wind speeds went from ~1 m/s to ~5 m/s according to Fig. 2. Moreover, the authors consider the impact of microphysical schemes in WRF even though this is a stable case. Are the authors anticipating cloud effects? Despite the relatively small impact of the microphysics options (e.g., Fig. 5), there are noticeable differences on 2019-01-12. I think the authors should include some metric of observed clouds (e.g., satellite images) since clearly the model is producing clouds.

*Response: Thank you for pointing out this, we removed figure 2 and added the distribution of cloud, geopotential height and winds as a new figure. The descriptions are in Lines 107-115: "Figure 2 depicts the distribution of geopotential height, winds, and cloud fraction during the study period, the geopotential height and winds data are from the ERA5 dataset (Hersbach et al., 2020), and the cloud fractions are from satellite observations of CLARA (CM SAF cLoud, Albedo and surface Radiation) product family (Karlsson et al., 2021). Figure 2 indicates that a weak high-pressure system persisted from 11 to 13 January, along with weak southwest wind in the study area, which would transport warm and wet air to the study area (Gao et al., 2016a), creating a favorable moisture condition for stable conditions and inhibiting pollutants dispersal (Zhang et al., 2014; Hua and Wu, 2022). Then the high-pressure system was replaced by strong northwest wind from 14 to 15 January 2019. The CLARA observations indicate cloud fraction exceeding 60% on 12 January at the study area, while for the rest of the time, cloud fraction is low. This stable event is used to investigate the impact of physical parameterizations of the WRF model."*

3. Physical explanations: By and large, this paper reports on the model performance with respect to near-surface wind speed and direction. However, I think the authors do not provide any physical explanations for any of their results. Ultimately, this ends up limiting the applicability of the study to other stable events in different seasonal periods and geographical locations. It would be good to address questions such as: why is QNSE so different from the other PBL schemes? How does the YSU topographic correction affect the forecast? Additionally, linking the low-level wind results to the PBL vertical structure (e.g., wind, temperature/stability, and moisture), as well as x-y spatial variability in the wind field, would be useful. For the planview, the authors could zoom in on a region where they see the largest differences in PBL schemes and overlay their observations. I think these comparisons are especially important for the different PBL schemes because the authors highlight the gray zone, which is where the choice of turbulent mixing tends to dominate the

solution.

*Response: Thanks for the comments, according to them, we added the comparisons in sections 4.1 (lines 318-330) and 4.3 (lines 341-349):*

*"4.1 Spatial distribution of wind field*

*Figure 13 shows the spatial distribution of observed and simulated wind fields during the study period, we choose 14:00 in local time as an example. The simulation using YSU, Dudhia-RRTM and WDM6 schemes is referred to as YSU, and the simulation using QNSE, Dudhia-RRTM and WDM6 is referred to as QNSE. YSU generally reproduces the wind field in the study area, especially in terms of wind speed. For example, the observed wind speed is lower on 13 January 2019, with values lower than 2 m/s in many stations, while on 15 January 2019, the observed wind speed is higher than 4 m/s in most of the stations. In the simulation with YSU, wind speed is about 2 m/s on 13 January 2019 and higher than 4 m/s on 15 January 2019 over the study area, which is close to the observation. On the contrary, simulation with QNSE fails to reproduce the distribution of wind speed, and shows strong overestimation, especially over the mountain areas of the study area (Figure 1a), for example, the peak wind speed in simulation with QNSE exceeds 20 m/s on 15 January 2019, which is more than five times greater than the observation, this overestimation is consistent with the large positive bias in previous investigation of Figure 3. For the wind-direction simulation, YSU shows degraded performance compared to wind speed, and generally fails to reproduce the wind-direction distribution for most of the stations, QNSE also fails to do so."*

*"4.3 Impact of options in the YSU scheme*

*The impact of different options in YSU on wind-speed simulation is illustrated in Figure 15, the simulation with the best Taylor skill score in previous investigation is selected and referred to as YSU, three extra simulations with top-down mixing option turning off (No_mix), topographic correction option turning off (No_topo), and both options turning off (No_topo_mix) are conducted for comparison. The simulated wind speed increases when we turn off the individual or both options, which enlarges the overestimation of wind speed under stable conditions in our study (Figure 15a). Further investigation indicates that turning off the two options in YSU mainly degrades model performance with worse evaluation metrics, for example, the BIAS score increases from 0.36 m/s to 0.67 m/s in No_topo, to 0.43 m/s in No_mix, and to 0.69 m/s in No_topo_mix, the RMSE scores show similar degradation to BIAS by turning off the options in YSU."*

Minor/specific comments:

1. Abstract, L14: Should be, "near-surface" rather than "surface".

*Response: Thank you for pointing out this, the errors were corrected in lines 14-15:"how different physical parameterizations impact simulated near-surface wind at 10-meter height over the coastal regions of North China"*

2. Abstract, L24: Please be more specific by what you mean by "land type".

*Response: Thanks for pointing out this, it was revised to "model sensitivity is also impacted by ocean*

*proximity and elevation"*

3. Abstract, L25: "For example, for the weather stations located in coastal regions." This is an awkward sentence; it would be good to combine with another sentence or rewrite.

*Response: Thanks for the comment, it was revised in lines 23-25: "for coastal stations, MYNN shows the best temporal correlation with observations among all PBL schemes, while Goddard shows the smallest bias out of SW-LW schemes"*

4. L47: Should add, "that occur at the sub-grid scale" after you say "key physical processes". Also, some of the processes listed (i.e., planetary boundary layers and cloud microphysics) aren't really processes. Please be more specific or re-phrase the beginning of the sentence.

*Response: Thank you for pointing out this, it was revised in line 50: "instead, parameterizations are needed to represent the effect of key physical processes, such as radiative transfer, turbulent mixing, and moist convection that occur at the sub-grid scale."*

5. L51: The impact of PBL schemes on wind simulations has been studied for many years (probably >60 years at this point).

*Response: Thanks for the comment, it was revised to "The impact of planetary boundary layer (PBL) schemes on wind simulations has been studied for many years".*

6. L55: Should be, "horizontal grid spacing" rather than "horizontal resolution".

*Response: Thank you for pointing out this, the errors were corrected in line 61, we also revised the Abstract accordingly.*

7. L77: I would say, "high resolution mesoscale simulation" rather than "very high resolution" considering dx=500 m is not high resolution with respect to typical LES scales (10s of meters).

*Response: Thanks for the comment, it was revised to "The investigation is conducted using the WRF model at a grid spacing of 0.5 km" in lines 83-84.*

8. L77: Please explain briefly what is mean by "gray zone" and provide references.

*Response: Thank you for pointing out this, it was revised in lines 84-86: "which belongs to the PBL "gray zone" resolution that is too fine to utilize mesoscale turbulence parameterizations and too coarse for a large-eddy-simulation (LES) scheme to resolve turbulent eddies (Shin and Hong, 2015; Honnert et al., 2016)"*

9. L82: Should be, "findings" rather than "foundings".

*Response: Thank you for pointing out this, it was corrected in line 91.*

10. L88-92: Some references would be nice here, especially for an audience who is not familiar with this region.

*Response: Thanks for the comment, it was revised in lines 96-102: "The study area is located in the central section of the "Bohai Economic Rim", which is bordered to the southeast by the Bohai Sea and to the northwest by the Yan Mountains (Figure 1a). This region traditionally hosts heavy industry and manufacturing businesses and is a significant region of economic growth and development in North China (Song et al., 2020; Zhao et al., 2020). Air quality in this area has declined over the past decades, and the frequency of winter haze events has increased due to increased pollutant emissions and favorable stable weather conditions with lower wind speed (Gao et al., 2016a; Cai et al., 2017)."*

11. L99-100: Please add references for why this wind direction indicated a weakened East Asia winter monsoon.

*Response: Thank you for pointing out this, the figure was removed and we added the synoptic description of this event in lines 107-115: "Figure 2 depicts the distribution of geopotential height, winds, and cloud fraction during the study period, the geopotential height and winds data are from the ERA5 dataset (Hersbach et al., 2020), and the cloud fractions are from satellite observations of CLARA (CM SAF cLoud, Albedo and surface Radiation) product family (Karlsson et al., 2021). Figure 2 indicates that a weak high-pressure system persisted from 11 to 13 January, along with weak southwest wind in the study area, which would transport warm and wet air to the study area (Gao et al., 2016a), creating a favorable moisture condition for stable conditions and inhibiting pollutants dispersal (Zhang et al., 2014; Hua and Wu, 2022). Then the high-pressure system was replaced by strong northwest wind from 14 to 15 January 2019. The CLARA observations indicate cloud fraction exceeding 60% on 12 January at the study area, while for the rest of the time, cloud fraction is low. This stable event is used to investigate the impact of physical parameterizations of the WRF model."*

12. L103: Why not use a newer version of WRF? Version 3.9.1.1 was released years ago (August 2017, I believe).

*Response: Thank you for the comments, WRF V3.9.1.1 was released on August 28, 2017, and we used this version for a long time in different areas across China, the overall performance is satisfactory to us, thus we continue to use it in this study to support comparative studies with previous simulations. we assume that the WRF version does not affect our results.*

13. L108-109: Were eta levels specified? It would be important to report the vertical grid spacing near the surface.

*Response: Thanks for pointing out this, it was revised in lines 123-124: "and the eta values of the first 10 levels are 0.996, 0.988, 0.978, 0.966, 0.956, 0.946, 0.933, 0.923, 0.912, and 0.901."*

14. L126: Discussing the "gray zone" here is fine, but please reference this section in the introduction so that the reader is not confused by what you mean by the terminology.

*Response: Thank you for the comments, we added the description of "gray zone' in the section 1 lines 84-86: "the PBL "gray zone" resolution that is too fine to utilize mesoscale turbulence parameterizations and too coarse for a large-eddy-simulation (LES) scheme to resolve turbulent eddies (Shin and Hong, 2015; Honnert et al., 2016)"*

*We also revised the sentences here as "The horizontal grid spacing of 0.5 km is within the PBL gray zone resolution, both PBL and LES assumptions are imperfect".*

15. L126-130: Please provide references for the "gray zone" discussion as well as the Deardorff SGS TKE scheme.

*Response: Thanks for the comments, the references for the "gray zone" are added in lines 85-86: "the PBL "gray zone" resolution that is too fine to utilize mesoscale turbulence parameterizations and too coarse for a large-eddy-simulation (LES) scheme to resolve turbulent eddies (Shin and Hong, 2015; Honnert et al., 2016)", and the reference for the Deardorff SGS TKE scheme is added in line 142: "the 1.5-order turbulence kinetic energy closure model is used to parameterize motion at the sub-grid scale (Deardorff, 1985)".*

16. L139: I disagree that the "Lin scheme is a sophisticated scheme", considering it is single moment and there are several double (or higher order) moment schemes.

*Response: Thanks for pointing out this, we removed it and rewrite the whole part in lines 151-160: "Sixteen MP schemes were applied in this study (Table 1), Lin, WSM3, WSM5, ETA, WSM6, Goddard, SBU and NSSL1 schemes are the single-moment bulk microphysical scheme, which predicts only the mixing ratios of hydrometeors (i.e., cloud ice, snow, graupel, rain, and cloud water) by assuming particle size distributions. The other eight schemes (Thompson, MYDM7, Morrison, CMA, WDM6, NSSL2, ThompsonAA and P3) use a double-moment approach, predicting not only mixing ratios of hydrometeors but also number concentrations. Among them, two types of hydrometeors are included in WSM3 (cloud water and rain), three types of hydrometeors are included in ETA (cloud water, rain and snow) and P3 (cloud water, rain and ice), four types of hydrometeors are included in WSM5 and SBU (cloud water, rain, ice and snow), five types of hydrometeors are included in Lin, WSM6, Goddard, Thompson, Morrison, CAM, WDM6 and ThompsonAA (cloud water, rain, ice, snow and graupel), six types of hydrometeors are included in MYDM7, NSSL1 and NSSL2 (cloud water, rain, ice, snow, graupel and hail)."*

17. L142: These microphysics schemes are not new, as they are over 10 years old now.

*Response: Thank you for the comments, we removed this and rewrite the whole part in lines 151-160.*

18. L144: Please define "ThompsonAA".

*Response: Thank you for the comments, ThompsonAA is short for "Aerosol-Aware Thompson", and we explained it in Table 1.*

19. L154-155: How exactly are the data screened? Please explain.

*Response: Thanks for the comments, it was explained in lines 169-171: "All data are screened before analysis in order to remove stations with data showing spurious jumps (e.g., wind speed jumps to 0 m/s due to frozen sensor). After this filtering, 105 out of 132 weather stations (Figure 1a) remained, including 89 inland stations and 16 coastal stations."*

20. L187: Only in the extreme case of QNSE did the absolute difference exceed 10 m/s; however, as written currently it sounds like many simulations have large differences.

*Response: Thank you for the comments, it was revised in lines 196-199: "reproduced by all schemes except for QNSE, with which the wind speed change is considerably larger than with all other schemes during the simulation period. Almost all the PBL schemes overestimate wind speed by 1 m/s, however, for the QNSE scheme, the largest overestimation exceeds 10 m/s during the daytime on 11 and 15 January 2019."*

21. L202-204: Do you have proof for the claim about QNSE? Are there other studies that support this claim? I wonder if MYNN used EDMF or just ED in these simulations. I know that in newer versions of WRF, EDMF is default for EDMF.

*Response: Thank you for pointing out this, the claim about QNSE is only a speculation, to avoid misleading, the sentence was revised in lines 208-210 as "For the QNSE scheme, the maximum BIAS and RMSE scores for individual stations exceed 10 m/s and 16 m/s, indicating that it has problems in reproducing wind speed under stable conditions over the study area.", For MYNN, in version 3.9.1.1, EDMF is turned off by default.*

22. L209-210: I don't understand this sentence. Wind direction depends upon the u and v-components, so one can have the correct wind speed but not the correct wind direction because the components are incorrect.

*Response: Thank you for pointing out this, it should be "the simulated wind direction was calculated using the u and v- components, the bias in modeled wind direction can be attributed to bias in the u and v- component simulations", this is redundant, so we deleted it.*

23. Fig. 5: There are repeat colors in this figures, please fix this.

*Response: Thanks for pointing out this, the errors are corrected in figures 5, 11 and 12.*

24. L252-253: It would be good here to briefly confirm/reiterate the best combination schemes for wind direction.

*Response: Thank you for pointing out this, we added the description of wind direction in lines 252-253: "For the wind direction simulation, LES combined with Dudhia-RRTM shows the best CORR score, while TEMF is the best scheme according to BIAS and RMSE scores".*

25. L265-266: The difference in statistical measures between the ensemble and WDM6 is very small, suggesting the ensemble is not improving things too much.

*Response: Thanks for pointing out this, we added discussion of this in line 373: "Thus, model ensemble does not always provide the best performance".*

26. L266-267: "reduces model bias by approximately half", compared to what?

*Response: Thank you for pointing out this, it was revised to "According to the statistic scores, ENS4 reduces model bias by approximately half compared to ENSall" in lines 276-277.*

27. L271-272: Please provide references.

*Response: Thanks for the comment, it was revised in lines 279-280: "which impacts local low-level circulation patterns and the wind distribution (Yu et al., 2013; Barlage et al., 2016)."*

28. L287: This is a broad statement. What exactly do you mean by it with respect to future model developments? Perhaps some discussion should be added to Section 4.

*Response: Thank you for pointing out this, we made it clear in lines 294-295: "This degradation may be caused by the uncertainties from the prescribed SST in our simulation, which may require a better description of atmosphere-ocean coupling process in future model development".*

29. L289: "previous investigations", please add references.

*Response: Thanks for the comment, previous investigation refers to the investigation in this study, so we revised it to "…for inland stations are generally consistent with those of previous investigations in this study," in lines 296-297 .*

30. L305: "surface topography does not induce new uncertainties into the simulation", this is simply not true. What about errors associated with horizontal diffusion and topographic data sets? These actually worse as terrain becomes steeper/more complex.

*Response: Thanks for pointing out this, the statement is misleading, so we removed it.*

31. L321-322: In general, the QNSE finding is surprising, given that it was originally developed for stable conditions. Do your results agree with other studies that have used QNSE under stable conditions? Discussion here is needed.

*Response: Thank you for the comments, we conducted further investigation of QNSE scheme in sections 4.1 and 4.2 (318-340):*

*"4.1 Spatial distribution of wind field*

*Figure 13 shows the spatial distribution of observed and simulated wind fields during the study period, we*

*choose 14:00 in local time as an example. The simulation using YSU, Dudhia-RRTM and WDM6 schemes is referred to as YSU, and the simulation using QNSE, Dudhia-RRTM and WDM6 is referred to as QNSE. YSU generally reproduces the wind field in the study area, especially in terms of wind speed. For example, the observed wind speed is lower on 13 January 2019, with values lower than 2 m/s in many stations, while on 15 January 2019, the observed wind speed is higher than 4 m/s in most of the stations. In the simulation with YSU, wind speed is about 2 m/s on 13 January 2019 and higher than 4 m/s on 15 January 2019 over the study area, which is close to the observation. On the contrary, simulation with QNSE fails to reproduce the distribution of wind speed, and shows strong overestimation, especially over the mountain areas of the study area (Figure 1a), for example, the peak wind speed in simulation with QNSE exceeds 20 m/s on 15 January 2019, which is more than five times greater than the observation, this overestimation is consistent with the large positive bias in previous investigation of Figure 3. For the wind-direction simulation, YSU shows degraded performance compared to wind speed, and generally fails to reproduce the wind-direction distribution for most of the stations, QNSE also fails to do so.*

*4.2 Vertical profile of wind speed*

*Figure 14 shows the observed and simulated vertical profile of wind speed at 08:00 and 20:00 during the study period, the location of the sounding station is illustrated in Figure 1. YSU reproduces the vertical structure of wind speed reasonably, with slightly larger model bias above the height of 15 km. Within the low levels below 2.5 km, simulated wind speed from the YSU scheme is close to the observation, with the bias lower than 2.5 m/s in most cases. Meanwhile, QNSE shows worse performance in reproducing the vertical structure of wind speed, with significant larger model bias compared to YSU. for example, QNSE overestimates the low-level (< 2.5 km) wind speed by about 10 m/s at 20:00 on 11 January 2019, and overestimate wind speed by 20 m/s at 20:00 on 11 January 2019. It is interesting to note that the simulation with QNSE is pretty similar to that with YSU at 08:00 during the study period, indicating that large difference between YSU and QNSE only occurs at specific time during the study period, which is also revealed in Figure 3a."*

32. L336-337: This broad statement about these "gray zone" resolutions being used "rarely" in previous simulation studies is not true. There are dozens (if not hundreds) of papers looking at gray zone modeling, especially in the last ~5-10 years. Please be more specific with your statement.

*Response: Thank you for pointing out this, we revised it to "to the PBL gray zone, which has rarely been used in previous simulation studies in China" in line 406.*

33. L342: To which PBL scheme parameters are you referring? It would be good to give some examples.

*Response: Thank you for pointing out this, we revised it in lines 375-377: "further tuning of the parameters within the PBL schemes, such as turbulent kinetic energy (TKE) dissipation rate, TKE diffusion factor, and turbulent length-scale coefficients is needed."*

34. Figs. 9 and 10: Please note in the caption that the y-axis range is not the same (alternatively,

make them the same).

*Response: Thanks for the comment, we make the range of y-axis identical in the revision.*

---

## Referee Report (RR1)

Title: Impact of physical parameterizations on wind simulation with WRF V3.9.1.1 under stable conditions at PBL gray-zone resolution: a case study over the coastal regions of North China
Author(s): Entao Yu et al.
MS No.: gmd-2022-53
MS type: Model evaluation paper
Iteration: Revised submission

**General Comments**

The authors have addressed most of my comments and I am happy to see that many of my suggestions were adapted in the revised manuscript. The quality of the analysis has improved; however, the discussion and figures can still be improved upon, some corrections are required (in come parts the text does not seem to agree with what the figures show), and a few concerns remain to be addressed and clarified.

**Specific Comments**

- "The stable weather event":
  - Based on Fig. 2 to me it seems that the study area is under the influence of the high-pressure system only on Jan 11[th]. Afterwards the study area experiences *north*west winds (lines 110/111 state "southwest") as the high-pressure system weakens and moves west. Except on Jan 11[th], I am not convinced that the conditions are stable based on Fig. 2. The subplot for Jan 13[th] even indicates a weak shortwave and a shift towards positive vorticity over the study area. Please also frame the area covering D03 to guide the reader where to find the "study area" in this figure. Consider assessing other variables that indicate static stability (e.g., vertical profiles of potential temperature).
  - Line 111: "which would transport warm and wet air to the study area" – weak winds do not transport air masses very far and there is no major body of water to the west of the study area. Where does the "wet air" come from? And why are moist conditions favorable for stable conditions (line 112)?
  - Under stable conditions, wind direction is (a) more variable, hence, more challenging to predict accurately, and (b) less important, as wind speeds are weak. The manuscript should frame the results accordingly.

- This manuscript includes numerous figures with subplots. I think the authors should reconsider which figures are meaningful to include in the manuscript. I.e., is it essential for the reader to see Fig 5b vs. a short description that CORR scores are indistinguishable among MP schemes at a precision of .XX?
  For all figures, ensure that Figure labels are readable on a printed paper size. (Larger font size is needed for most figures, but in particular, Fig 2, 11 & 12.)

- Figure 2:
  - Is cloud fraction data available from ERA5? Although the observational data product is interesting, for your study it might be also (maybe even more) important to see whether the initial condition model produced clouds. Several cloud products do not necessarily have to be shown in the manuscript, but it would be nice if differences among observed, ERA5 and WRF cloudiness would be explained and discussed in the manuscript.
  - Could you please add a box framing the "study area" (i.e., D03)? Is it necessary to show these maps at this scale? Perhaps the area could be a more zoomed in on the study area. (Also note that the labels are not readable.)
  - With respect to the existing topography in the area, 925hPa is a very low level. Consider showing 750 or 500hPa geopotential heights instead, which can be more descriptive for synoptic situations.

- Figures with error bar plots (Fig. 3-8): Please explain why the error bars (blue) sometimes have values outside of the range of station values (orange dots). Consider whether the metrics should be calculated over all stations (and lead times?), then averaged over simulations, or whether the metrics should be calculated over all lead times, then averaged over stations and simulations. Accordingly, show error bar ranges either across stations or simulations or both.

- It is challenging for the reader to remember which acronym belongs to each parameterization type. Therefore, it would be helpful to mention parameterization types more often throughout the manuscript please. E.g., in Fig. 11: Please use same scales on y-axis and add a label on the left for the corresponding parameterization types.

- Figure 13: The many overlapping vectors are hard to decipher. Consider using wind barbs instead of vectors or less dense vector distributions with smaller reference vector. At the scale of this figure, it is also difficult to identify direction and size of the simulated wind vectors. In figure 13 and 14 it is unclear why QNSE is shown when it was previously determined to perform poorly, and the authors decided to disregard it from further investigation.

- Figure 14: Why do you show and discuss the wind speeds up to a height of 20 km (stratospheric altitudes)? How relevant is that for your study and analysis? I further suggest re-ordering the subfigures and showing all profiles at 8:00 at the top and 20:00 at the bottom.

- Lines 214-216: "Further comparison indicates that all PBL schemes strongly overestimate the speed of north wind compared to the observations, which is the main cause of positive bias in wind speed (Figure 3)." Please elaborate. The wind speed bias could well be attributed to several wind directions. When and where does north wind occur in the simulations? Could this value be from an isolated (e.g., high elevation) station?

- Line 170: "wind speed jumps to 0 m/s due to frozen sensor" - Is there a clear "jump" when a sensor freezes? How do you distinguish between real 0m/s in wind speed vs 0m/s resulting from a frozen sensor? Do you consider measured temperatures?

- Results and conclusions: Instead of discussing each metric separately and listing the best schemes, I think it would be better to present the big picture to the reader, and discussing the schemes that have better overall scores across metrics.

- Line 291-293: "WRF simulates wind speed less accurately for coastal stations compared to inland stations." and further "the BIAS and RMSE scores are generally worse for coastal stations compared to inland stations" – actually Fig. 11 shows that bias is consistently worse for inland stations and RMSE is a tie. Revise!
  Is the larger ensemble spread for coastal stations (the ensemble being less confidence) maybe a better representation for the forecast than the narrower spread for inland stations that have a larger bias (the ensemble being overconfident)?

- Lines 302-311: Could the distribution of station elevations also be included in Fig. 1? Since different marker shapes are already used for inland vs coastal station, different marker colors could be used for the elevation categories.
  I would still like to see a synoptic discussion explaining why the high elevation station have lower wind speeds.
  It is interesting that WRF wind speeds are similar or increase while observations decrease with increasing elevation.

- The current conclusions are more of a summary. And the current discussion section contains mainly content that belongs into the results section. Since the actual discussion is limited (I only see two discussion points: the need for bias correction and other potential sources of errors), I suggest combing the discussion with the current conclusions and renaming this section to "summary and discussion".

**Technical Corrections**

Line 16: "We performed 640  simulations using  combinations of"

Lines 17/18: "Model performance is evaluated using measurements from *105* weather station observations"

Line 19: Where does the sensitivity to land surface models fit in?

Lines 37/38: To avoid repetition: "The haze events are most frequent in boreal winter and are closely related to local weather conditions with low wind speeds"

Lines 38-42: "Projections of future climate change suggest that global temperatures  and weather conditions conducive to severe haze  will increase

haze event over affecting North China (Cai et al., 2017). However, numerical models always *often* show large bias in wind prediction*s* over China […]"

Lines 44-46: "In recent years, numerical models have been used extensively to study *and forecast* the weather and climate over China, as they have high spatial and temporal resolutions, and employ sophisticated physical parameterization schemes that can reproduce detailed atmospheric and land surface processes"

Lines 46/47: "However, *studies* mostly focus on temperature or precipitation, and only a few studies have attempted to simulate winds over China" The models simulate all variables, but *studies* focus on certain variables.

Line 52: "[..] as it *can* strongly influences the model results" Your study shows that it does not always influence model results "strongly"

Line 57: "A lot of *Many* studies"

Line 59: [New sentence] "*F*or example […]"

Line 63: "is used instead. [New sentence] *T*he YSU scheme also shows […]"

Line 65: "There are also some Other studies suggesting that MYNN and ACM2 are more appropriate […]"

Lines 67-69: The word "affect" is used 3 times in two sentences – see if you can alter language more.

Lines 70/71: The citation of Cheng et al. (2013) feels out of place – it considers a different season and location than this study. Maybe use it as general citation for the sentence in line 67 or with the other citations in line 69 if appropriate.

Lines 76/77: "The *combination of* physical parameterizations are also vital to wind simulation, as they may alter the processes of atmosphere-land interactions, radiation transport, and moist convection *interact*, and *may* amplify the uncertainties in wind prediction."

Line 81: "parameterization schemes. [New sentence] *To* the best of our knowledge, the sensitivity"

Line 105: New paragraph

Figure 1: Why is there such a large difference in station number between TangShan City and QuinHuangDao City?

Figure 2 caption: "The daily averaged geopotential height (contour *lines*, units: gpm) and winds (vectors, units: m/s) *at 925 hPa* and cloud fraction (shading, units: %) at 925 hPa during 11 *through* 15 January 2019"

Line 130: "All the simulations …"

Lines 148/149: "In this study, ETA, QNSE, MYNN, Pleim-Xiu, and TEMF _SL_ schemes are chosen separately for PBL schemes of MYJ, QNSE, MYNN, ACM2, and TEMF, *respectively(?)*." All the SL schemes need references too.

Table 1: Either clarify in the caption that the schemes that share rows are not specifically assigned to each other (except for SW-LW), or change back to the previous layout, with columns "Parameterization Type", "Scheme" and "Reference", then in the rows list all PBL, MP, SW-LW schemes.

Line 176/177: "including  Pearson's correlation coefficient (CORR)" or "including the Pearson correlation coefficient (CORR)"

Lines 183/184: Perhaps also mention what the perfect scores for the other metrics are.

Line 188: "… vector notation approach*. [New sentences] C*ircular correlation coefficient …"

Line 194: "Figure 3a shows the time series of observed and simulated wind speeds at local time. *Model wind speeds are shown for different PBL schemes averaged over all other parameterization types.*" Also please clarify whether the day starts with the background shading or at the tick marks.

Line 200: "*smaller* difference" or better "more similar to measurements"

Line 203: "… Figure 3b-d*. [New sentence]* MYJ shows the best CORR score of 0.96*; [semicolon]* MYNN, ACM2 and UW are *next best*…"

Line 223/224: They are not "the same", but "very similar" (perhaps use "to a precision of 0.XX [fill in the correct precision number]")

Line 235: "Strong overestimation" seems extreme considering that total errors are often within a 1m/s margin and not much different from the other two schemes. Maybe: "RRTMG and CAM show larger overestimation than Dudhia-RRTM and Goddard at daytime peaks."

Line 242: "BIAS of -15.7" – be consistent with rounding precisions (please check everywhere in the manuscript)

Line 248/249: "the results are expected to be consistent with evaluations using other MP schemes" – why? If you see differences between Fig 9a and 9b, is it not plausible that there would also be differences with other MP schemes?

Line 251: "… Dudhia-RRTM or Goddard*. [Period]*"
"when [YSU?] is applied with RRTMG, the BIAS and RMSE scores show an obvious increasement compared with that with Dudhia-RRTM" – I can't follow.

Line 252: "[New paragraph] For the wind direction simulation …"

Line 257/258: "this indicates that a systematic variation of parameterizations is not necessary" & "this indicates that a systematic variation of parameterizations is important when focusing on these variables." (line 261) – These are my words from the previous review – it was meant as a comment on the authors' motivation for this study, however, to the reader this wording is confusing here. Please use your own words to rephrase this depending on what you wish to say.

Figure 10: Please note in the figure caption that only MP schemes are labeled because all configurations use the same PBL and SW/LW schemes. You may also highlight in your discussion that the best individual schemes can reduce the bias of the ensemble by ~50%, which is significant.

Line 275: Be consistent with abbreviations ENSall in test body vs ENS(576) in Fig. 10. (I prefer ENS(576).)

Line 281/282: "" Redundant. Skip.

Line 284/285: "Figure 11 compares the results of wind speed for coastal stations (closer than 5 km from the shoreline, 89 stations in total) and inland stations (over 5 km from the shoreline, 16 stations in total), the locations of these stations are shown in Figure 1a." – In Fig 1a looks there are more inland stations than coastal stations – should it be 89 inland & 16 coastal station?

Line 289/290: ", " It looks like this is valid everywhere.

Sections 4.1-4.4 should be part of the results (under section 3)

Line 325: "which is *similar* to the observations"

Line 337/338: Do you mean Jan 12$^{th}$ for one of the two? To me it looks almost like 30m/s on the 12$^{th}$, and like 20m/s on the 11$^{th}$.

Line 336: This sentence states generally "significant" differences between YSU and QNSE with an example; later the text says they are "pretty similar".
"For example" (capitalize at the beginning of a new sentence)

Lines 362-365: This fits in better in the methods sections where the metrics are first introduced.

Lines 367-370: This fits in better with section 3.1.5 where the ensemble results are shown.

Line 381: "during a relatively long period of stable conditions" – see my comment above

Line 391: "*followed* by MYNN"?

The authors should try to vary their language more. Many sentences start with "further investigation" and "further comparison". This phrase can be omitted.

---

## Editor Decision (ED1)

**Referee Comments**

Title: Impact of physical parameterizations on wind simulation with WRF V3.9.1.1 under stable conditions at PBL gray-zone resolution: a case study over the coastal regions of North China Author(s): Entao Yu et al. MS No.: gmd-2022-53 MS type: Model evaluation paper Iteration: Revised submission

**General Comments**

The authors have addressed most of my comments and I am happy to see that many of my suggestions were adapted in the revised manuscript. The quality of the analysis has improved; however, the discussion and figures can still be improved upon, some corrections are required (in come parts the text does not seem to agree with what the figures show), and a few concerns remain to be addressed and clarified.

**Specific Comments**

- "The stable weather event":
  - Based on Fig. 2 to me it seems that the study area is under the influence of the high-pressure system only on Jan 11th. Afterwards the study area experiences *north*west winds (lines 110/111 state "southwest") as the high-pressure system weakens and moves west. Except on Jan 11th, I am not convinced that the conditions are stable based on Fig. 2. The subplot for Jan 13th even indicates a weak shortwave and a shift towards positive vorticity over the study area. Please also frame the area covering D03 to guide the reader where to find the "study area" in this figure. Consider assessing other variables that indicate static stability (e.g., vertical profiles of potential temperature).
  - Line 111: "which would transport warm and wet air to the study area" weak winds do not transport air masses very far and there is no major body of water to the west of the study area. Where does the "wet air" come from? And why are moist conditions favorable for stable conditions (line 112)?
  - Under stable conditions, wind direction is (a) more variable, hence, more challenging to predict accurately, and (b) less important, as wind speeds are weak. The manuscript should frame the results accordingly.
- This manuscript includes numerous figures with subplots. I think the authors should reconsider which figures are meaningful to include in the manuscript. I.e., is it essential for the reader to see Fig 5b vs. a short description that CORR scores are indistinguishable among MP schemes at a precision of .XX?

For all figures, ensure that Figure labels are readable on a printed paper size. (Larger font size is needed for most figures, but in particular, Fig 2, 11 & 12.)

- Figure 2:
  - Is cloud fraction data available from ERA5? Although the observational data product is interesting, for your study it might be also (maybe even more) important to see whether the initial condition model produced clouds. Several cloud products do not necessarily have to be shown in the manuscript, but it would be nice if differences among observed, ERA5 and WRF cloudiness would be explained and discussed in the manuscript.
  - Could you please add a box framing the "study area" (i.e., D03)? Is it necessary to show these maps at this scale? Perhaps the area could be a more zoomed in on the study area. (Also note that the labels are not readable.)
  - With respect to the existing topography in the area, 925hPa is a very low level. Consider showing 750 or 500hPa geopotential heights instead, which can be more descriptive for synoptic situations.
- Figures with error bar plots (Fig. 3-8): Please explain why the error bars (blue) sometimes have values outside of the range of station values (orange dots). Consider whether the metrics should be calculated over all stations (and lead times?), then averaged over simulations, or whether the metrics should be calculated over all lead times, then averaged over stations and simulations. Accordingly, show error bar ranges either across stations or simulations or both.
- It is challenging for the reader to remember which acronym belongs to each parameterization type. Therefore, it would be helpful to mention parameterization types more often throughout the manuscript please. E.g., in Fig. 11: Please use same scales on y-axis and add a label on the left for the corresponding parameterization types.
- Figure 13: The many overlapping vectors are hard to decipher. Consider using wind barbs instead of vectors or less dense vector distributions with smaller reference vector. At the scale of this figure, it is also difficult to identify direction and size of the simulated wind vectors. In figure 13 and 14 it is unclear why QNSE is shown when it was previously determined to perform poorly, and the authors decided to disregard it from further investigation.
- Figure 14: Why do you show and discuss the wind speeds up to a height of 20 km (stratospheric altitudes)? How relevant is that for your study and analysis?
  I further suggest re-ordering the subfigures and showing all profiles at 8:00 at the top and 20:00 at the bottom.
- Lines 214-216: "Further comparison indicates that all PBL schemes strongly overestimate the speed of north wind compared to the observations, which is the main cause of positive bias in wind speed (Figure 3)." Please elaborate. The wind speed bias could well be attributed to several wind directions. When and where does north wind occur in the simulations? Could this value be from an isolated (e.g., high elevation) station?

- Line 170: "wind speed jumps to 0 m/s due to frozen sensor" Is there a clear "jump" when a sensor freezes? How do you distinguish between real 0m/s in wind speed vs 0m/s resulting from a frozen sensor? Do you consider measured temperatures?
- Results and conclusions: Instead of discussing each metric separately and listing the best schemes, I think it would be better to present the big picture to the reader, and discussing the schemes that have better overall scores across metrics.
- Line 291-293: "WRF simulates wind speed less accurately for coastal stations compared to inland stations." and further "the BIAS and RMSE scores are generally worse for coastal stations compared to inland stations" actually Fig. 11 shows that bias is consistently worse for inland stations and RMSE is a tie. Revise!
  Is the larger ensemble spread for coastal stations (the ensemble being less confidence) maybe a better representation for the forecast than the narrower spread for inland stations that have a

larger bias (the ensemble being overconfident)?

• Lines 302-311: Could the distribution of station elevations also be included in Fig. 1? Since different marker shapes are already used for inland vs coastal station, different marker colors could be used for the elevation categories.

I would still like to see a synoptic discussion explaining why the high elevation station have lower wind speeds.

It is interesting that WRF wind speeds are similar or increase while observations decrease with increasing elevation.

 The current conclusions are more of a summary. And the current discussion section contains mainly content that belongs into the results section. Since the actual discussion is limited (I only see two discussion points: the need for bias correction and other potential sources of errors), I suggest combing the discussion with the current conclusions and renaming this section to "summary and discussion".

**Technical Corrections**

Line 16: "We performed 640 ensemble simulations using multiple combinations of"

Lines 17/18: "Model performance is evaluated using measurements from 105 weather station observations"

Line 19: Where does the sensitivity to land surface models fit in?

Lines 37/38: To avoid repetition: "The haze events are most frequent in boreal winter and are closely related to local weather conditions, with haze forming in regions with low wind speeds"

Lines 38-42: "Projections of future climate change suggest that global temperatures will increase, and the frequency of conducive and weather conditions conducive to severe haze is projected to will increase substantially in response to the climate change, which in turn may increase the frequency of

haze event over affecting North China (Cai et al., 2017). However, numerical models always often show large bias in wind predictions over China [...]"

Lines 44-46: "In recent years, numerical models have been used extensively to study *and forecast* the weather and climate over China, as they have high spatial and temporal resolutions, and employ sophisticated physical parameterization schemes that can reproduce <del>detailed</del> atmospheric and land surface processes"

Lines 46/47: "However, *studies* mostly focus on temperature or precipitation, and only a few <del>studies</del> have attempted to simulate winds over China" The models simulate all variables, but *studies* focus on certain variables.

Line 52: "[..] as it *can* strongly influence<del>s the</del> model results" Your study shows that it does not always influence model results "strongly"

Line 57: "A lot of Many studies"

Line 59: [New sentence] "For example [...]"

Line 63: "is used instead. [New sentence] The YSU scheme also shows [...]"

Line 65: "There are also some Other studies suggesting that MYNN and ACM2 are more appropriate [...]"

Lines 67-69: The word "affect" is used 3 times in two sentences – see if you can alter language more.

Lines 70/71: The citation of Cheng et al. (2013) feels out of place – it considers a different season and location than this study. Maybe use it as general citation for the sentence in line 67 or with the other citations in line 69 if appropriate.

Lines 76/77: "The *combination of* physical parameterizations are also vital to wind simulation, as they may alter the processes of atmosphere-land interactions, radiation transport, and moist convection *interact*, and *may* amplify the uncertainties in wind prediction."

Line 81: "parameterization schemes. [New sentence] To the best of our knowledge, the sensitivity"

Line 105: New paragraph

Figure 1: Why is there such a large difference in station number between TangShan City and QuinHuangDao City?

Figure 2 caption: "The daily averaged geopotential height (contour *lines*, units: gpm) and winds (vectors, units: m/s) *at 925 hPa* and cloud fraction (shading, units: %) <del>at 925 hPa</del> during 11 *through* 15 January 2019"

Line 130: "All the simulations ..."

Lines 148/149: "In this study, ETA, QNSE, MYNN, Pleim-Xiu, and TEMF SL schemes are chosen separately for PBL schemes of MYJ, QNSE, MYNN, ACM2, and TEMF, *respectively(?)*." All the SL schemes need references too.

Table 1: Either clarify in the caption that the schemes that share rows are not specifically assigned to each other (except for SW-LW), or change back to the previous layout, with columns "Parameterization Type", "Scheme" and "Reference", then in the rows list all PBL, MP, SW-LW schemes.

Line 176/177: "including the Pearson's correlation coefficient (CORR)" or "including the Pearson's correlation coefficient (CORR)"

Lines 183/184: Perhaps also mention what the perfect scores for the other metrics are.

Line 188: "... vector notation approach. [New sentences] Circular correlation coefficient ..."

Line 194: "Figure 3a shows the time series of observed and simulated wind speeds at local time. *Model wind speeds are shown for different PBL schemes averaged over all other parameterization types.*" Also please clarify whether the day starts with the background shading or at the tick marks.

Line 200: "smaller difference" or better "more similar to measurements"

Line 203: "... Figure 3b-d. [New sentence] MYJ shows the best CORR score of 0.96; [semicolon] MYNN, ACM2 and UW are next best..."

Line 223/224: They are not "the same", but "very similar" (perhaps use "to a precision of 0.XX [fill in the correct precision number]")

Line 235: "Strong overestimation" seems extreme considering that total errors are often within a 1m/s margin and not much different from the other two schemes. Maybe: "RRTMG and CAM show larger overestimation than Dudhia-RRTM and Goddard at daytime peaks."

Line 242: "BIAS of -15.7" – be consistent with rounding precisions (please check everywhere in the manuscript)

Line 248/249: "the results are expected to be consistent with evaluations using other MP schemes" – why? If you see differences between Fig 9a and 9b, is it not plausible that there would also be differences with other MP schemes?

Line 251: "... Dudhia-RRTM or Goddard. [Period]"

"when [YSU?] is applied with RRTMG, the BIAS and RMSE scores show an obvious increasement compared with that with Dudhia-RRTM" – I can't follow.

Line 252: "[New paragraph] For the wind direction simulation ..."

Line 257/258: "this indicates that a systematic variation of parameterizations is not necessary" & "this indicates that a systematic variation of parameterizations is important when focusing on these variables." (line 261) – These are my words from the previous review – it was meant as a comment on the authors' motivation for this study, however, to the reader this wording is confusing here. Please use your own words to rephrase this depending on what you wish to say.

Figure 10: Please note in the figure caption that only MP schemes are labeled because all configurations use the same PBL and SW/LW schemes. You may also highlight in your discussion that the best individual schemes can reduce the bias of the ensemble by ~50%, which is significant.

Line 275: Be consistent with abbreviations ENSall in test body vs ENS(576) in Fig. 10. (I prefer ENS(576).)

Line 281/282: "The effects of these parameters on the model results are presented below." Redundant. Skip.

Line 284/285: "Figure 11 compares the results of wind speed for coastal stations (closer than 5 km from the shoreline, 89 stations in total) and inland stations (over 5 km from the shoreline, 16 stations in total), the locations of these stations are shown in Figure 1a." – In Fig 1a looks there are more inland stations than coastal stations – should it be 89 inland & 16 coastal station?

Line 289/290: ", especially among the MP schemes during 11-13 January 2019 with lower wind speed" It looks like this is valid everywhere.

Sections 4.1-4.4 should be part of the results (under section 3)

Line 325: "which is similar to the observations"

Line 337/338: Do you mean Jan 12th for one of the two? To me it looks almost like 30m/s on the 12th, and like 20m/s on the 11th.

Line 336: This sentence states generally "significant" differences between YSU and QNSE with an example; later the text says they are "pretty similar". "For example" (capitalize at the beginning of a new sentence)

Lines 362-365: This fits in better in the methods sections where the metrics are first introduced.

Lines 367-370: This fits in better with section 3.1.5 where the ensemble results are shown.

Line 381: "during a relatively long period of stable conditions" – see my comment above

Line 391: "followed by MYNN"?

The authors should try to vary their language more. Many sentences start with "further investigation" and "further comparison". This phrase can be omitted.

---

## Author Response (AR2)

**Response to referee comments**

**We are grateful for the thoughtful and constructive feedback from the reviewer. We have revised the text in the manuscript to answer the referee's points and we believe this revision has improved the clarity and quality of our manuscript. This response provides a complete description of the changes that have been made in response to each comment. Referee comments are shown in plain text, author responses are shown in bold blue text. All line numbers in the responses refer to locations in the revised manuscript.**

**Referee 1:**

**Referee Comments**

Second round review of "Impact of physical parameterizations on wind simulation with WRF V3.9.1.1 over the coastal regions of North China at PBL gray-zone resolution" by Yu et al.

In general, the authors have responded quite well to my feedback. I thank them for their attention to details. However, there are two remaining points that the authors should address before the paper is ready for publication.

*Response: Thanks, we really appreciate all your comments and suggestions. Please find my responses in below and my revisions/corrections in the re-submitted files. Thanks again.*

Specific comments:

1. Impact of different surface layer (SL) schemes: I understand that it is not possible to run all of the simulations with the same SL scheme, but it still would be nice to see the impact. For example, they could choose one configuration that is compatible with multiple SL schemes and run one more simulation while modifying the SL scheme. If the authors do not wish to run any additional simulations at this point, then they should at least provide some brief discussion on this issue.

*Response: Thank you for pointing out this, according to this comment, we added the experiment considering the effects of SL options using UW as the PBL scheme, the results are added in Lines 350-359:*

*"3.3.4 Impact of surface layer schemes*

*In the WRF model, the surface layer (SL) schemes are somehow binding with PBL schemes, it is not possible to run all PBL schemes with the same SL scheme. However, it is meaningful to conduct simulations using a specific PBL scheme that can work with multiple SL schemes to investigate the effect of SL schemes on wind simulation. Figure 16 compares the simulations results of different SL (MM5, Janjic, GFS, MYNN and PX, Table 4) schemes using UW as the PBL scheme, the other model configurations are the same as the simulation with the best Taylor skill score. Simulations with different SL schemes generally reproduce the timeseries of wind speed well, with CORR scores of about 0.93 for most schemes. However, all simulations overestimate the wind speed, especially for the Janjic scheme. At the same time, according to the BIAS and RMSE scores, MYNN shows the best performance, followed by GFS and PX schemes. Thus, SL schemes also have major influence on the wind simulation."*

2. Figure 13: Please indicate at which height you are plotting the wind field.

*Response: Thanks for the comment, we added "the height of the wind is 10 meters" in the caption of Figure 13, in Line 713.*

**Referee 2:**

The authors have addressed most of my comments and I am happy to see that many of my suggestions were adapted in the revised manuscript. The quality of the analysis has improved; however, the discussion and figures can still be improved upon, some corrections are required (in some parts the text does not seem to agree with what the figures show), and a few concerns remain to be addressed and clarified.

*Response: Thank you for reviewing this paper, we really appreciate all your comments and suggestions. Please find my responses in below and my revisions/corrections in the re-submitted files. Thanks again.*

**Specific Comments**

- "The stable weather event":
  - Based on Fig. 2 to me it seems that the study area is under the influence of the high- pressure system only on Jan 11th. Afterwards the study area experiences *north*west winds (lines 110/111 state "southwest") as the high-pressure system weakens and moves west. Except on Jan 11th, I am not convinced that the conditions are stable based on Fig. 2. The subplot for Jan 13th even indicates a weak shortwave and a shift towards positive vorticity over the study area. Please also frame the area covering D03 to guide the reader where to find the "study area" in this figure. Consider assessing other variables that indicate static stability (e.g., vertical profiles of potential temperature).
  - Line 111: "which would transport warm and wet air to the study area" – weak winds do not transport air masses very far and there is no major body of water to the west of the study area. Where does the "wet air" come from? And why are moist conditions favorable for stable conditions (line 112)?
  - Under stable conditions, wind direction is (a) more variable, hence, more challenging to predict accurately, and (b) less important, as wind speeds are weak. The manuscript should frame the results accordingly.

*Response: Thank you for pointing out this.*

1. *The previous figure covers too large an area and does not show the location of the study area, which makes it difficult for the reader to accurately identify the weather conditions over the study area. We have made the following changes to Fig. 2: (1) The outline of D03 was added, (2) the cloud fraction from ERA5 was used and (3) the surface wind at 10m height was used to illustrate the stable weather condition. We revised the analysis of the weather condition in Lines 104-112: "Figure 2 depicts the distribution of geopotential height at 500 hPa, surface winds at 10 meters and total cloud fraction from the ERA5 dataset (Hersbach et al., 2020) during the study period. A weak high-pressure system persisted from 11 to 14 January 2019 over the study area, with the geopotential height at 500 hPa of about 5400 gpm, at the surface level (10 meters), weak southwest winds occurred at the south side of the study area on 11, 12 and 14 January 2019. The surface wind speeds over the study area were weaker than 5 m/s during the first four days of the study period, then the geopotential height decreased and strong northwest winds occurred over the study area on 15 January 2019. Although there were slight differences between ERA5 and satellite products (e.g., CLARA, Karlsson et al., 2021), both datasets indicated higher cloud fraction on 11 and 14 January 2019, while for the rest of the time, the cloud fraction was low. This stable weather event is used to investigate the impact of physical parameterizations of the WRF model."*

2. *We agree with the comments that weak winds do not transport air masses very far during the study period, so we remove the statement of "which would transport warm and wet air to the study area" in the*

*manuscript.*

3. *Points are well taken and we added the following in Lines 242-244: "As wind direction is more variable but less important under stable conditions with weak wind speed, the subsequent investigations mainly focus on wind speed."*

- This manuscript includes numerous figures with subplots. I think the authors should reconsider which figures are meaningful to include in the manuscript. I.e., is it essential for the reader to see Fig 5b vs. a short description that CORR scores are indistinguishable among MP schemes at a precision of .XX? For all figures, ensure that Figure labels are readable on a printed paper size. (Larger font size is needed for most figures, but in particular, Fig 2, 11 & 12.)

*Response: Thanks very much, all the figures are replotted according to the comments, we used larger font size for Figures 2-17, we revised Figure 1, 2 and 13 according to the latter valuable comments.*

*We also revised Lines 224-226 for Figure 5: "The CORR scores are very similar for all the MP schemes at a precision of 0.01, while MYDM7 is the best scheme according to BIAS and RMSE scores, followed by P3 and ETA."*

- Figure 2:
  o Is cloud fraction data available from ERA5? Although the observational data product is interesting, for your study it might be also (maybe even more) important to see whether the initial condition model produced clouds. Several cloud products do not necessarily have to be shown in the manuscript, but it would be nice if differences among observed, ERA5 and WRF cloudiness would be explained and discussed in the manuscript.
  o Could you please add a box framing the "study area" (i.e., D03)? Is it necessary to show these maps at this scale? Perhaps the area could be a more zoomed in on the study area. (Also note that the labels are not readable.)
  o With respect to the existing topography in the area, 925hPa is a very low level. Consider showing 750 or 500hPa geopotential heights instead, which can be more descriptive for synoptic situations.

*Response: Thank you for pointing out this, we have made the following changes to Figure 2: (1) The cloud fraction from ERA5 was used, and the cloud fraction from different dataset was discussed in the manuscript, (2) we added the box indicating the boundary of D03, and zoomed to the regions of the study area, (3) the geopotential height at 500 hPa was used.*

*The manuscript was revised in Lines 104-112: "Figure 2 depicts the distribution of geopotential height at 500 hPa, surface winds at 10 meters and total cloud fraction from the ERA5 dataset (Hersbach et al., 2020) during the study period. A weak high-pressure system persisted from 11 to 14 January 2019 over the study area, with the geopotential height at 500 hPa of about 5400 gpm, at the surface level (10 meters), weak southwest winds occurred at the south side of the study area on 11, 12 and 14 January 2019. The surface wind speeds over the study area were weaker than 5 m/s during the first four days of the study period, then the geopotential height decreased and strong northwest winds occurred over the study area on 15 January 2019. Although there were slight differences between ERA5 and satellite products (e.g., CLARA, Karlsson et al., 2021), both datasets indicated higher cloud fraction on 11 and 14 January 2019, while for the rest of the time, the cloud fraction was low. This stable weather event is used to investigate the impact of physical parameterizations of the WRF model."*

*We also revised the caption of Figure 2 in Lines 658-660: "Figure 2: The daily averaged geopotential height (contour, units: gpm) at 500 hPa, total cloud fraction (shading) and surface winds at 10 meters (vectors, units: m/s) from ERA5 during 11 to 15 January 2019, the box indicates the D03 domain in Figure 1"*

- Figures with error bar plots (Fig. 3-8): Please explain why the error bars (blue) sometimes have values outside of the range of station values (orange dots). Consider whether the metrics should be calculated over all stations (and lead times?), then averaged over simulations, or whether the metrics should be calculated over all lead times, then averaged over stations and simulations. Accordingly, show error bar ranges either across stations or simulations or both.

*Response: Thank you for pointing out this, in Figures 3-8, the blue bars indicate the statistic scores (CORR, BIAS and RMSE) of a specific physical scheme, which are calculated by first averaging over all stations, then over the simulations using that physical scheme; while the orange dots indicate the spread of the statistic scores across all the stations. In some cases (e.g., RMSEs of MP schemes in Figure 5), the results of station mean (ensemble) are better than any of the individual station, thus the statistic scores for the station mean (blue) are outside of the spread for all stations (orange dots). According to the comments from the first round, the orange dots were added to illustrate the distribution across the 105 stations for different physical schemes, thus for each station, the metrics are averaged over simulations using the same physical scheme.*

- It is challenging for the reader to remember which acronym belongs to each parameterization type. Therefore, it would be helpful to mention parameterization types more often throughout the manuscript please. E.g., in Fig. 11: Please use same scales on y-axis and add a label on the left for the corresponding parameterization types.

*Response: Thanks for the comments, we revised Figure 11 according to the comments, and we mentioned the parameterization types more often in the manuscript in the following:*

*Line 272: The result indicates that ensemble mean of four simulations with WDM6, Goddard, NSSL1 and MYDM7 MP schemes shows the best BIAS and RMSE scores.*

*Line 315: "the same simulation but with QNSE PBL scheme (i.e., using QNSE, Dudhia-RRTM and WDM6 schemes) is used for comparison between the simulations with good and poor performances"*

*Line 385: "The simulation using YSU PBL, WDM6 MP and Dudhia-RRTM SW-LW schemes shows the best performance with the highest Taylor skill score."*

- Figure 13: The many overlapping vectors are hard to decipher. Consider using wind barbs instead of vectors or less dense vector distributions with smaller reference vector. At the scale of this figure, it is also difficult to identify direction and size of the simulated wind vectors.

*Response: Thanks for the comments, in the revision, we used less dense vectors so that there were fewer overlapping vectors, and we made the simulated wind vectors larger and clear.*

- In figure 13 and 14 it is unclear why QNSE is shown when it was previously determined to perform poorly, and the authors decided to disregard it from further investigation.

*Response: Thanks for the comments, according to the comments from the first round, we added comparisons of*

*YSU and QNSE to illustrate the differences between schemes with good and poor performances.*

- Figure 14: Why do you show and discuss the wind speeds up to a height of 20 km (stratospheric altitudes)? How relevant is that for your study and analysis? I further suggest re-ordering the subfigures and showing all profiles at 8:00 at the top and 20:00 at the bottom.

*Response: Thanks for the comments, we re-ordered the subfigures in Figure 14, and the profiles at 08:00 were shown at the upper panel and profiles at 20:00 were shown at the lower panel, and we limited the profiles to the height of 5km. we revised the manuscript in Lines 334-340: "YSU reproduces the vertical structure of wind speed reasonably, for example, within the low levels below 2.5 km, the simulated wind speed from the YSU scheme is similar to the observation, with model bias lower than 2.5 m/s in most cases. Meanwhile, QNSE shows worse performance in reproducing the vertical structure of wind speed, with large model bias compared to YSU. QNSE overestimates the wind speed by almost 20 m/s at 20:00, 11 January 2019, and by 30 m/s at 20:00, 12 January 2019. It is interesting to note that at 08:00, the simulations using QNSE show smaller differences with that using YSU, thus the largest differences between YSU and QNSE generally occur at specific time during the study period, which is also revealed in Figure 3a."*

- Lines 214-216: "Further comparison indicates that all PBL schemes strongly overestimate the speed of north wind compared to the observations, which is the main cause of positive bias in wind speed (Figure 3)." Please elaborate. The wind speed bias could well be attributed to several wind directions. When and where does north wind occur in the simulations? Could this value be from an isolated (e.g., high elevation) station?

*Response: Thanks for the comments, in the wind rose charts, the circles represent the relative frequency (%), and the colors represent wind speed (m/s), we can find in Figure 4 that for the observations, the majority of the north winds are lower than 2 m/s, at the same time, in most of the simulations, the frequency of north winds is similar to the observations, but the speed is higher than 5m/s, which may be the main cause of positive bias in wind speed. To avoid misleading, we revised it as "Further comparison indicates that all PBL schemes strongly overestimate the speed of north wind compared to the observations, which may be the main cause of positive bias in wind speed (Figure 3)" in Lines 214-216.*

- Line 170: "wind speed jumps to 0 m/s due to frozen sensor" - Is there a clear "jump" when a sensor freezes? How do you distinguish between real 0m/s in wind speed vs 0m/s resulting from a frozen sensor? Do you consider measured temperatures?

*Response: Thank you for the comments, the quality control of station observations was completed by Heibei Climate Center (two authors work at this agency), they were able to distinguish the jumps of the wind speed resulting from a frozen sensor because the sensors were operated by them.*

- Results and conclusions: Instead of discussing each metric separately and listing the best schemes, I think it would be better to present the big picture to the reader, and discussing the schemes that have better overall scores across metrics.

*Response: Thanks very much, according to the comments, we added the following in Lines 385-387: "The simulation using YSU PBL, WDM6 MP and Dudhia-RRTM SW-LW schemes shows the best performance with the*

*highest Taylor skill score."*

- Line 291-293: "WRF simulates wind speed less accurately for coastal stations compared to inland stations." and further "the BIAS and RMSE scores are generally worse for coastal stations compared to inland stations" – actually Fig. 11 shows that bias is consistently worse for inland stations and RMSE is a tie. Revise! Is the larger ensemble spread for coastal stations (the ensemble being less confidence) maybe a better representation for the forecast than the narrower spread for inland stations that have a larger bias (the ensemble being overconfident)?

*Response: Thanks very much and sorry for the error, and we totally agreed with the comments, we revised in Lines 292-294: "The statistic scores are also illustrated in Figure 11, the CORR scores are consistently lower for coastal stations compared to inland stations, while the BIAS scores are generally worse for the inland stations. Thus, the model performance tends to degrade for the inland stations according to the BIAS scores."*

- Lines 302-311: Could the distribution of station elevations also be included in Fig. 1? Since different marker shapes are already used for inland vs coastal station, different marker colors could be used for the elevation categories. I would still like to see a synoptic discussion explaining why the high elevation station have lower wind speeds. It is interesting that WRF wind speeds are similar or increase while observations decrease with increasing elevation.

*Response: Thanks for the comments, we made changes to Figure 1, and added the information of station elevations with different marker size. The caption of Figure 1 was revised to "Solid yellow and blue circles in (a) represent coastal (16 stations in total) and inland stations (89 stations in total), the size of circles represents the station elevations, and white circle represent the sounding station."*
*Further investigations were needed to confirm the mechanism for lower wind speeds of the high elevation stations, and the mismatch between the observations and the WRF simulations. We added the following in Lines 305-307: "Further investigations are needed to reveal the underlying mechanisms for lower wind speed of high elevation stations and the mismatch between observations and model simulations"*

- The current conclusions are more of a summary. And the current discussion section contains mainly content that belongs into the results section. Since the actual discussion is limited (I only see two discussion points: the need for bias correction and other potential sources of errors), I suggest combing the discussion with the current conclusions and renaming this section to "summary and discussion".

*Response: Thanks for the comments, we revised the manuscript accordingly in Lines 369-404:*
*"4. Summary and discussion*
*……*
*As in our study, the model ensemble does not always provide the best performance, model post-processing, especially the bias correction techniques are needed to be taken into consideration, which can significantly reduce the systematic errors in model simulation. In addition, the PBL schemes play a dominant role in wind simulation, further tuning of the parameters within the PBL schemes, such as turbulent kinetic energy (TKE) dissipation rate, TKE diffusion factor, and turbulent length-scale coefficients is needed. In addition to PBL and SW-LW schemes, LSM and SL schemes also has unneglectable influence on the wind simulation, which should be taken into consideration in future studies. Finally, it is worth pointing out that the presented findings in this study could be*

*unique to the meteorological setup of the event, the location, the input dataset, the domain setup, and other unchanged parameterization types or model settings."*

**Technical Corrections**

**Technical Corrections**

Line 16: "We performed 640  simulations using  combinations of"

*Response: Thanks for the comments, it was revised to "We performed 640 simulations using combinations of 10 planetary boundary layer (PBL), 16 microphysics" in Line 16.*

Lines 17/18: "Model performance is evaluated using measurements from *105* weather station observations"

*Response: Thanks for the comments, it was revised to "Model performance is evaluated using measurements from 105 weather station observations" in Line 17.*

Line 19: Where does the sensitivity to land surface models fit in?

*Response: Thanks for the comments, the effect of LSM and SL schemes was not investigated in a systematic way as other schemes, thus we were not able to compare the sensitivity of LSM and SL to that of PBL, MP and SW-SW schemes. We added the discussion in Lines 401-402: "In addition to PBL and SW-LW schemes, LSM and SL schemes also has unneglectable influence on the wind simulation, which should be taken into consideration in future studies."*

Lines 37/38: To avoid repetition: "The haze events are most frequent in boreal winter and are closely related to local weather conditions with low wind speeds"

*Response: Thanks for the comments, it was revised to "The haze events are most frequent in boreal winter and are closely related to local weather conditions with low wind speeds" in Lines 37-38.*

Lines 38-42: "Projections of future climate change suggest that global temperatures  and weather conditions conducive to severe haze  will increase  affecting North China (Cai et al., 2017). *H*owever, numerical models  *often* show large bias in wind prediction*s* over China […]"

*Response: Thanks for the comments, it was revised to "Projections of future climate change suggest that global temperatures and weather conditions conducive to severe haze will increase affecting North China (Cai et al., 2017). However, numerical models often show large bias in wind prediction over China (Gao et al., 2016b; Zhao et al., 2016; Pan et al., 2021)," in Lines 38-41.*

Lines 44-46: "In recent years, numerical models have been used extensively to study *and forecast* the weather and climate over China, as they have high spatial and temporal resolutions, and employ sophisticated physical parameterization schemes that can reproduce  atmospheric and land surface processes"

*Response: Thanks for the comments, it was revised to "In recent years, numerical models have been used extensively to study and forecast the weather and climate over China, as they have high spatial and temporal*

*resolutions, and employ sophisticated physical parameterization schemes that can reproduce atmospheric and land surface processes (Wang et al., 2011; Zhou et al., 2019; Kong et al., 2021)." in Lines 43-45.*

Lines 46/47: "However, *studies* mostly focus on temperature or precipitation, and only a few  have attempted to simulate winds over China" The models simulate all variables, but *studies* focus on certain variables.

*Response: Thanks for the comments, it was revised to "However, studies mostly focus on temperature or precipitation, and only a few have attempted to simulate winds over China (Li et al., 2019; Xia et al., 2019; Pan et al., 2021)." in Lines 45-47.*

Line 52: "[..] as it *can* strongly influence model results" Your study shows that it does not always influence model results "strongly"

*Response: Thanks for the comments, it was revised to "Different physical parameterization schemes depict natural phenomena to different degrees of accuracy and choosing appropriate combinations is important, as it can strongly influences model results (Yu et al., 2011; Gómez-Navarro et al., 2015; Stegehuis et al., 2015; Gao et al., 2016b; Yang et al., 2017; Taraphdar et al., 2021)." in Lines 49-52.*

Line 57: " *Many* studies"

*Response: Thanks for the comments, it was revised to "Many studies indicate an overestimation of wind speed in WRF simulations with different PBL schemes (Jiménez and Dudhia, 2012; Carvalho et al., 2014a, b; Pan et al., 2021; Gholami et al., 2021; Dzebre and Adaramola, 2020)," in Lines 56-58.*

Line 59: [New sentence] "*F*or example […]"

*Response: Thanks, we revised the paper according to the comments in Line 58.*

Line 63: "is used instead. [New sentence] *T*he YSU scheme also shows […]"

*Response: Thanks, we revised the paper according to the comments in Line 62.*

Line 65: " Other studies suggest that MYNN and ACM2 are more appropriate […]"

*Response: Thanks, we revised the paper according to the comments in Lines 64-65.*

Lines 67-69: The word "affect" is used 3 times in two sentences – see if you can alter language more.

*Response: Thanks for the comments, it was revised to "The performance of wind simulation is also influenced by the choice of cloud microphysics (MP) parameterizations. Cloud microphysical processes, such as moisture evaporation and condensation, can change thermodynamic and dynamic interactions in the atmosphere (Rajeevan et al., 2010; Santos-Alamillos et al., 2013; Li et al., 2020), then affect the vertical distribution of heat and wind fields close to the surface." in Lines 66-69.*

Lines 70/71: The citation of Cheng et al. (2013) feels out of place – it considers a different season and location than this study. Maybe use it as general citation for the sentence in line 67 or with the other citations in line 69 if appropriate.

*Response: Thanks for the comments, we added Cheng et al. (2013) to the general citations, now it was revised to "Cloud microphysical processes, such as moisture evaporation and condensation, can change thermodynamic and dynamic interactions in the atmosphere (Rajeevan et al., 2010; Cheng et al., 2013; Santos-Alamillos et al., 2013; Li et al., 2020)" in Lines 66-68.*

Lines 76/77: "The *combination of* physical parameterizations are also vital to wind simulation, as  the processes of atmosphere-land interactions, radiation transport, and moist convection *interact*, and *may* amplify the uncertainties in wind prediction."

*Response: Thanks for the comments, it was revised to "The combinations of physical parameterizations are also vital to wind simulation, as the processes of atmosphere-land interactions, radiation transport, and moist convection interact, and may amplify the uncertainties in wind prediction." in Lines 74-75.*

Line 81: "parameterization schemes. [New sentence] *To*  our knowledge, the sensitivity"
*Response: Thanks, we revised the paper according to the comments in Line 79.*

Line 105: New paragraph
*Response: Thanks, we revised the paper according to the comments.*

Figure 1: Why is there such a large difference in station number between TangShan City and QuinHuangDao City?
*Response: Thanks for the comments, we got the observations form Heibei Climate Center, which belongs to the China Meteorology Administration (CMA), during the study period, there are more observations available at QinHuangDao City than other cities.*

Figure 2 caption: "The daily averaged geopotential height (contour *lines*, units: gpm) and winds (vectors, units: m/s) *at 925 hPa* and cloud fraction (shading, units: %)  during 11 *through* 15 January 2019"
*Response: Thanks for the comments, the caption was revised to "The daily averaged geopotential height (contour lines, units: gpm) at 500 hPa, total cloud fraction (shading) and surface winds at 10 meters (vectors, units: m/s) from ERA5 during 11 through 15 January 2019, the box indicates D03 in Figure 1"*

Line 130: "All  simulations …"
*Response: Thanks, we revised the paper according to the comments in Line 127.*

Lines 148/149: "In this study, ETA, QNSE, MYNN, Pleim-Xiu, and TEMF SL schemes are chosen separately for PBL schemes of MYJ, QNSE, MYNN, ACM2, and TEMF, *respectively(?)*." All the SL schemes need references too.

*Response: Thanks, we revised the paper according to the comments in Lines 145-146: "In this study, ETA, QNSE, MYNN, Pleim-Xiu, and TEMF SL schemes are chosen separately for PBL schemes of MYJ, QNSE, MYNN, ACM2, and TEMF, respectively". References were listed in Table 4.*

Table 1: Either clarify in the caption that the schemes that share rows are not specifically assigned to each other

(except for SW-LW), or change back to the previous layout, with columns "Parameterization Type", "Scheme" and "Reference", then in the rows list all PBL, MP, SW-LW schemes.

*Response: Thanks, we revised the caption of Table 1 to "Table 1: List of microphysics (MP), planetary boundary layer (PBL), and shortwave-longwave radiation (SW-LW) schemes investigated in the 640 simulations, schemes that share rows are not specifically assigned to each other (except for SW-LW)."*

Line 176/177: "including  Pearson's correlation coefficient (CORR)" or "including the Pearson correlation coefficient (CORR)"

*Response: Thanks for the comments, it was revised to "Several metrics are employed for evaluating the performance of each model configuration, including the Pearson correlation coefficient (CORR)" in Line 172.*

Lines 183/184: Perhaps also mention what the perfect scores for the other metrics are.

*Response: Thanks for the comments, we added "while higher CORR, lower BIAS and RMSE scores indicate better model simulations." in Line 184.*

Line 188: "… vector notation approach. [New sentences] Circular correlation coefficient …"

*Response: Thanks, we revised the paper according to the comments in Line 188.*

Line 194: "Figure 3a shows the time series of observed and simulated wind speeds at local time. *Model wind speeds are shown for different PBL schemes averaged over all other parameterization types.*" Also please clarify whether the day starts with the background shading or at the tick marks.

*Response: Thanks for the comments, it was revised to "Figure 3a shows the time series of observed wind speed in local time. Model wind speeds are shown for different PBL schemes averaged over all other parameterization types." in Lines 194-195.*

*We clarified in the caption: "Figure 3: (a) Time series of observed and simulated wind speeds (m/s) and the corresponding statistics of (b) CORR, (c) BIAS and (d) RMSE for the PBL schemes. In a, the frequency of wind speed is hourly, and the tick marks in x-axis indicate 12:00 in local time of that day, for each PBL scheme, the average is calculated over the 105 stations and then over all the simulations with that scheme; the dots in b, c and d represent the range across the stations, for each station, the metrics are calculated by averaging all the simulations with the specific PBL scheme".*

Line 200: "*smaller* difference" or better "more similar to measurements"

*Response: Thanks for the comments, it was revised to "while YSU is more similar to the measurements during 14-15 January 2019." in Line 200.*

Line 203: "… Figure 3b-d. [New sentence] MYJ shows the best CORR score of 0.96; [semicolon] MYNN, ACM2 and UW are *next best*…"

*Response: Thanks for the comments, it was revised to "The statistics of CORR, BIAS, and RMSE are illustrated in Figure 3b-d. MYJ shows the best CORR score of 0.96; MYNN, ACM2 and UW are next best according to this verification score" in Lines 203-204.*

Line 223/224: They are not "the same", but "very similar" (perhaps use "to a precision of 0.XX [fill in the correct precision number]")

*Response: Thanks for the comments, it was revised to "The CORR scores are very similar for all the MP schemes at a precision of 0.01, while MYDM7 is the best scheme according to BIAS and RMSE scores" in Line 225.*

Line 235: "Strong overestimation" seems extreme considering that total errors are often within a 1m/s margin and not much different from the other two schemes. Maybe: "RRTMG and CAM show larger overestimation than Dudhia-RRTM and Goddard at daytime peaks."

*Response: Thanks for the comments, it was revised to "RRTMG and CAM show a larger overestimation than Dudhia-RRTM and Goddard at daytime peaks." in Lines 235-236.*

Line 242: "BIAS of -15.7" – be consistent with rounding precisions (please check everywhere in the manuscript)

*Response: Thanks for the comments, it was revised to "the RRTMG scheme shows the best BIAS of -15.69°, and Dudhia-RRTM shows the best RMSE of 61.13°." in Line 242.*
*In Lines 206-207: "for the schemes except for QNSE, the range of CORR is 0.18-0.88, the range of BIAS is -2.10-2.91 m/s, and the range of RMSE is 0.79-3.85 m/s"*
*Line 218: "while TEMF shows the best BIAS and RMSE scores of -11.33° and 56.19°."*

Line 248/249: "the results are expected to be consistent with evaluations using other MP schemes" – why? If you see differences between Fig 9a and 9b, is it not plausible that there would also be differences with other MP schemes?

*Response: Thanks for the comments, as we cannot show all the combinations of PBL, MP and SW-LW schemes, the results of combinations using MP scheme of MYDM7 and P3 were chosen as examples, and we can find that the interactions among parameterization schemes are also important. We expected that there should be differences between MYDM7 and P3 results (fig 9a and 9b), however, the results from the investigation should be consistent (the interactions among parameterization schemes are also important). To avoid misleading, we revised as "thus for each MP scheme, a total of 36 simulations (excluding QNSE) are evaluated. For wind speed simulations with both MYDM7 and P3, MYJ shows the best CORR score, while YSU shows the lowest BIAS and RMSE scores".*

Line 251: "… Dudhia-RRTM or Goddard. [Period]"
"when [YSU?] is applied with RRTMG, the BIAS and RMSE scores show an obvious increasement compared with that with Dudhia-RRTM" – I can't follow.

*Response: Thanks for the comments, it was revised to "However, it is worth noting that YSU shows pretty low BIAS (< 0.4 m/s) and RMSE (<0.6 m/s) scores only when combined with the Dudhia-RRTM or Goddard schemes, when it is combined with RRTMG schemes, the BIAS and RMSE scores increase a lot" in Lines 257-259.*

Line 252: "[New paragraph] For the wind direction simulation …"
*Response: Thanks, we revised the paper according to the comments in Line 252.*

Line 257/258: "this indicates that a systematic variation of parameterizations is not necessary" & "this indicates that a systematic variation of parameterizations is important when focusing on these variables." (line 261) – These are my words from the previous review – it was meant as a comment on the authors' motivation for this study, however, to the reader this wording is confusing here. Please use your own words to rephrase this depending on what you wish to say.

*Response: Thanks for the comments, it was revised to "Overall, for BIAS and RMSE scores of wind speed, within each PBL scheme, the same SW-LW group ranks best, and within each SW-LW group, the same PBL scheme ranks best. For example, no matter which SW-LW group, YSU is always the best, which indicate the good performance of YSU. However, it is worth noting that YSU shows pretty low BIAS (< 0.4 m/s) and RMSE (<0.6 m/s) scores only when combined with the Dudhia-RRTM or Goddard schemes, when it is combined with RRTMG schemes, the BIAS and RMSE scores increase a lot. For the wind direction simulation, the pattern is different from that of wind speed. For example, for BIAS and RMSE scores, the best PBL scheme depends on the choice of SW-LW schemes, which indicates the influence of scheme interaction on model performance in wind direction simulations." in Lines 255-261.*

Figure 10: Please note in the figure caption that only MP schemes are labeled because all configurations use the same PBL and SW/LW schemes. You may also highlight in your discussion that the best individual schemes can reduce the bias of the ensemble by ~50%, which is significant.

*Response: Thanks for the comments, the caption was revised to "Time series of wind speed (m/s) from observation and different ensembles, and (b) CORR, (c) BIAS and (d) RMSE scores for the best 10 simulations along with different ensembles. The shading in a shows the spread for ENS(4) and ENS(576). As the best 10 simulations use the same PBL (YSU) and SW-LW (Dudhia-RRTM) schemes, only the MP schemes are labeled in the figure." Lines 274-276: "According to the statistic scores, ENS(4) reduces model bias by approximately half compared to ENS(576). At the same time, the best individual schemes (NSSL1, MYDM7, P3 and ETA) can also reduce the bias by ~50%."*

Line 275: Be consistent with abbreviations ENSall in test body vs ENS(576) in Fig. 10. (I prefer ENS(576).)

*Response: Thanks, we revised the figure and text according to the comments, we used ENS(576). The revision are in Lines 271-276: "Figure 10 also shows the time series of wind speed from ensemble of 4 [ENS(4)) and all 576 simulations [ENS(576)], the spread of ENS(4) is significantly lower than that of ENS(576), and ENS(4) shows smaller difference with the observation compared to ENS(576). According to the statistic scores, ENS(4) reduces model bias by approximately half compared to ENS(576). At the same time, the best individual schemes (NSSL1, MYDM7, P3 and ETA) can also reduce the bias by ~50%."*

Line 281/282: "" Redundant. Skip.

*Response: Thanks, we revised the paper according to the comments.*

Line 284/285: "Figure 11 compares the results of wind speed for coastal stations (closer than 5 km from the shoreline, 89 stations in total) and inland stations (over 5 km from the shoreline, 16 stations in total), the locations of these stations are shown in Figure 1a." – In Fig 1a looks there are more inland stations than coastal stations – should it be 89 inland & 16 coastal station?

*Response: Thanks for the comments and sorry for the mistakes, we corrected them as "Figure 11 compares the results of wind speed for coastal stations (closer than 5 km from the shoreline, 16 stations in total) and inland stations (over 5 km from the shoreline, 89 stations in total), the locations of these stations are shown in Figure 1a.".*

Line 289/290: ", " It looks like this is valid everywhere.

*Response: Thanks, we revised the paper according to the comments.*

Sections 4.1-4.4 should be part of the results (under section 3)

*Response: Thanks, we revised the paper according to the comments.*

Line 325: "which is *similar* to the observations"

*Response: Thanks, we revised the paper according to the comments in Line 325.*

Line 337/338: Do you mean Jan 12$^{th}$ for one of the two? To me it looks almost like 30m/s on the 12$^{th}$, and like 20m/s on the 11$^{th}$.

*Response: Thanks for the comments and sorry for the mistakes, we revised in Lines 337-338: "QNSE overestimates the wind speed by almost 20 m/s at 20:00, 11 January 2019, and by 30 m/s at 20:00, 12 January 2019."*

Line 336: This sentence states generally "significant" differences between YSU and QNSE with an example; later the text says they are "pretty similar".

*Response: Thanks for the comments and sorry for the mistakes, we revised in Lines 338-340: "It is interesting to note that at 08:00, the simulations using QNSE show smaller differences with that using YSU, thus the largest differences between YSU and QNSE generally occur at specific time during the study period, which is also revealed in Figure 3a."*

"For example" (capitalize at the beginning of a new sentence)

*Response: Thanks, we revised the paper according to the comments.*

Lines 362-365: This fits in better in the methods sections where the metrics are first introduced.

*Response: Thanks, we revised the paper according to the comments in Lines 173-176: "CORR is a measure of the strength and direction of the linear relationship between simulation and observation, BIAS is a measure of mean difference between simulation and observation, and RMSE is the square root of the average of the set of squared differences between simulation and observation, thus each of this score gives a partial view of the model performance."*

Lines 367-370: This fits in better with section 3.1.5 where the ensemble results are shown.

*Response: Thanks, we revised the paper according to the comments in Lines 276-279: "It is worth to mention that the best CORR score of ENS(576) is also result of single model simulation with Goddard MP scheme. At the same*

*time, the best BIAS score (0.33 m/s) is result of the single model simulation with MYDM7 or ETA, and the best RMSE score (0.52 m/s) is result of either the single model simulation with Goddard, NSSL1, MYDM7, or the ensemble using the 3, 4 or 5 best simulations."*

Line 381: "during a relatively long period of stable conditions" – see my comment above

*Response: Thanks for the comments, it was revised to "we investigate wind simulations under stable conditions when a haze event affected North China" in Line 370.*

Line 391: "*followed* by MYNN"?

*Response: Thanks for the comments, it was revised to "YSU is the best scheme according to the BIAS and RMSE scores (0.45 m/s and 0.61 m/s), followed by MYNN (0.55 m/s and 0.70 m/s)" in Lines 383-384.*

The authors should try to vary their language more. Many sentences start with "further investigation" and "further comparison". This phrase can be omitted.

*Response: Thanks for the comments, we checked the manuscript and delete the phrase of "further investigation" and "further comparison".*